# Estimating and Reporting Uncertainties in Remotely Sensed Atmospheric Composition and Temperature

Thomas von Clarmann[1], Douglas A. Degenstein[2], Nathaniel J. Livesey[3], Stefan Bender[4], Amy Braverman[3], André Butz[5], Steven Compernolle[6], Robert Damadeo[7], Seth Dueck[2], Patrick Eriksson[8], Bernd Funke[9], Margaret C. Johnson[3], Yasuko Kasai[10], Arno Keppens[6], Anne Kleinert[1], Natalya A. Kramarova[11], Alexandra Laeng[1], Bavo Langerock[6], Vivienne H. Payne[3], Alexei Rozanov[12], Tomohiro O. Sato[10], Matthias Schneider[1], Patrick Sheese[13], Viktoria Sofieva[14], Gabriele P. Stiller[1], Christian von Savigny[15], and Daniel Zawada[2]

[1]Karlsruhe Institute of Technology, Institute of Meteorology and Climate Research,
Karlsruhe, Germany
[2]University of Saskatchewan,
Saskatoon, Canada
[3]NASA Jet Propulsion Laboratory and California Institute of Technology,
Pasadena, CA, USA
[4]Department of Physics, Norwegian University of Science and Technology (NTNU), NO-7491 Trondheim, Norway
[5]Institut für Umweltphysik, Department of Physics and Astronomy, Universität Heidelberg
[6]Department of Atmospheric Composition, Royal Belgian Institute for Space Aeronomy (BIRA-IASB), 1180 Brussels, Belgium
[7]NASA Langley Research Center, Hampton, VA
[8]Chalmers University of Technology, SE-412 96 Gothenburg, Sweden
[9]Instituto de Astrofísica de Andalucía, CSIC
[10]National Institute of Information and Communications Technology (NICT)
, 4-2-1 Nukui-kita, Koganei, Tokyo 184-8795, JAPAN.
[11]NASA Goddard Space Flight Center, Greenbelt, MD, USA
[12]Institute of Environmental Physics (IUP), University of Bremen, Germany
[13]Department of Physics, University of Toronto
[14]Finnish Meteorological Institute, Helsinki, Finland
[15]Greifswald University

**Correspondence:** T. von Clarmann (thomas.clarmann@kit.edu)

**Abstract.** Remote sensing of atmospheric state variables typically relies on the inverse solution of the radiative transfer equation. An adequately characterized retrieval provides information on the uncertainties of the estimated state variables as well as on how any constraint or a priori assumption affects the estimate. Reported characterization data should be intercomparable between different instruments, empirically validatable, grid-independent, usable without detailed knowledge of the instrument or retrieval technique, traceable, and still have reasonable data volume. The latter may force one to work with representative rather than individual characterization data. Many errors derive from approximations and simplifications used in real-world retrieval schemes, which are reviewed in this paper, along with related error estimation schemes. The main sources of uncertainty are measurement noise, calibration errors, simplifications and idealizations in the radiative transfer model and retrieval scheme, auxiliary data errors, and uncertainties in atmospheric or instrumental parameters. Some of these errors affect the result in a

random way, while others chiefly cause a bias or are of mixed character. Beyond this, it is of utmost importance to know the influence of any constraint and prior information on the solution. While different instruments or retrieval schemes may require different error estimation schemes, we provide a list of recommendations which should help to unify retrieval error reporting.

## 1 Introduction

Observations from remote sensing instruments are central to many studies in atmospheric science. The robustness of the conclusions drawn in these studies is critically dependent on the characteristics of the reported data, including their uncertainty, resolution, and dependence on any a priori information. Adequate communication of these data characteristics is therefore essential. Further, when, as is increasingly the case, observations from multiple sensors are considered, it is important that these characteristics be described in a manner that allows for appropriate intercomparison of those characteristics and the ob-
servations they describe. In the satellite community, however, the definition of what constitutes "adequate communication" is far from uniform. Currently, multiple retrieval methods are used by different remote sounding instrument groups, and various approaches to error or uncertainty estimation are applied. Furthermore, reported uncertainties are not always readily intercomparable. For example, the metrics used as uncertainty values for a data set might not be properly defined (as, say, $1\sigma$ or $2\sigma$ values, or as an appropriate confidence interval), uncertainty values might not be adequately described as "random" or "sys-
tematic" in nature (let alone any more nuanced description of inter-error correlations), spatial resolution information or the influence of a priori content might not be given, etc. The mischief of inconsistent data characterization in the quantitative use of data from multiple instruments is obvious. Two prominent examples from this plethora of problem areas are error-weighted multi-instrument time series and trend calculations, or data merging.

This paper discusses these issues and proposes a common framework for the appropriate communication of uncertainty and
20 other measurement characteristics.

This review has been undertaken under the aegis of the 'Towards Unified Error Reporting' (TUNER) project and was carried out by retrieval experts from the atmospheric remote sensing community (including active participation from eight different instrument science teams) who have come together to tackle the (arguably daunting) goal of establishing a consensus approach for reporting errors, hopefully enabling more robust scientific studies using the retrieved geophysical data products.
This review paper, the first 'foundational' paper from the TUNER team, is mainly addressed to the providers of remotely sensed data. Major parts of this work have been carried out from the perspective of passive satellite-borne limb sounding and occultation observations, which accounts for a bias of the examples presented towards these techniques. The underlying theoretical considerations, however, should be applicable to a wider context. A paper addressed to the data users, guiding them through the correct use of the uncertainty information, is currently being written (Livesey et al., in preparation).

Most concepts presented in this paper rely on the assumption that providing the user with the result of the retrieval, a measure of estimated error or uncertainty along with correlation information, and sensitivity to possible a priori information used is sufficient for most scientific uses. In other words, there is no need for more detailed discussion of the expected distribution

of the retrieved values around a true value (or around the expectation value of the retrievals) to be provided. That said, we recognize that they might be useful for some specialized quantitative applications.

The well-informed reader will already be acquainted with most of the material in this paper, although those less familiar with retrieval algorithms may find it a useful introduction. Firstly we list conditions of adequacy of the reporting of error and uncertainty (*desiderata*), which summarize the information that should be provided to the data user (Section 2). Next, before diving headlong into the technical details, Section 3 attempts to offer some necessary clarification of various terminological issues. In Section 4 we lay down the formal background. In particular, we discuss the retrieval equation and try to provide unambiguous interpretations of all involved terms, enabling the informed reader to map their own notation and terminology to that discussed herein. In our discussions of retrieval theory we will not reinvent the wheel but build heavily on the framework laid out by Rodgers (1976, 1990, 2000). Importantly, however, our discussion of the data characterization is done in the context of retrieval schemes beyond those endorsed therein, including many in every day use among remote sounding teams. Section 5 discusses how the theory translates into real world problems, centering on how the full retrieval problem is decomposed into sub-problems. Following this, we turn towards error estimation and uncertainty assessment. We then systemize and discuss the various sources of retrieval error (Section 6) and, if applicable, their dependence on the retrieval scheme chosen. We identify data characterization methods currently in use and relate them to the theoretical concepts presented. Recommendations on unified error reporting for space-borne atmospheric temperature and composition measurements are given in Section 7. In these recommendations we refrain from stipulating conventions and confine ourselves to recommendations that can directly be inferred from the conditions of adequacy. Finally, we identify unsolved problems and applications which might not be fully covered by our framework (Section 8).

## 2   Conditions of adequacy for diagnostic metadata

With the ultimate goal of presenting a list of recommendations to the community of data providers, we first discuss a list of desired properties of diagnostic metadata from the point-of-view of a data user. By diagnostic metadata we mean error or uncertainty estimates and all information on the content of a priori data, spatial resolution, and the like. The list of possible metadata to characterize retrievals of atmospheric state variables is huge, but some of them are more useful than others. Here we define conditions of adequacy (CoA) for error and uncertainty reporting. These conditions will be used as criteria which metadata are indeed essential and should thus find their way into the recommendations.

**CoA 1.**  The error estimates should be intercomparable among different instruments, retrieval schemes, and/or error estimation schemes.

**CoA 2.**  The estimated errors should be independent of the vertical grid in the sense that correct application of the established error propagation laws to the transformation of the data from one grid to another yields the same error estimates as the direct evaluation for a retrieval on the new grid would do. For characterization data not fulfilling this criterion, means should be provided for transformation from one grid to another.

**CoA 3.** The error budget and characterization data shall contain all necessary information needed by the data user to use the data in a proper way. The error budget shall be useable without detailed technical knowledge of the instrument or retrieval technique. This enables the data user to correctly apply error propagation laws and calculate uncertainty in higher level data products.

**CoA 4.** The error analysis shall be traceable in a sense that all relevant underlying assumptions are documented.

**CoA 5.** In principle the error estimates should be empirically validatable. Empirical validation is achieved via comparison between independent measurements because the true values of the atmospheric state are unknowable. We consider error estimates as empirically adequate if differences between independent measurements can be fully explained by the proper combination of their error bars, natural variability in the case of less than perfect collocations, different resolution in time and space, and different amounts of possibly different prior information.

**CoA 6.** The data volumes associated with this reporting should be reasonable. This is particularly important because involved matrices (e.g., covariances and averaging kernels) exceed the data volume of the data themselves by orders of magnitude.

These conditions of adequacy comply in part with the principles issued by the Quality Assurance Framework for Earth Observation (QA4EO) task team (2010) . That document requests traceability and fitness for purpose. We endorse traceability of the uncertainty estimates but we consider it unrealistic to assign quality indicators for 'fitness for purpose' for all conceivable applications. With generic error characterization data available, the fitness for a specific purpose can easily evaluated.

## 3   Terminological issues

Unification of error reporting is only achievable if at least a minimum agreement on terminology and the underlying concepts is achieved. Most of the terms used are largely self-explanatory and are introduced in the following sections. There are, however, two troublesome terminological issues. One consists of the dispute as to whether 'estimated error' and 'uncertainty' relate to the same concept and, if not, which concept is appropriate. The other is related to the exact connotation of these terms with respect to the underlying methodology. In the following, both issues will be briefly discussed.

### 3.1   Error versus uncertainty

A particularly troublesome terminological issue is the use of the term 'error' and the concept behind it. Given that the Joint Committee for Guides in Metrology (JCGM) and the Bureau International des Poids et Mesures (BIPM) aim to replace the concept of error analysis by the concept of uncertainty analysis (Guide to the expression of uncertainty in measurement (GUM), 2008a), some conceptual and terminological remarks are in order. While on the face of it, this is quibbling about words, it is actually claimed in these documents that there are conceptual differences between error analysis and uncertainty estimation. A deeper discussion of this issue is beyond the scope of this paper. The interested reader is referred to, e.g., von Clarmann et al. (paper in preparation), Bich (2012), Grégis (2015), Elster et al. (2013), and European Centre for Mathematics and Statistics in Metrology (2019). Von Clarmann challenges the principal difference between the error concept and the uncertainty concept;

Bich (2012), although a Working Group leader of the JCGM, claims inconsistencies between the GUM document and its supplements; Grégis (2015) challenges the position that one can dispense with the notion of 'true value' in metrology as suggested in GUM. Elster et al. (2013) and European Centre for Mathematics and Statistics in Metrology (2019) critically discuss the applicability of the GUM concept to inverse problems. Conversely, QA4EO task team (2010), Merchant et al. (2017), and Povey and Grainger (2015), e.g., largely endorse the GUM-based uncertainty concept. The latter authors, however, state that the GUM conventions "[...] apply equally to satellite remote sensing data but represent an impractical ideal that does not help an analyst fully represent their understanding of the uncertainty in their data. This is due to the simplistic treatment of systematic errors." Those of the QA4EO documents listed on https://qa4eo.org/documentation/ (last visited on 2 April 2020) which discuss issues spanning multiple instrument types are targeted at data management issues and workflows rather than scientific and technical details. Among these QA4EO documents, only Fox (2010) deals with error estimation, but in a very general way without covering the issues specific to remote measurements of atmospheric composition and temperature. The application of GUM to remote sensing of the atmosphere are hampered by the fact that GUM does not explicitly take indirect measurements into account, that GUM assumes a well defined measurand while the atmosphere is characterized by statistical variables which do not relate to a canonical ensemble, and that the problem of a priori information is not considered. For our purposes it is sufficient to say that the claim of the conceptual difference is still under debate, and that we have not fully adopted the terminology stipulated by the JCGM. Instead, we invoke the statement in Joint Committee for Guides in Metrology (JCGM) (2008a) that the error concept and the uncertainty concept are 'not inconsistent'; we understand this in a sense that the underlying methodology and mathematical tools are the same, and that the differences are restricted to the interpretation of the terms under dispute.

The GUM-stipulated framework, however, does present a dilemma when seeking to unify terminology in the TUNER arena. On the one hand, we are not in favour of brushing away the common interpretation whereby the term 'estimated error' is used for a statistical quantity that reflects the difference between the true value and the value inferred from the measurement. It remains to be seen whether the new terminology stipulated by the JCGM (JCGM, 2008a and 2008b) will be widely accepted. Accordingly, given the significant heritage within the atmospheric remote sensing community, renaming long-established concepts would not promote our goal of 'unification'. In recent scientific literature, terms like 'estimated measurement error', 'error analysis', 'error covariance matrix' or 'standard error of the mean' are still widely in use, and replacement terms like 'standard uncertainty of the mean' etc., are rarely invoked. On the other hand, we recognize that explicitly breaking with the official stipulations of the JCGM does not advance the overall goal of 'unification' either.

For the purposes of the following discussion we define 'error' to be the difference between an unknown truth and a value inferred from measurements. 'Uncertainty' describes the distribution of an error. This can be summarized with metrics such as the total squared error, which can be decomposed into systematic and random components that are reflected by bias and variance. We will often use the word 'error' as a part of composite terms, (e.g., 'parameter error', 'noise error', 'retrieval error', 'estimated error', etc.). When we use a composite containing the term 'error', this does not imply that the uncertainty interpretation is excluded, and conversely, when we use a composite term containing the term 'uncertainty', this does not imply that the error interpretation is excluded. The use of the term 'error' as a generic term in the sense of 'measurement noise causes

an error in the inferred quantity' is probably uncontroversial and can be accepted both by adherents of the error concept and adherents of the uncertainty concept.

We think that no particular terminology is *per se* better than another one, as long as it is clearly defined. Instead of further fueling the terminological conflict, we try to concentrate on the content and to lay down an error reporting framework custom-tailored to remote measurements of atmospheric temperature and constituents that is more detailed and specific than most of the previous literature.

## 3.2   Ex ante versus ex post error estimates

Regardless of whether one prefers to call the estimated retrieval error 'uncertainty' or the uncertainty of the measurement 'estimated error', there are still two different ways to evaluate this quantity. One relies on generalized Gaussian error propagation or, particularly in grossly nonlinear problems, on sensitivity studies, either as case studies or in a Monte Carlo sense. Uncertainties of input quantities are propagated through the data analysis system to yield the uncertainties of the target quantities. The other way relies on a statistical analysis of the results, e.g., by comparison to other observations. Many different terms are commonly used to distinguish between these different approaches. In Joint Committee for Guides in Metrology (JCGM) (2008a), the first fall into their 'category B', while the second are 'category A'. Von Clarmann (2006) distinguishes between *ex ante* and *ex post* error estimates, reflecting the fact that error propagation can be calculated even before the measurement has been made, while the statistical analysis of the measurements requires the availability of actual measurements. Along the same line of thought, one could also talk about error prediction *versus* evidence of errors. Since error estimation is deterministic with respect to the estimated variances (but certainly not with respect to the actual realizations of the measurement error), and since statistical analysis of any evidence follows the laws of inductive logic (Carnap and Stegmüller, 1959), one could also distinguish between deductive and inductive error estimation. Others prefer to use the terms 'bottom up' and 'top down' for this dichotomy. This study focuses chiefly on *ex ante* error estimation. To validate these estimates, *ex post* error estimation is relevant, as expounded, e.g., by Keppens et al. (2019).

## 4   Retrieval theory and notation

Measurements – also most so-called direct measurements – invoke inverse methods. The only exception is a direct comparison where the measurand is directly accessible via human sensation, like length measurement by comparison with a yardstick or determination of colour by comparison with a colour table. The inverse nature of most measurements is due to the fact that the measurand $x$ is the cause and the measured signal $y$ is the effect. These are connected via a natural regularity which is formalized via a function

$$\boldsymbol{f} : \mathbb{R}^n \to \mathbb{R}^m : \boldsymbol{x} \mapsto \boldsymbol{y} = \boldsymbol{f}(\boldsymbol{x}), \tag{1}$$

which maps the discretized measurand onto the respective observable signal, and where $m$ and $n$ designate the number of measured data points and the number of state values, respectively.

In the macroscopic world, exempt from quantum processes, the measured effect is thus, for given conditions, a deterministic unambiguous function of the measurand. While microscopic processes can admittedly be indeterministic, their statistical treatment for ensembles of sufficient size leads to deterministic laws. Irreducibly non-deterministic components contribute to the measurement noise. In contrast, the conclusion from the measured signal $y$ to the measurand $x$ is not always unambiguous because in many cases the inverse function

$$g : \mathbb{R}^m \to \mathbb{R}^n : y \mapsto x = f^{-1}(y) \tag{2}$$

can only be approximated due to the over- or underdetermined or otherwise ill-posed nature of the problem and the large rank of the matrix to be inverted.

In some cases, the inverse process can be quite trivial, e.g., in the case of a temperature measurement with a mercury thermometer. The causal process is the thermal expansion of mercury, and the inverse conclusion goes from the volume of the mercury to the ambient temperature. The scale of the mercury thermometer is simply a pretabulated solution of the inverse process for various temperatures. In other applications, such as remote sensing of the atmosphere from space, the inverse process is slightly more complicated because an explicit $f^{-1}(y)$ does not usually exist. Related workarounds to solve this problem are discussed below.

Remote sensing of the atmospheric state from space relies in one form or another on the radiative transfer equation (Chandrasekhar, 1960). This equation is deterministic in a sense that its formulation $f$ simulates the measured signal via causal processes. The deterministic characteristic of $f$ in the macroscopic world is achieved via a statistical treatment of the underlying microscopic processes. While its forward solution allows the calculation of the radiance received by the instrument, its inverse solution allows for the determination of the state of the atmosphere from a known radiance signal.

Roughly following the notation of Rodgers (2000), we define $F$ to be the radiative transfer model which approximates $f$. $F$ is a vector-valued non-linear function and deviates from $f$ in that it is discrete in space and frequency, involves numerical approximations and may not include the full physics of radiative transfer. $x \in \mathbb{R}^n$ is the vector representing the atmospheric state, and $y \in \mathbb{R}^m$ the vector containing the measured radiance signal. The elements of $x$ contain both the 'target variables' and 'joint-fit' variables. Target variables are those variables we are actually interested in. Conversely, the joint-fit variables are variables needed by $F$ that, while not the focus of our interest, have to be sought in the inversion because they may not be accurately known and their uncertainties would otherwise make an unacceptably large contribution to the total error budget.

Typically $m \neq n$, i.e., the dimension of $x$ does not equal the dimension of $y$. For $m > n$, Gauss (1809)[1] suggested an approximate inversion obtained by minimizing the sum of squares of the residual $F(x) - y$. If we assume, for now, that $F$ is linear and that Gaussian distributions of are adequate to characterize the measurement (see Section 5.5 for related problems) the unconstrained [2] solution of the inverse problem is

$$\hat{x}_{\mathrm{ML}} = x_0 + \left[ \mathbf{K}^T \mathbf{S}_{y,\mathrm{total}}^{-1} \mathbf{K} \right]^{-1} \mathbf{K}^\mathrm{T} \mathbf{S}_{y,\mathrm{total}}^{-1} \left[ y - F(x_0) \right]. \tag{3}$$

---

[1]The first publication of a least squares method was actually by Legendre (1805), but Gauss is said to have had this idea about ten years before. Obviously unaware of Legendre's work, also Robert Adrain (1808) proposed the least squares method as the most advantageous solution in this context. See Merriman (1877), Sprott (1978), or Stahl (2006) for a deeper discussion of the priority regarding this method.

[2]See below for a deeper discussion of this term.

$\mathbf{K}$ is the Jacobian matrix with the elements $K_{ij} = \frac{\partial y_i}{\partial x_j}$, $\boldsymbol{x}_0$ represents an initial guess of the atmospheric state, and $\mathbf{S}_{\mathrm{y,total}}$ is the covariance matrix characterizing the total measurement error. Here the ˆ symbol indicates that, due to measurement noise mentioned above, and other uncertainties and ambiguities which will be discussed below, the result of the inversion is only an estimate of the measurand $\boldsymbol{x}$. In most real-world applications, only measurement noise is considered here, while other measurement uncertainties like calibration errors are neglected at this stage. Since the solution provided by Eq. 3 does not consider any prior information, it is a "maximum likelihood" solution in the sense of Fisher (1922, 1925)[3].

One major difference between our notation and Rodgers' notation refers to the error covariance matrices $\mathbf{S}$. We use two subscripts. The first indicates if the uncertainties refer to the retrieved quantities $\boldsymbol{x}$ or to the ingoing quantities. The second subscript specifies the source of the uncertainty. For example, $\mathbf{S}_{\mathrm{y,noise}}$ is noise in the measurement data, while $\mathbf{S}_{\mathrm{x,noise}}$ is the measurement noise mapped into the retrieved atmospheric state. In other words, $\mathbf{S}_{\mathrm{x,noise}}$ is the error component in $\boldsymbol{x}$ due to the error source $\mathbf{S}_{\mathrm{y,noise}}$. In some cases, e.g., if any ambiguity can be excluded or if the sources of the error are not known, the second subscript can be missing.

By explicitly assuming equally distributed, i.e., uniform prior, state values Gauss (1809, p. 211) gave this solution a probabilistic interpretation without clashing with the Bayes (1763) theorem. In a linear context and for measurement errors following a normal[4] distribution around the true value, the Gaussian least squares solution corresponds formally — but certainly not in terms of its interpretation — to a maximum likelihood solution in the terminology of Fisher (1922, 1925) (thus the index ML in $\hat{\boldsymbol{x}}_{\mathrm{ML}}$). An interesting overview on the history of maximum likelihood estimates is given by Hald (1999), while the justification of this method is critically discussed in Aldrich (1997). For instructive discussions of the relevance of the Bayes theorem in inductive statistics, see, e.g., Bar-Hillel (1980) and Thompson and Shuman (1987). The original Gaussian least squares method was valid for independent measurement errors only. The introduction of the correlation coefficient by Galton (1888) and Pearson (1896) paved the way towards a wider range of applications. The matrix formulation as used today, where correlated measurement errors are represented in the measurement error covariance matrix $\mathbf{S}_{\mathrm{y}}$, owes much to Yule (1907), Fisher (1925), and Aitken (1935). A reconstruction of the historical development of this technique was performed by Aldrich (1998).

If the inverse problem is underdetermined ($m < n$) or ill-posed in a sense that the $\left[ \mathbf{K}^T \mathbf{S}_{\mathrm{y,total}}^{-1} \mathbf{K} \right]$ matrix is singular or has a high condition number, then a constraint has to be used. Even in formally well-conditioned problems but large measurement noise, the use of a constraint can be helpful. With a prior assumption on the atmospheric state $\boldsymbol{x}_a$ and a regularization matrix $\mathbf{R}$ we can modify Eq. (3) in a way that the matrix inversion can be accomplished. This so-called regularized solution is (von Clarmann et al. 2003, building upon Rodgers 2000; Phillips 1962; Tikhonov 1963; Twomey 1963; Steck and von Clarmann 2001)

$$\hat{\boldsymbol{x}}_{\mathrm{reg}} = \boldsymbol{x}_a + \left[ \mathbf{K}^T \mathbf{S}_{\mathrm{y,total}}^{-1} \mathbf{K} + \mathbf{R} \right]^{-1} \mathbf{K}^T \mathbf{S}_{\mathrm{y,total}}^{-1} \left[ \boldsymbol{y} - \boldsymbol{F} \left( \boldsymbol{x}_a \right) \right]. \tag{4}$$

---

[3]See below for a deeper discussion of this term.

[4]Normal distribution and Gaussian distribution are the same. The term 'normal distribution' was probably coined by Karl E. Pearson in 1893. While this term evades the question of priority in its discovery, it "has the disadvantage of leading people to believe that all other distributions of frequency are in one sense or another *abnormal*", as Pearson (1920) self-critically states.

Many choices of the regularization matrix $\mathbf{R}$ are possible. With the $(n-1) \times n$ first order differences matrix $\mathbf{L}_1$ and $\gamma$ a scaling parameter to control the strength of the regularization and balancing the units, the choice of

$$\mathbf{R} = \gamma \mathbf{L}_1 \mathbf{L}_1^T, \tag{5}$$

renders fields of profiles of atmospheric state variables that are smoothed in the sense of reduced altitude-to-altitude differences of the $\hat{x}_{\text{reg}} - x_{\text{a}}$ profile, thus avoiding unphysical oscillations that typically result from instabilities associated with ill-posed inverse problems.

If we represent the best known a priori statistics about the targeted atmospheric state as $x_a$, its covariance matrix as $\mathbf{S}_a$, the inverse of this matrix as $\mathbf{R}$, and continue to assume Gaussian error distributions, then we get a Bayesian solution that is usually referred to as 'optimal estimate' (Rodgers, 1976) or 'maximum a posteriori (MAP) solution' (Rodgers, 2000) and is fully compatible with the Bayes (1763) theorem and information theory by Shannon (1948), and thus gives the solution a probabilistic interpretation in the sense of the *maximum a posteriori* probability:

$$\hat{x}_{\text{MAP}} = x_{\text{a}} + \left[ \mathbf{K}^T \mathbf{S}_{\text{y,total}}^{-1} \mathbf{K} + \mathbf{S_a}^{-1} \right]^{-1} \mathbf{K}^T \mathbf{S}_{\text{y,total}}^{-1} \left[ y - F\left( x_{\text{a}} \right) \right]. \tag{6}$$

The formalism of Eq. 6 can also be used without committing oneself to a probabilistic interpretation of $\mathbf{S}_a$. For example, $\mathbf{S}_a$ can be rescaled to give less weight to the priori information.

This equation, however, has a Bayesian interpretation only if the variability of the atmospheric state is fairly well covered by a Gaussian probability density function. To characterize the variability of highly variable trace gases, a log-normal probability density function can be more adequate. It avoids, e.g., that non-zero a priori probability densities are assigned to negative mixing ratios. Technically, this is achieved by using Eq. (6) but re-interpreting $x$ as the logarithm of the concentrations and $\mathbf{S}_a$ as the covariance matrix of these logarithms. This is, for instance, important for tropospheric water vapour (e.g., Hase et al., 2004 or Schneider et al., 2006). However, there is a price to be paid, in that this then casts the measurement error in terms of a log-normal distribution also. The positive bias caused by the retrieval of logarithms of concentrations in the case of measurement noise oscillating arount zero signal has been investigated by Funke and von Clarmann (2012).

For brevity, we define the gain matrix (Rodgers, 2000)

$$\mathbf{G} = \frac{\partial \hat{x}}{\partial y} = \left( \mathbf{K}^T \mathbf{S}_{\text{y,total}}^{-1} \mathbf{K} + \mathbf{R} \right)^{-1} \mathbf{K}^T \mathbf{S}_{\text{y,total}}^{-1}, \tag{7}$$

which will play an essential role in error estimation. The remainder of this paper broadly identifies all relevant sources of uncertainties including measurement noise, approximations, idealizations, and assumptions.

Tables 1 and 2 summarize the retrieval schemes used by a number of satellite data processors.

## 5 Retrieval in the real world

Application of Eqs. (3–6) usually involves many approximations and idealizations including discretization, decomposition of the argument of the radiative transfer function into variables and parameters, spatial decomposition, and non-linearity issues,

just to name a few. Since all these approximations give rise to retrieval errors, a full understanding of them is of utmost importance when quantifying the error budget of a measurement.

## 5.1 Discretization

At least on macroscopic scales, atmospheric state variables are construed as continuously varying in space and time. In the retrieval equations they are, however, represented by vectors with a finite number of elements. A frequent discretization is the representation of the atmospheric state at a limited number of gridpoints. The profile shape between these gridpoints depends on the interpolation scheme chosen. Often profiles are conceived as piecewise linear. The finite grid can be conceived as a surrogate regularization because it places a hard constraint on the shape of the profile between two gridpoints. If the discretization is too fine, a stronger regularization is needed to fight ill-posedness of the inversion, while a too coarse discretization can cause errors in the radiative transfer modelling and limits the spatial resolution of the solution. Also the abrupt gradient changes tend to be more and more unphysical the coarser the grid is. In a maximum likelihood retrieval, the grid-width is identical to the theoretical spatial resolution of the retrieval. However, if the gridwith is chosen too fine, the useful resolution of the maximum likelihood retrieval will be much worse because the fine structures of the profile will be masked by the noise.

Alternatively, vertical profiles can be conceived as a set of layers, each represented by layer-averages of atmospheric state values and/or partial column amounts of species. In this case no assumptions on the profile shape between the layer boundaries are obvious but they would be implicit because the details of the averaging may depend on them.

In this context we note that the atmospheric state does not necessarily need to be represented as vertical profiles where each element of $x$ is a state variable at a certain altitude or location, or represents an atmospheric layer. Alternative representations include, e.g., principal components/empirical orthogonal functions (see, e.g., Boukabara et al. 2011; Munchak et al. 2016; Duncan and Kummerow 2016). These can be inferred from an ensemble of spatially highly resolved prior measurements. The unknowns in the retrieval are the weights of the principal components. Complete or partial neglect of higher principal components will regularize the retrieval. Such an approach is under consideration for the Atmospheric Limb Tracker for Investigation of the Upcoming Stratosphere (ALTIUS) mission (Fussen et al., 2016). A similar approach was tried for the Scanning Imaging Absorption Spectrometer for Atmospheric Chartography (SCIAMACHY) (Doicu et al., 2007), for the multi-channel infrared radiometer on the Geostationary Operational Environmental Satellite (GOES-13) and infrared sounder on the Indian National Satellite (INSAT-3D) by Jindal et al. (2016). The retrieval of vertical column amounts by simply scaling the initial guess profile reduces the profile retrieval to a single degree of freedom. Alternatively, the altitude axis of the profile can be stretched or compressed using the so-called 'downwelling factor' as suggested by Toon et al. (1992). These approaches are often used for analysis of measurements which do not provide direct information on the vertical distribution of the target species. Particularly in the greenhouse gas monitoring community, retrieved column amounts of target species are divided by the molecular oxygen column amount retrieved with the same instrument. Rescaling of the quotient by the 0.20946 gives the column-averaged dry-air mole fraction $XCO_2$ or $XCH_4$. The benefit of this approach is a cancellation of multiplicative systematic error components (see, e.g., Wallace and Livingston, 1990; Yang et al., 2002; Wunch et al., 2010; Reuter et al., 2011). Similar arguments hold for isotopic ratios (e.g., Piccolo et al., 2009; Schneider et al., 2016) or ratios between trace gas

profiles (e.g. García et al., 2018). For measurement techniques in the visible and UV, scattering is particularly important as it governs how long the actual lightpath is in each layer; accordingly, a certain amount of the target gas will have a stronger or weaker effect on the measured signal, depending in which layer it is. This altitude-dependence is accounted for by an airmass factor which governs the weight of each layer in the total column. The total column thus can be conceived as a weighted sum over the layers, where the weight of each layer is propotional to the sensitivity (see Section 5.4.7 for further details).

## 5.2 The measurement error covariance matrix

Typically in real-world applications, the measurement error $\mathbf{S}_y$ in Eqs. (3–6) contains only measurement noise, while other sources of measurement error are often ignored during the retrieval (see Section 6.1 for details) and typically analyzed after performing the retrieval. Since this treatment deprives any solution from its claimed optimality (Cressie, 2018), in some cases the measurement noise is artificially "inflated" to account for potential calibration uncertainties. A method to include multiple types of uncertainties in the measurement error covariance matrix is discussed in Marks and Rodgers (1993), Tarantola and Valette (1982), Eriksson (2000), and von Clarmann et al. (2001). These authors discuss the possibility of mapping all relevant error contributions into the measurement space and include them in the $\mathbf{S}_y$ matrix[5]. Rodgers (2000, Section 4.1.2) views this problem from a different perspective but the suggested solution is mathematically equivalent to the approach suggested above.

## 5.3 Variables and parameters

While the measurement typically depends on a large number of geophysical state variables, only a few of them are actually dealt with as unknowns. The other variables are assumed to be known and are dealt with as constant parameters. For example, in an ozone profile retrieval the atmospheric temperature profile may be assumed to be known and thus not be included in the retrieval vector $\boldsymbol{x}$. With this, the forward problem can be formalized as

$$\boldsymbol{y} = \boldsymbol{F}(\boldsymbol{x}; \boldsymbol{b}), \tag{8}$$

where $\boldsymbol{b}$ is the vector of parameters, which are separated in the argument of function $f$ by the semicolon. The respective inverse solution reads in its general form

$$\hat{\boldsymbol{x}}_{\text{reg}} = \boldsymbol{x}_a + \left( \mathbf{K}^T \mathbf{S}_{y,\text{total}}^{-1} \mathbf{K} + \mathbf{R} \right)^{-1} \mathbf{K}^T \mathbf{S}_{y,\text{total}}^{-1} \left[ \boldsymbol{y} - \boldsymbol{F}\left( \boldsymbol{x}_a; \boldsymbol{b} \right) \right]. \tag{9}$$

The uncertainties of parameters $\boldsymbol{b}$ affect the estimate $\hat{\boldsymbol{x}}$ and thus a parameter error term has to be included in the error budget.

## 5.4 Decomposition of the inverse problem

Practical reasons typically force one to decompose the inverse problem, e.g., to reduce the size of the problem in order to achieve numerical efficiency. Often a part of the measurement is virtually insensitive to some of the atmospheric state variables.

---

[5]This issue seems to be of particular importance when observation error covariance matrices are built in contexts where a data assimilation scheme uses radiance measurements instead of retrieved state variables, as suggested by Andersson et al. (1994).

The general idea of decomposition is to isolate subsets of the entire set of measurements that are mainly sensitive to only a subset of the unknown variables. This decomposition can be made according to spectral or geometrical criteria (see below).

Decomposition of the inverse problem can be done either in an "optimal" or in a "non-optimal" way. The optimal decomposition solves the inverse problem sequentially, where at each step the retrieval is made for the full $x$-vector but based only on a subset of the measurements, whereby each measurement is used only once during the entire process (see, e.g., Rodgers, 2000, Ch. 5.8.1.3; his requirement of a diagonal measurement covariance matrix can be replaced by the weaker requirement of a block-diagonal covariance matrix if the algebra is adjusted accordingly). Initially, the retrieval, which typically is patently under-determined because of the temporarily ignored measurements, is constrained by an initial $\mathbf{S}_a$ matrix. For each subsequent step, the $\mathbf{S}_a$ matrix is replaced by the so-called 'retrieval covariance matrix'

$$\mathbf{S}_x = \left( \mathbf{K}^T \mathbf{S}_{y,\text{total}}^{-1} \mathbf{K} + \mathbf{S}_a^{-1} \right)^{-1} \tag{10}$$

of the preceding step. Within linear theory, the solution of such sequential methods is equivalent to the direct solution of the full inverse problem.

More frequently used is non-optimal decomposition. Here the relevance of some components of the state vector for the measurements is temporarily ignored, and the retrieval solves the inverse problem only for a part of the state values, using only a subset of the measurements. This approach lends itself to problems where it is adequate to assume that the Jacobian matrix $\mathbf{K}$ has an almost block diagonal structure, that is, that there are state variables which have no significant influence on some of the measurements under analysis and vice versa. In the following we discuss spectral and spatial decomposition.

### 5.4.1 Spectral decomposition

Not all spectral gridpoints or channels of a spectrometer or a multi-channel radiometer are equally sensitive to all unknown variables. For example, the subset of the measurements used to retrieve the ozone concentration may be insensitive to the concentration of water vapour (Flittner et al., 2000). In such cases, the abundances of various species can be retrieved in sequence, using dedicated "microwindows" in infrared spectroscopy (see, e.g. von Clarmann and Echle, 1998; Echle et al., 2000; Dudhia et al., 2002), different spectral regions in microwave radiometry (Livesey et al., 2003, 2006) or measurements in the ultraviolet and visible (UV-VIS) spectral range (e.g., Bovensmann and M. Gottwald, 2011). In these cases, a subset of spectral points is selected for analysis. Those unknowns which have sizeable impact on the signal at these spectral points are retrieved. When in later steps other spectral points are analyzed, the results of the first steps can be used and either be treated as known parameters, or as a priori information in an optimal sequential scheme. Uncertainties entailed by this procedure are associated with the following considerations: (1) In the first step some of the disregarded variables may still introduce some error; (2) retrieval errors of all kinds resulting from a prior step of the sequential scheme propagate onto the results of later steps; and (3) inconsistencies in spectroscopic parameters between different spectral points can cause a spurious residual signal when, e.g., the concentration of a gas retrieved in one part of the spectrum is used as a known parameter in the analysis of another part of the spectrum.

Spectral decomposition is also often used for the retrieval of a single species. For example, Kramarova et al. (2018) retrieve ozone sequentially in different spectral bands. An alternative to spectral decomposition is the simultaneous analysis of the full spectrum (e.g., Serio et al., 2016). In cases when spectroscopic data are consistent over the entire spectral range it will best exploit the observational information.

### 5.4.2 Geometric decomposition

In the case of nadir sounding, lines of sight referring to different ground-pixels cross different parts of the atmosphere and can thus be analyzed independently without sizeable loss of information. In optical limb sounding of the Earth's atmosphere, first suggested around the same time by Gille (1968), Blamont and Luton (1972), Hays et al. (1973) and Donehue et al. (1974) for different scientific contexts, the situation is more complicated because the retrieval of a state value at a given altitude depends on the knowledge of the same state value at other altitudes passed by the line of sight.

If the same air parcel is seen under multiple geometries, the measurements have a tomographic nature. Since the simultaneous retrieval of all these intertwined measurements easily exceeds available computational resources, often only a subset of the measurement geometries are analyzed in one step.

More specifically, the algorithm can be constructed such that only a subset of the measurements are needed to retrieve the atmospheric state corresponding to a given subset of the state vector elements that affect signals along the raypath of the considered measurement. The two most prominent examples are single profile retrievals and onion peeling. Typical approaches to decompose the entity of measurements geometrically are listed in Table 1.

In some cases, the geometric profile reconstruction is decoupled from the spectral inversion. In order to gain numerical efficiency, the inversion can be performed in sequential steps. Such an approach is realized for GOMOS two-step inversion, which decompose the retrievals into the spectral inversion followed by the vertical inversion using the concept of effective cross-sections (Kyrölä et al., 1993).

### 5.4.3 Optimal decomposition techniques

Optimal decomposition techniques formally retrieve all relevant variables $x$ in each step but measurement information $y$ of only a subset of the measurement geometries is used. Since, in a maximum likelihood setting, such a retrieval would be hopelessly underdetermined, sequential estimation as described above lends itself to this class of problem. Every state variable can be updated as soon as new information becomes available. In contrast, non-optimal techniques will not update any quantity once retrieved.

### 5.4.4 Single profile retrieval vs. 2D/3D-retrievals

The vast majority of limb sounding retrievals assume local spherical homogeneity of the atmosphere, i.e. considering only vertical variations in the atmospheric state around the line of sight, and neglecting horizontal variability (e.g. Gille, 1968; McKee et al., 1969a, b; House and Ohring, 1969; Carlotti, 1988). Russell III and Drayson (1972) explicitly state the assumption,

and only a small number of retrieval schemes relinquish it. In solar occultation observations, where the measurement geometry is determined by the position of the sun and the instrument and where at most one sunset and one sunrise can be observed per orbit, there is not much choice; tomographic multi-limb-scan retrievals are out of reach and the single profile retrieval is the way to go.

For limb measurements, von Clarmann (1993) suggested a non-optimal decomposition similar to "onion peeling" (see below) but in the horizontal domain. This approach, however, was never put into action. Carlotti et al. (2001) proposed to solve the inverse problem for a full satellite orbit instead of for single limb-scans. This tomographic method was published under the name 'geofit'. Steck et al. (2005) tested an implementation of sequential estimation in the horizontal domain, while the vertical domain was treated in one leap. Livesey and Read (2000); Livesey et al. (2008); Christensen et al. (2015) employ a tomographic approaches, whereby a 2-dimensional along-track curtain of profiles is simultaneously retrieved from multiple sets of limb scans. A similar approach is used for SCIAMACHY retrievals of metals (Scharringhausen et al., 2008; Langowski et al., 2014) and NO (Bender et al., 2013, 2017) and for OMPS-LP ozone (Zawada et al., 2018).

Dudhia and Livesey (1996) and von Clarmann et al. (2009) use prior information on the horizontal variation of state variables in a single limb-scan retrieval. The latter scheme lends itself particularly to reprocessing of data when the initial processing information on the horizontal variability is already available. This approach has been critically analyzed by Castelli et al. (2016). Tomographic approaches and the effect of horizontal gradients were investigated for SCIAMACHY limb measurements by Pukite et al. (2008) and Pukite et al. (2010). A series of OSIRIS orbits allowed the tomographic analysis of polar mesospheric clouds (Hultgren et al., 2013). This application was preceded by theoretical studies on OSIRIS infrared channels tailored for tomography.

Most other limb sounding retrieval schemes use the spherical homogeneity approximation, although this approach can be challenged for limb sounders. For example, Kiefer et al. (2010) provided evidence of biases in trace gas retrievals from MIPAS limb emission spectra due to horizontal temperature gradients. Thus, neglect of the horizontal variation of the atmospheric state needs either to be corrected or to be considered in the error budget.

In the case of nadir sounding at mid-infrared and longer wavelengths, single profile or column density retrievals seem to be the natural thing to do, since a raypath associated with one geolocation intersects each altitude level only once. However, in the UV-VIS spectral range, where backscattered solar light is the source of the radiation, multiple scattering along with strong inhomogeneities in the surface reflection or cloud coverage might cause some interplay between the neighbouring pixels.

One specific geometrical decomposition applied to nadir observations is the retrieval of tropospheric column densities. Since the raypath also travels through the stratosphere, knowledge on the stratospheric column is needed to model the measured signal correctly. Boersma et al. (2004) summarizes three techniques to obtain this information. Leue et al. (2001) uses measurements from cloudy pixels to infer the stratospheric amount; Richter and Burrows (2002) and Martin et al. (2002) use data from remote Pacific regions where the total column of their target gas $NO_2$ is approximately identical to the stratospheric column, and Richter et al. (2002) use stratospheric column information from a chemistry transport model. Alternatively, collocated limb measurements can be used to constrain the stratospheric column using a limb-nadir matching technique (Ziemke et al., 2006; Hilboll et al., 2013; Ebojie et al., 2014).

### 5.4.5 Onion peeling

In the "onion peeling" approach, (Gille, 1968; McKee et al., 1969a, b; House and Ohring, 1969; Russell III and Drayson, 1972; Goldman and Saunders, 1979) the collection of limb measurements in a vertical scan is decomposed into a sequence of retrievals, each dealing with one tangent altitude, starting at the top and working down. This method builds upon the fact that the bulk of the information obtained along the horizontal line of sight originates from the vicinity of the tangent point, with limited information from above and essentially none from below. In the first step, the measurement associated with the uppermost tangent altitude is analyzed and the profile above is scaled. Then the second tangent altitude from the top is used and the profile between this tangent altitude and the tangent altitude above is scaled; this is repeated until the lowermost tangent altitude is reached. Often the discretization of the atmospheric state corresponds to the tangent altitude pattern, i.e., there is one profile point per tangent altitude, and the profile shape between the points is determined by interpolation. Gaussian elimination is already provided by the measurement geometry, and the Jacobian $\mathbf{K}$ has a quasi-triangular structure[6]. This approach, however, is prone to instabilities because $n$ layers go along with $n + 1$ neighbouring levels.

(von Clarmann et al., 1991). The alternative would be that layer values are retrieved instead of level values.

In the early era of limb sounding and solar occultation measurements, onion peeling was the work-horse data analysis algorithm and was used, among others, in the following missions: LIMS (Bailey and Gille, 1986), ATMOS (Norton and Rinsland, 1991), HALOE (Russell III et al., 1993), CRISTA (Offermann et al., 1999). More recently, onion-peeling related algorithms have been used, e.g., for, TIMED-SABER (Russell III et al., 1994), AIM-SOFIE (Gordley et al., 2009), and SCIAMACHY (Noël et al., 2018). When more computer power along with quasi-analytical algorithms to calculate larger Jacobians became available, onion peeling was often superseded by global-fit-like algorithms (Carlotti, 1988) which solve the inverse problem for the entire limb sequence in one leap.

Approaches related to onion peeling are the Mill-Drayson method (1978) and the 'interleave method' (Thompson and Gordley, 2009). The Mill-Drayson method starts with the lowermost tangent altitudes and scales the entire profile of the atmospheric state variables above to minimize the residual between measurement and modeled signal. Next, the second tangent altitude from bottom is used to scale the related upper segment of the profile above the related tangent altitude. Several iterations over the limb scan are made. The goal is to avoid the typical onion-peeling error propagation which tends to trigger oscillations in the profiles. This method became somewhat obsolete with the advent of numerical regularization. Without knowledge of the original method by Mill and Drayson this method has been applied to the SOFIE instrument by Marshall et al. (2011).

The interleave method decomposes the limb scan in multiple disjoint subsets of measurements, e.g., such that one set contains the tangent altitudes with even numbers and the other those with the odd numbers. For each subset of measurements an independent onion peeling retrieval is performed. Finally both the resulting profiles are merged to give one profile. The goal of this method is to get rid of the onion-peeling oscillations, which is achieved by having thicker layers and thus better sensitivity – at the cost of degraded vertical resolution – in each retrieval step. The interleave method has been used, e.g., for HALOE and SABER.

---

[6]We are saying 'quasi-traingular' here because, due to over-determination at each tangent altitude, $\mathbf{K}$ can have more rows than columns.

As will be seen below, rigorous error propagation for onion peeling retrievals and its variants is tedious and thus rarely performed. Instead, Monte-Carlo-type sensitivity studies can be performed on the basis of simulated measurements superimposed with artificial noise, which are analyzed using the onion-peeling scheme. The error estimate is then provided by the variance of the ensemble results around the reference value at each altitude.

## 5.4.6   Chahine's relaxation method

The Chahine relaxation method (Chahine, 1968, 1970) was originally suggested to retrieve vertical profiles of the temperature from measurements of the emerging specific intensity at several frequencies in the infrared spectral range. Later this method has been adapted by employing the geometrical decomposition to the retrieval of vertical distributions of atmospheric trace gases from the measurements of the scattered solar light in limb viewing geometry (e.g., Sioris et al., 2003, 2004).

Essentially, the measurement and state vectors have to be constructed in a way that for each of its components the following linear relationship can be considered as an acceptable approximation:

$$\frac{[\hat{\boldsymbol{x}}]_j}{[\boldsymbol{x}_{\mathrm{a}}]_j} = \frac{[\boldsymbol{y}]_j}{[\boldsymbol{F}(\boldsymbol{x}_{\mathrm{a}})]_j}, \tag{11}$$

where $[\ldots]_j$ denotes the $j$-th component of the corresponding vector. To obtain the solution, Eq. 11 needs to be solved for each component $[\boldsymbol{x}_j]$ of the state vector independently. In the original approach the number of measurements and the number of the retrieved values need to be the same. However, Sioris et al. (2004) suggested an extension of the method which solved a slightly underestimated problem with larger number of state vector components and a combination of the components of the measurement vector in the right hand side.

In the original approach of Chahine, the measurement vector $\boldsymbol{y}$ comprised measured radiances, $\boldsymbol{F}(\boldsymbol{x}_{\mathrm{a}})$ the related modeled radiances, the state vector $\boldsymbol{x}$ comprised Planck functions at certain pressure levels, and spectral decomposition was applied, i.e., Eq. 11 was solved for each frequency independently. In the approach of Sioris, the measurement vector contained trace gas slant columns at each line-of-sight, the state vector contained trace gas number densities at altitude levels and Eq. 11 needed to be solved for each line-of-sight independently, i.e., the geometrical decomposition was employed.

The Chahine relaxation method is a nested iteration of the type

$$[\hat{\boldsymbol{x}}_{i+1}]_j = [\boldsymbol{x}_i]_j \frac{[\boldsymbol{y}]_j}{[\boldsymbol{F}(\boldsymbol{x}_i)]_j}. \tag{12}$$

The inner loop runs over the altitude indices $j$ and is usually started at the top of the atmosphere and procedes downwards, similarly to the onion peeling method. However, in this inner loop, the information retrieved at higher levels is not directly used when Eq. 11 is solved for lower layers. Instead, the same current guess profile $\boldsymbol{x}_i$ is used to evaluate $\hat{\boldsymbol{x}}_{i+1}$ for all altitudes $j$. Only after finishing the inner iteration over the altitudes $j$, the state vector $\boldsymbol{x}_i$ is updated with $\boldsymbol{x}_{i+1}$. The outer iteration over $i$ is repeated until convergence is reached.

Similarly to the onion peeling, rigorous error propagation for the Chahine relaxation method is challenging and the same approach as suggested for the onion peeling method can be used instead.

### 5.4.7 Two-step DOAS methods

The characteristic feature of Differential Optical Absorption Spectroscopy (DOAS) is that the information on the target quantity $x$ is not obtained from the total measured signal but from its component varying rapidly with frequency while the smoothly varying component is approximated by a polynomial, whose coefficients are determined in the spectral fit procedure (e.g., Platt and Stutz, 2008; Eskes and Boersma, 2003). This polynomial describes the smoothly varying component in terms of optical thickness, i.e., as an additive term in the exponent in Beer's law. The fit of the differential spectrum is often realized by fitting the full measured signal whereby the coefficients of the polynomial are jointly fitted, and the retrieved total column amount has to account only for the differential signal.

When the DOAS principle is applied to limb measurements, data analysis can be performed using a formalism such as that presented in Eqs. 3 or 4 directly. In this case $x$ corresponds to the vertical absorber number density profile, and $y$ represents the measured limb radiance spectra whose smoothly varying components are parametrized as described above. Examples of this approach have been presented, e.g., by Rozanov et al. (2005).

Total column retrievals from nadir measurements can also be carried out in one step. In these approaches the total column is directly retrieved by fitting a forward-modelled differential spectrum to an observed differential spectrum. An example of these approaches is the Weighting Function DOAS (WFDOAS, e.g., Coldewey-Egbers et al., 2005). In this case the formalism such as that presented in Eq. 3 is fully applicable to column amounts in exactly the same way as is done for the vertical profile retrievals.

More often, however, the retrieval is decomposed into a two-step retrieval (e.g., Platt and Stutz, 2008; Eskes and Boersma, 2003). In a first step, slant path column densities (SCDs) are fitted to explain the rapidly varying component of the spectral signal (again the smoothly varying component is approximated by a polynomial, whose coefficients are typically jointly fitted). The resulting SCDs are the integrated absorber number densities along the effective light paths. In nadir sounding, this results in one SCD per species, while in limb sounding, one SCD per tangent altitude and target species is obtained. Referring to Equations 3 or 4, in this first step, $y$ contains the measurements and $x$ the slant SCDs. In limb sounding, this SCD profile is then inverted in the second step to yield a vertical absorber number density profile. Referring again to Equations 3 or 4, $x$ corresponds to the vertical absorber number density profile and $y$ to the SCD profile. Examples for the application of these two-step retrievals are Sioris et al. (2003) and Haley et al. (2004). In nadir sounding, $x$ is the total vertical column density and the second step of the inversion requires knowledge of the airmass factor, which is closely related to the Jacobian $\mathbf{K}$. It is important to note that, even if the fit of the slant path column amounts does not use any prior information, the air mass factor, which relates the slant column to the vertical column, does depend on altitude-resolved prior information, even though a column retrieval has only one degree of freedom (Eskes and Boersma, 2003).

## 5.5 Nonlinearity issues

The radiative transfer equation is nonlinear. This problem can be remedied by putting the retrieval equation used in an iterative context, e.g.,

$$\hat{\boldsymbol{x}}_{\text{ML};i+1} = \boldsymbol{x}_i + \left(\mathbf{K}_i^T \mathbf{S}_{\text{y,total}}^{-1} \mathbf{K}_i\right)^{-1} \times \tag{13}$$
$$\left(\mathbf{K}_i^{\mathrm{T}} \mathbf{S}_{\text{y,total}}^{-1} \left(\boldsymbol{y} - \boldsymbol{F}\left(\boldsymbol{x}_i; \boldsymbol{b}\right)\right)\right)$$

or

$$\hat{\boldsymbol{x}}_{\text{reg};i+1} = \boldsymbol{x}_i + \left(\mathbf{K}_i^T \mathbf{S}_{\text{y,total}}^{-1} \mathbf{K}_i + \mathbf{R}\right)^{-1} \times \tag{14}$$
$$\left(\mathbf{K}_i^{\mathrm{T}} \mathbf{S}_{\text{y,total}}^{-1} \left(\boldsymbol{y} - \boldsymbol{F}\left(\boldsymbol{x}_i; \boldsymbol{b}\right)\right) - \mathbf{R}(\boldsymbol{x}_i - \boldsymbol{x}_\text{a})\right)$$

for maximum likelihood or regularized problems, respectively, where $i$ is the iteration index. The last term in the latter equation assures that the prior information will not be 'forgotten' during the iteration (see, Rodgers 2000, p. 88).

To avoid seeking an $\hat{\boldsymbol{x}}$ that is beyond the range of validity of the linear approximation $\boldsymbol{y}(\boldsymbol{x}_{i+1}) \approx \boldsymbol{F}(\boldsymbol{x}_i) + \mathbf{K}(\boldsymbol{x}_{i+1} - \boldsymbol{x}_i)$,Levenberg (1948) and Marquardt (1963) suggested a method that limits the stepwidth $\boldsymbol{x}_{i+1} - \boldsymbol{x}_i$ and turns it towards the direction of the steepest descent of the object function. The simplest formulations of this scheme are, for unconstrained and constrained inverse problems, respectively,

$$\hat{\boldsymbol{x}}_{\text{ML};i+1} = \boldsymbol{x}_i + \left(\mathbf{K}_i^T \mathbf{S}_{\text{y,total}}^{-1} \mathbf{K}_i + \lambda \mathbf{I}\right)^{-1} \times \tag{15}$$
$$\left(\mathbf{K}_i^{\mathrm{T}} \mathbf{S}_{\text{y,total}}^{-1} \left(\boldsymbol{y} - \boldsymbol{F}\left(\boldsymbol{x}_i; \boldsymbol{b}\right)\right)\right)$$

and

$$\hat{\boldsymbol{x}}_{\text{reg};i+1} = \boldsymbol{x}_i + \left(\mathbf{K}_i^T \mathbf{S}_{\text{y,total}}^{-1} \mathbf{K}_i + \mathbf{R} + \lambda \mathbf{I}\right)^{-1} \times \tag{16}$$
$$\left(\mathbf{K}_i^{\mathrm{T}} \mathbf{S}_{\text{y,total}}^{-1} \left(\boldsymbol{y} - \boldsymbol{F}\left(\boldsymbol{x}_i; \boldsymbol{b}\right)\right) - \mathbf{R}(\boldsymbol{x}_i - \boldsymbol{x}_\text{a})\right),$$

where $\lambda$ is a scalar that is adjusted during the iteration according to the local non-linearity of $\mathbf{F}$ and $\mathbf{I}$ is unity. There exist many variants of this approach, particularly with respect to the dynamical choice of $\lambda$ and the rescaling of the problem to avoid problems associated with the $\lambda \mathbf{I}$ term, first recognized by Marquardt (1963). Marks and Rodgers (1993) and Rodgers (2000)suggest the following variant:

$$\hat{\boldsymbol{x}}_{\text{reg};i+1} = \boldsymbol{x}_i + \left(\mathbf{K}_i^T \mathbf{S}_{\text{y,total}}^{-1} \mathbf{K}_i + \lambda \mathbf{R}\right)^{-1} \times \tag{17}$$
$$\left(\mathbf{K}_i^{\mathrm{T}} \mathbf{S}_{\text{y,total}}^{-1} \left(\boldsymbol{y} - \boldsymbol{F}\left(\boldsymbol{x}_i; \boldsymbol{b}\right)\right) - \mathbf{R}(\boldsymbol{x}_i - \boldsymbol{x}_\text{a})\right).$$

Butz et al. (2012) have found that in some cases a reduced step-size Gauss-Newton algorithm works much better than the Levenberg-Marquardt algorithm (Eq. 15).

Many inverse radiative transfer problems are only "moderately non-linear" (in the sense of Rodgers, 2000) in that the retrieval equations are solved iteratively, to cope with non-linearity, but linear error estimation around the best estimate is

considered adequate. If error bars are so large that they exceed the range around the best estimate where the true function $\boldsymbol{y} = \boldsymbol{F}(\boldsymbol{x})$ is sufficiently well approximated by the tangent $\boldsymbol{y} \approx \boldsymbol{F}(\boldsymbol{x}_0) + \mathbf{K}\Delta\boldsymbol{x}$, then Monte Carlo or ensemble type sensitivity studies are the only remaining options. A further benefit of Monte Carlo methods, and in particular Markov Chain Monte Carlo methods, is that the posterior distributions, which can significantly deviate from the Gaussian ones, can be explored and characterized in detail, as demonstrated by Tamminen and Kyrölä (2001), Tamminen (2004), and Brynjarsdottir et al. (2018). Also neural network based concepts have been developed and investigated in this context (see, e.g. Pfreundschuh et al., 2018). Monte Carlo error estimates exceed the computational resources needed for the retrieval by far. Thus, they are often not apt for routine applications but their range of application remains limited to representative test cases.

The issues discussed above still assume a nonlinear forward model, and only in the iterative inversion scheme the forward model is approximated by its tangent. If, however, the atmosphere is fairly transparent in the frequency range chosen, linear radiative transfer is justified, and the contributions of different atmospheric constituents become additive (Eskes and Boersma, 2003).

## 6 Sources of Errors

There are multiple categories of errors and uncertainties in atmospheric state variables retrieved from satellite measurements. These are:

1. errors caused by less than perfect measurements, which include measurement noise and calibration errors, and a less than perfect characterization of the instrument by the instrument model,

2. errors caused by inaccuracies of the radiative transfer model used in the data analysis, which include numerical approximations, missing physical processes, or uncertainties in the values used as constants by the model, particularly spectroscopic parameters,

3. errors caused by decomposing the inverse problem, giving rise to parameter errors,

4. errors caused by the constraint applied to the retrieval, which does not allow the retrieval to produce the solution that is best compatible with the measurements.

Another factor that can cause discrepancies between two sets of measurements is that the measurements might not refer to exactly the same air mass or the same time. This, along with natural variability, often explains the differences encountered (see, e.g., Sofieva et al., 2008; Verhoelst et al., 2015; Laeng et al., 2019). In the following sections, these categories of errors and uncertainties are discussed in more detail.

### 6.1 Measurement Errors

In remote sensing a number of processing steps are necessary to obtain a calibrated signal in physical units from the raw data. The latter are usually referred to as the Level-0 data. Their units depend on the instrument type and the related quantities can

be detector voltages, photon counts or similar. Level-1 processing transforms the Level-0 data into calibrated measurement data, which no longer depend on the particular measurement device used, such as radiance units or transmission. These are conventionally referred to as Level-1 data. If multiple processing steps are required, distinctions can be made between Level-1a, Level-1b, etc. data, but this distinction is of no relevance here. These Level-1 data come with auxiliary data describing the geolocation and time of the measurement, the measurement geometry, and so forth. The Level-1 data are the input to the retrieval of the atmospheric state. Estimates of the atmospheric state variables are referred to as the Level-2 data product. We use a convention that all uncertainties in the Level-1 data – including metadata – fall into the category "measurement uncertainties". The main sources of measurement uncertainties include but are not limited to measurement noise, including discretization noise; zero calibration error (i.e., that the measurement signal is non-zero even though the true radiance signal is zero, which can be understood as an additive calibration error); gain calibration (this is a multiplicative calibration error); higher order errors (e.g., nonlinear detector response); uncertainties in auxiliary data, such as measurement geometry in terms of tangent altitude, the exact time of the measurement, etc.; and straylight. Further, all these errors can be subject to a drift, i.e., there can be some time-dependence.

Unless explicitly mentioned otherwise, we apply linear theory to error estimation. This leads to generalized Gaussian error propagation of the type

$$\mathbf{S}_{\mathrm{r}} = \mathbf{J}\mathbf{S}_{\mathrm{q}}\mathbf{J}^{T}, \tag{18}$$

which is applicable separately to each independent error that can be described by a covariance matrix $\mathbf{S}_{\mathrm{q}}$ and represents the uncertainty of an input variable of $\boldsymbol{q}$. $\mathbf{S}_{\mathrm{r}}$ is the error covariance matrix of the output variable $\boldsymbol{r}$, and $\mathbf{J}$ is the Jacobian with elements $\partial r_i/\partial q_j$.

### 6.1.1 Measurement Noise

Although often 'measurement noise' is conceived as all errors which are uncorrelated in successive measurements, we use a narrower definition. In our terminology, noise encompasses only the statistical uncertainty of the measured signal caused by the indeterministic or unpredictable nature of radiative processes in the atmosphere or the instrument. Measurement noise is described by the error variance of each single spectral data point. The uncertainties are considered as uncorrelated between the single components of the measurement vector, which implies a diagonal noise covariance matrix. In some cases, however, the measurement noise covariance matrix $\mathbf{S}_{\mathrm{y}}$ has off-diagonal elements, e.g., in Fourier transform spectrometry if apodization (see, e.g., Norton and Beer 1976) and/or zero-filling is applied.

According to generalized Gaussian error analysis, the mapping of measurement noise $\boldsymbol{\epsilon}$ onto the result $\hat{\boldsymbol{x}}_{\mathrm{reg}}$ is

$$\mathbf{S}_{\mathrm{x,noise}} = \mathbf{G}\mathbf{S}_{\mathrm{y,noise}}\mathbf{G}^{T}, \tag{19}$$

with $\mathbf{G}$ as defined in Eq. 7. This method is used by the MIPAS-IMK, TES, GOMOS, OMPS-LP (NASA, IUP Bremen, and Saskatchewan), OSIRIS, SBUV or SCIAMACHY-Greifswald (Lednyts'kyy et al., 2015) and SCIAMACHY-IUP data proces-

sors. Equation (19) is applicable also to maximum likelihood retrievals just by setting the $\mathbf{R}$ term in the gain function $\mathbf{G}$ to zero. After excessive and cheerful cancellation this finally gives

$$\mathbf{S}_{\text{x,noise,ML}} = \left[\mathbf{K}^T \mathbf{S}_{\text{y,noise}}^{-1} \mathbf{K}\right]^{-1}. \tag{20}$$

Error correlations between the elements of $x$ are implicitly considered. It is important to note that such correlations will typically be present even if the measurement errors are uncorrelated and if no regularization is applied. For some retrievals, the so-called retrieval covariance matrix $\mathbf{S}_x$ is evaluated using Eq. (10). These error estimates, however, represent not only the mapping of the measurement noise onto the retrieved quantity but also the error introduced by the application of the constraint, i.e., the 'smoothing error' in the terminology by Rodgers (2000). Related problems are discussed in Section 6.4. The retrieval error evaluated by this method will represent a meaningful quantity only if the a priori covariance matrix $\mathbf{S}_a$ represents the actual variability of the atmospheric state rather than any ad hoc assumptions.

For some instruments the error estimate is based on the analysis of the residuals between the measurements and the best fitting modeled spectrum. Gauss (1821) has proven that the "residual sum of squares divided by the number of degrees of freedom is an unbiased estimator of $\sigma^2$" (translation into modern terminology by Aldrich 1998). This Gaussian $\sigma$ contains not only measurement noise but also other error components. Application of Eq. (19) to a residual-based noise characterization may be deemed more realistic than the application of this equation to pure measurement noise. However, not all uncertainties will show up in the residual. For example, spectroscopic band intensity errors of the target species will be fully compensated by erroneous retrieved concentrations and will thus create no additional spectral residual. Thus the residual-based error analysis will not provide the total uncertainty of the retrieved state variable, nor does it allow for decomposition of the error budget into its components. In particular, it will not be possible to separate random error components from systematic components. Residual-based error analysis is suitable to estimate the retrieval noise error $\mathbf{S}_{\text{x,noise}}$ only if it can be assumed that the residual is dominated by the measurement noise. In turn, the analysis of the fit residuals is an important means to validate error estimates. Residual based uncertainty estimation is used for, e.g., SCIAMACHY (U. Bremen) or ACE-FTS.

Non-optimal decomposition of the inverse problem, such as single profile retrieval, single species retrieval, etc. (Section 5.4.2), however, causes the following complication: $\mathbf{S}_{\text{x,noise}}$ contains only the noise-induced uncertainties associated with the current step of the inversion process. Propagated noise from preceding retrieval steps is formally dealt with as parameter error, although from a user perspective it is still noise (see Section 6.3).

The mapping of measurement noise into the retrieval domain depends on the retrieval approach chosen. Naturally, noise has a larger effect when regularization is kept small in order to get the best possible spatial resolution, because noise and resolution are competing quantities. However, there are also other choices in the retrieval scheme which have bearing on the measurement noise as evaluated above. In the ideal case, when the retrieval vector represents the entire atmospheric state with all its relevant variables, $\mathbf{S}_{\text{x,noise}}$ covers all uncertainties associated with everything other than the target variable. For example, if one is interested in the error of ozone abundances, any uncertainty in the ozone mixing ratio caused by water vapour uncertainties is implicitly included in $\mathbf{S}_{\text{x,noise}}$, as suggested by Marks and Rodgers (1993), Tarantola and Valette (1982), Eriksson (2000), or von Clarmann et al. (2001); for a different perspective on this issue, see Section 4.1.2 in Rodgers (2000). The situation

is different in a decomposed retrieval (Section 5.4). In the case of species-wise decomposition, the uncertainty entailed by the uncertainty of an interfering species is evaluated as parameter error. The same holds for onion peeling error propagation (Section 5.4.5). Here retrieval noise, i.e., the mapping of the measurement noise on the retrieval, accounts only for the noise of the analysis of a single tangent altitude, while the noise propagated downwards from higher altitudes is formally considered to be a parameter error. As a consequence, retrieval noise estimates from two datasets are not necessarily intercomparable. A sensible comparison is only possible between the total random errors, because the partitioning between noise and parameter errors depends on the retrieval system chosen and in particular how the inverse problem is decomposed into sub-problems.

In the context of error propagation in the the Levenberg-Marquardt algorithm (Section 5.5), it is important to distinguish two different applications.

(a) If the Levenberg-Marquardt algorithm is used only to dampen each iteration step and the iteration is only truncated after full convergence has been reached, then the $\lambda \mathbf{I}$ term has no sizeable impact on the solution, even if $\lambda \neq 0$ at the final iteration. Thus, $\lambda \mathbf{I}$ must not be included in the gain matrix $\mathbf{G}$ used for error estimation.

(b) Sometimes the Levenberg-Marquardt iteration is intentionally stopped before full convergence is reached. The rationale is to use the regularizing characteristics of the $\lambda \mathbf{I}$ term which would be lost after too many iterations. The discussion of this approach of regularization is beyond the scope of this paper, and it must suffice to mention that in this case the retrieval error has to be evaluated as suggested by Ceccherini and Ridolfi (2010).

### 6.1.2 Calibration Uncertainties

Besides measurement noise, calibration uncertainties also contribute to the measurement error (see, e.g., Kleinert et al., 2018). Often the transformation from the raw data $y_{\mathrm{raw}}$ (such as detector voltage) to the data in physical units $y$ (such as spectral radiance) uses a linear scheme such as

$$y = y_{\mathrm{raw}}/b - a, \tag{21}$$

where $a$ is a zero level offset correction and $b$ is a gain calibration coefficient (e.g., Revercomb et al., 1988, their Eq. 2). In the case of spectral measurements, both $a$ and $b$ are usually a function of frequency. Even after careful radiometric calibration, there will always be a residual zero level and gain calibration uncertainty.

Among the satellite missions considered here, the following schemes to assess the zero level calibration error are in use, or at least possible:

- Propagation of the assumed zero level calibration error in the retrieved target quantity $\sigma_{\mathrm{x,zero}}$ from the zero level calibration uncertainty in the measurement domain $\sigma_{\mathrm{y,zero}}$, using linear mapping of the type

$$\sigma_{\mathrm{x,zero}} = \mathbf{G}\sigma_{\mathrm{y,zero}}. \tag{22}$$

- A zero level correction is jointly fitted along with the target variables. In this case, this error component does not need to be assessed separately but is automatically included in the noise-induced error, at least if no constraint is applied to

the zero offset correction. Since this additional fit variable tends to destabilize the retrieval, noise-induced errors will become larger. This approach has been chosen for MIPAS-IMK, Odin/SMR, and for some of the MLS data products.

- The zero level uncertainty is added as a fully correlated component to the measurement error covariance matrix $\mathbf{S}_{x,\text{noise}}$ and thus needs no extra treatment. It is then accounted for by the error evaluated using Eq. 19. We are not aware of any processor using this method.

- The zero level uncertainty is deemed negligibly small and thus not evaluated. This approach has been chosen by SAGE I, SAGE II, SAGE III, SCIAMACHY, ACE-FTS, and OMPS LP.

Similar arguments hold for the gain calibration uncertainty, and in theory the same methods can be applied. In emission spectroscopy, however, gain calibration uncertainty is much harder to distinguish from concentration changes of the target species or temperature changes than offset calibration. For MIPAS-IMK the linear mapping method is used. On the contrary, for many limb-scatter retrievals a normalization with respect to a higher tangent height is done. As a result, the gain correction, $b$, mostly cancels out (von Savigny et al., 2003, e.g.,).

Occasionally, application of Eq. (21) is inadequate, e.g., if the detector response function is nonlinear (see, e.g., Kleinert et al., 2018). We are not aware of any data product where uncertainties of the coefficients of the nonlinear detector response function are routinely considered in the error budget of the Level-2 products. Arguably all calibration constants can be time-dependent and thus cause a drift. This issue is discussed in Section 6.7.

Another issue is frequency calibration. A spectral shift translates into a radiometric error that is highly correlated across the spectral line. The impact of such an error on the retrieval result is highly dependent on the retrieval setup and the selection of microwindows. A spectral shift correction can be jointly fitted with the target variables as it can be done in the framework of the the zero level correction. Residual frequency calibration errors after correction are still an issue of the Level-2 error budget. Since the radiometric error induced by a spectral calibration error is antisymmetric to the line center, its effect on the retrieval results will be different when the microwindow contains only part of the line.

For Odin/SMR and MIPAS a frequency offset is fitted as a scalar value characterizing a complete limb scan. Where necessary, for SCIAMACHY and OMPS (IUP Bremen), in addition to the Level-1 correction from ESA or NASA, respectively, a spectral shift/squeeze correction is determined during the pre-processing step by performing spectral fits for each line of sight and spectral window individually. IUP-DOAS and BIRA retrievals also use a shift/squeeze correction. For TES, the frequency calibration is performed as part of the Level-1B processing and is not included in the error covariances supplied with the Level 2 product. OMPS LP depends on the well-characterized Fraunhofer structure in the solar spectrum to establish and maintain its spectral registration (Jaross et al., 2014), and this work is done as a part of the Level-1 processing. For SAGE II, the filter used for the water vapor retrieval changed after launch but appeared to stabilize and a static correction for the filter spectral location and bandpass is applied in the retrieval (Thomason et al., 2004). For SAGE III/ISS, spectral calibration is performed for each observation by analyzing the apparent unobstructed solar spectrum.

### 6.1.3 Instrument characterization errors

Under instrument characterization errors we subsume incorrect estimates of measurement noise, instrument line shape errors (uncertainties in the spectral response function of the instrument), uncertainties in the field of view characterization, and so forth. Which of the error sources in this category are relevant depends on the particular instrument under assessment.

Wrongly estimated instrument noise will not only lead to incorrect error estimates but will directly affect the results. The reason is twofold. First, each element of the measurement vector $y$ is weighted by its uncertainty, and distorted weights can lead to different results. And second, incorrect noise estimates will change the weights of the a priori information and the information contained in the measurement.

The preflight characterization of the spectral response function of the instrument typically relies on a monochromatic signal.
Once in space, narrow spectral lines can be used to determine possible drifts in the instrument spectral line shape.

Depending on the field-of-view width and a shape of the response function, the field-of-view characterization can be of crucial importance for limb-scatter sensors, because the limb-scatter radiance varies by more than 5 orders of magnitude between tangent altitudes of 0 km and 100 km. In this case, small errors in the field-of-view characterization may lead to large errors in the measured limb radiances at higher tangent altitudes. Also limb scanning emission and solar occultation
measurements show a sizeable sensitivity to field-of-view uncertainties.

A number of instrument-specific level-1 issues for nadir looking UV/vis instruments are discussed in Boersma et al. (2018). These include issues with the diffuser plate used to reflect solar irradiance in the case of the GOME, and to a lesser degree, SCIAMACHY; in the case of OMI, a CCD detector row anomaly is reported.

Less than perfect correction of such instrumental issues leads to instrument characterization errors. These are, if at all,
typically evaluated using linear mapping.

### 6.1.4 Auxiliary data errors

We understand auxiliary data errors to refer to quantities that come along with the measurement data but are not usually thought of as part of the $y$ vector. Also data used for the post-retrieval conversion of results fall into this category. Typical examples are time registration errors, uncertainties in the measurement geometry such as tangent altitude pointing, and so forth. Due to
the variable nature of the errors under this category, it is impossible to suggest a common scheme. Some of these errors can be assessed by sensitivity studies or linear mapping, using the same formalism as discussed under parameter errors. Alternatively, the uncertain auxiliary data can be jointly retrieved with the target variables. In the following, the most prominent auxiliary data uncertainties are listed, and their treatment by the instrument groups is documented.

In limb sounding, pointing errors propagate to the result for various reasons. Depending on the design of the retrieval
scheme, different mechanisms may play a role. For example, the amount of air seen along the line of sight and the atmospheric state variables depend crucially on the tangent altitude. In the case of vertical gradients of atmospheric state variables, the assignment of a value which is *per se* correct to an erroneous altitude causes an error. Occultation measurements using the sun as background radiation source can depend on which part of the solar disk is seen by the instrument. The residual pointing

error to be considered in the error estimation depends on the pointing correction schemes applied. For MIPAS-IMK limb emission measurements, the first step of the retrieval chain is the simultaneous retrieval of temperature and tangent altitudes (von Clarmann et al., 2003). Results are used as known parameters in subsequent retrieval steps where trace gas abundances are retrieved. Residual errors of temperature and tangent altitudes are treated as parameter error in subsequent steps. For OMPS-LP measurements, the pointing correction is derived from radiance measurements using the absolute radiance residual method (ARRM) and the Rayleigh scattering attitude sensor (RSAS) methods (Scott et al., 1996; Moy et al., 2017). Also OSIRIS uses the RSAS method (Bourassa et al., 2018). For SCIAMACHY a correction to the pointing information is derived by analyzing measurements in the occultation geometry (Bramstedt et al., 2017) and is implemented to Level 0 to 1 data processing. The effect of residual pointing errors is assessed via Monte-Carlo-type studies for representative profiles (Rahpoe et al., 2013). Earlier SCIAMACHY analysis relied on a pointing retrieval using limb radiances below 300 nm. This so-called "knee-method" uses the known altitude of the maximum of the limb radiances originating from Rayleigh-scattering (Kaiser et al., 2004; von Savigny et al., 2005). For ODIN/SMR a scalar pointing correction is fitted for the entire limb scan.

Other auxiliary data whose uncertainties need consideration are air density profiles from external sources used for the conversion from pressure vertical coordinates to geometrical height coordinates or vice versa, as well as the conversion of mixing ratios to partial densities or vice versa (see, e.g., Keppens et al., 2015, for practical examples and their implementation by use of matrix algebra).

## 6.2 Model errors

The radiative transfer model used in the retrieval solves the radiative transfer equation and usually involves an instrument model which makes the signal comparable to what the instrument would see. Depending on the instrument type, the instrument model will include the integration of the radiance field over the finite field of view, the convolution with the spectral instrument response function, etc.

A lot can go wrong in radiative transfer modelling as our knowledge of related processes can be erroneous or inaccurate. Some known physics, such as non-local thermodynamic equilibrium, line coupling, or more sophisticated than usual line-shape functions, may be disregarded for reasons of computational efficiency. Time constraints can also lead to numerical integration being performed with limited precision or weak spectral transitions being ignored. The goal in formulating the radiative transfer model is to keep model errors from known sources much smaller than the measurement error while maintaining computational efficiency. Naturally, any unknown sources of model error are hardest to quantify. In the following, the most relevant types of known model errors are discussed.

### 6.2.1 Incomplete models

Some relevant physical processes included in $f$ may be left unaccounted for by the radiative transfer model $F$ in use (Section 4). Typical examples are non-local thermodynamic equilibrium (Non-LTE) emission, line coupling, or line shape issues. Non-LTE emissions occur when air density is so low that the excited molecule after absorption of a photon or in its nascent state will re-emit radiation before quenching redistributes the energy towards a Boltzmann distribution (e.g., López-Puertas and Taylor,

2001). Line mixing is a high pressure phenomenon where collisions transfer angular momentum, entailing energy transfer between energetically adjacent transitions (e.g., Armstrong, 1982; Bulanin et al., 1984; Strow and Gentry, 1986; Hartmann and Boulet, 1991; Hartmann et al., 2009; Thompson et al., 2012; Alvarado et al., 2013). The usual line-shape models, such as the Voigt lineshape (Voigt, 1912), may not adequately represent the true line shape (e.g., Galatry, 1961; Berman, 1972; Pickett, 1980; Thompson et al., 2012; Long and Hodges, 2012; Mendonca et al., 2019). Issues related to Zeeman coefficients, most relevant in microwave spectroscopy of mesospheric oxygen, are discussed in Larsson et al. (2019). The impacts of Zeeman splitting on microwave and submillimeter lines are often (but not universally) ignored, although proper accounting is essential for adequate representation of mesospheric signals.

Critical issues in ultraviolet or visible remote sensing are scattering and polarization. Different levels of sophistication of models refer to the treatment of sphericity of the atmosphere and orders of scattering accounted for.

If a complete model is available but not used for the operational retrieval for reasons of computational efficiency, the effect of the missing processes can be assessed via sensitivity analyses based on the complete model and considered in the error budget. If the error is of a systematic nature, the related bias can even be corrected for, and only the residual scatter begs consideration in the error analysis.

In stellar occultation, the forward model for retrievals of traces gases from UV-VIS measurements does not include the deterministic description of stellar spectra perturbations due to scintillations. This omission is not only due to complicated description of wave propagation in random media, but also by a stochastic nature of small-scale air density irregularities generated by small-vertical-scale gravity waves and turbulence. These perturbations can be, however, characterized and added as an additional, correlated in wavelength component to the measurement noise, as shown in Sofieva et al. (2009).

If no complete model is available, then it can only be hoped that the related error is sufficiently small compared to the other error sources that it has no bearing on the total error budget.

### 6.2.2 Parametric models

Not all effects of radiative transfer are always modelled according to their physical causes. Often it is more efficient to parametrize some effects and to add related parameters to the list of fit variables, i.e., to include them in the $x$ vector. A prominent example is the background signal of spectroscopic measurements where the useful information is included chiefly in the highly structured components of the measured signal, while the smooth components of the signal do not carry the desired information. The smooth background signal is often hard to model on a physical basis because it depends on too many unknowns, but is essential for a good spectral fit. To solve this problem, in different parts of the remote sensing community almost equivalent solutions have been identified. In infrared emission spectroscopy, it is common practice to fit an either flat or tilted background continuum optical thickness which accounts for aerosol and particle emission and the far-wing-contributions of remote spectral lines which are not considered explicitly in the line-by-line calculation. This approach has been discussed and justified by von Clarmann et al. (2003) but has long been used in the context of occultation measurements with the Atmospheric Trace Molecule Experiment (ATMOS; C. P. Rinsland, personal communication, 1987). The equivalent in DOAS retrievals is the polynomial which is added to the optical thickness governed by the transmission of the trace species (see, e.g., Eskes and

Boersma, 2003, their Eq. (6)). If the parametrization chosen offers too few degrees of freedom, it will not describe the smooth part of the signal properly and thus cause an error in the retrieved value of the target quantity. Conversely, a polynomial with too many degrees of freedom may remove a part of the differential signal of the target gas.

### 6.2.3 Numerical issues

The numerical solution of the radiative transfer equation requires a lot of integration, e.g., to integrate the spectral radiances over the field of view based on a finite number of so-called pencil beams; the spectral grid on which the radiative transfer is calculated has a finite width; radiative transfer through the atmosphere is in most models based on a finite number of layers or levels, just to name a few. Any improvement of computational accuracy goes along with increased computational effort. For most satellite data processors, the setting is chosen in a way that these issues produce a retrieval error which is so small compared to the leading error sources that it can be ignored in the error budget.

### 6.2.4 Model constants

The main constants of relevance here include spectroscopic data, quenching rates, refractive indices, etc. The values of other constants (radius of Earth, gas constant, molecular weights, etc.) are known at an accuracy which renders analysis of related retrieval errors unnecessary. Estimation of the impact of spectroscopic errors poses some serious problems.

A major problem in the propagation of spectroscopic data errors is that, in some cases, no uncertainties of cross-sections are available. Also, when they are available, information on error correlations is not provided. If a retrieval uses, say, a large number of ozone lines, it would be of utmost importance to know whether errors in the intensity of these lines are correlated (e.g., because the uncertainties are attributed to uncertainties in the gas amount in the cell used in the lab where the spectroscopic parameters were measured) or uncorrelated (because errors are dominated by noise in the lab measurement or because the spectroscopic information stems from different lab measurements). In the uncorrelated case the errors would randomize while in the correlated case they would fully survive the error propagation for a retrieval using multiple spectral lines.

To exemplify another issue, consider a gas-wise sequential retrieval where $H_2O$ is retrieved first, and this $H_2O$ profile is then used as a known parameter in a retrieval of ozone in another spectral region. It is possible for the spectroscopic errors of $H_2O$ to cancel out in the ozone retrieval if these errors are consistent over the entire spectrum. For example, if $H_2O$ line intensities are too high, too little $H_2O$ will be retrieved. Subsequently, during the ozone retrieval, the combination of the too little $H_2O$ with the too large line intensities produce the correct impact of $H_2O$ on the modeled spectra. This results in the $H_2O$ line intensity errors not propagating into the retrieved ozone concentrations.

The usual way to estimate the propagation of spectroscopic data errors is to conduct sensitivity studies with perturbed spectroscopic data. Since, as stated above, the correlations between spectroscopic data errors are unknown and not reported in commonly used spectroscopic databases, these sensitivity studies render only a crude estimate of the related retrieval error.

In the case of retrievals of trace gas abundances, one might argue that uncertainties of the line intensity can be mapped directly onto the target concentration retrieval. Because both the line intensity and abundance appear reciprocally in the exponent of Beer's law, the non-linearity of the radiative transfer equation has no bearing on the line intensity error propagation. It

has, however, been shown that it is not sufficient to restrict related error analysis to the line intensities. For example, pressure broadening has a sizeable effect in the infrared and microwave regions (e.g. Urban et al., 2005; Glatthor et al., 2018). In this case no direct mapping is possible and full sensitivity studies are needed. Connor et al. (2016) had found, in their linear error analysis for OCO-2 retrievals, that a constant perturbation even to $CO_2$ line intensities did not in fact map to a constant impact on the $XCO_2$ retrievals within the NASA OCO-2 algorithm. Within that algorithm, the impact of line intensity perturbation on the retrieved $XCO_2$ varies spatially and appears to depend on the surface brightness. Inclusion of surface albedo terms in the state vector for the OCO-2 algorithm gives rise to this information cross-talk.

The propagation of uncertainties of model constants follows the same formalisms as proposed for uncertainties in atmospheric parameters (Section 6.3).

In the NASA ACOS/OCO-2 and OCO-3 $CO_2$ retrieval algorithm spectral residuals caused by imperfect spectroscopy, solar model and instrument characterization are dealt with by fitting scaling factors to fixed spectral residual patterns. These patterns are the Empirical Orthogonal Functions (EOFs) that result from a singular value decomposition of spectral residuals from training retrievals (O'Dell et al., 2018). A similar approach was adopted independently by Lange and Landgraf (2018) for retrievals of methane from GOSAT thermal infrared spectra.

## 6.3  Parameter errors

We define parameter errors as those errors originating from the decomposition of the full retrieval problem such that a part of the atmospheric state is assumed to be already known and thus not included in the retrieval vector $\boldsymbol{x}$ (see, Section 5.4). Parameter errors can be caused by, e.g., temperature uncertainties in a trace gas retrieval, not accurately known abundancies of interfering species or aerosol parameters, or surface albedo in the case of nadir sounding or limb scattering measurements, just to name a few. The assumed values can derive from either a preceding retrieval step or from climatologies or any other source of prior information. The ideal sequence of operations has the first atmospheric state variables retrieved being those whose signal is only weakly dependent on or interfered by other state variables. Once known, these values can be used for subsequent retrieval steps as "fixed" parameters. Error propagation has to be considered.

The impact $\Delta x$ of errors in parameters can be estimated via sensitivity studies, where a measurement is simulated with parameter $b$ that is perturbed by a certain amount $\Delta b$, e.g., one standard deviation of its uncertainty.

$$\Delta\boldsymbol{x} = \mathbf{G}\left(\boldsymbol{F}(\hat{\boldsymbol{x}};b+\Delta b) - \boldsymbol{F}(\hat{\boldsymbol{x}};b)\right) \tag{23}$$

This scheme is used, e.g., for MIPAS IMK, SCIAMACHY-Greifswald, SCIAMACHY IUP Bremen (see, Rahpoe et al. 2013 or E. Malinina et al. 2018 and SMILES-NICT (see, e.g., Sato et al. 2012 or Sato et al. 2014).

If the parameter $b$ is a vector, whose elements' error correlations are known and relevant, generalized Gaussian error propagation can be applied

$$\mathbf{S}_{\mathrm{x,b}} = \mathbf{G}\mathbf{K}_{\mathrm{b}}\mathbf{S}_{\mathrm{b}}\mathbf{K}_{\mathrm{b}}^{T}\mathbf{G}^{T}, \tag{24}$$

where $\mathbf{K}_{\mathrm{b}}$ is the Jacobian matrix representing the sensitivities $\frac{\partial y_{\mathrm{m}}}{\partial b_{j}}$ of the measurements with respect to a changing parameter $b_{j}$.

Depending on the source of the information on the parameter vector – climatology, preceding retrieval step, independent measurements, or whatsoever, the parameter errors can be correlated or uncorrelated in space and time.

Occasionally errors are of a mixed nature, e.g., if a quantity is jointly retrieved along with the target quantity but strongly constrained. In this case, the parameter actually is part of the retrieval vector $\hat{x}$ but its value still depends largely on the a priori information. Uncertainties that derive for this situation are discussed in Section 6.4.

### 6.3.1 Error propagation in onion peeling

In onion peeling (Section 5.4.5) the ray path with the highest tangent altitude is analyzed first. In the second step, the results of the first step are used as known parameters. Thus the retrieval error of the first step has to be considered as a source of parameter error in the second step, and so forth. Explicit error propagation through an onion peeling retrieval has been studied, e.g., by Noël et al. (2016).

Alternatively, the onion peeling retrieval error can be estimated using a Monte Carlo method. For the solution profile $x$ a limb sequence of measurements is calculated. Artificial noise with the same characteristics as the real measurement noise is superimposed upon the measurements. A sample of limb sequences is generated, based on the same forward radiative transfer calculations but different in the actual realization of the random noise. For each of these simulated limb sequences a retrieval is performed and, from the scatter of these results, the retrieval error covariance matrix $\mathbf{S}_{x,noise}$ is calculated.

### 6.4 A priori information

In order to avoid wording that is too abstract, we assume that the retrieval vector represents vertical profiles of atmospheric state variables. However, with some adjustments the mathematical concept is applicable to 2-D or 3-D fields of atmospheric state variables as well. The framework is also applicable to column retrievals. In this case, the retrieval vector has only one element.

By performing regularized retrievals invoking Eq. (4) or variants of it, the retrieved atmospheric state will deviate from that one of the discrete profiles which is most consistent with the pure measurement information. Depending on the regularization chosen, the profile can be pushed towards the a priori profile and the vertical resolution can be worse than what the gridwidth might suggest. The resulting profile is a mixture of the measurement information and the a priori information used. For the interpretation of constrained retrievals it is of utmost importance to have tools available to diagnose the content of a priori information in the retrievals. As in Rodgers (1976, 1990, 2000), one can calculate the derivative of the retrieved state with respect to the true state, and call the resulting matrix the "averaging kernel matrix"

$$
\begin{aligned}
\mathbf{A} &= \frac{\partial \hat{x}_i}{\partial x_j} = \mathbf{G}\mathbf{K} = \\
&= \left(\mathbf{K}^T \mathbf{S}_{y,total}{}^{-1} \mathbf{K} + \mathbf{R}\right)^{-1} \mathbf{K}^T \mathbf{S}_{y,total}{}^{-1} \mathbf{K}.
\end{aligned} \tag{25}
$$

The rows of the averaging kernel represent the weighting functions, which determine to what degree the result at altitude level $i$ depends on the true atmospheric state at altitude level $j$. Its columns represent the response of the retrieval to a delta perturbation at a single altitude level.

The discrete averaging kernel presented above is only an approximation because it describes only the response to perturbations of the true atmosphere which can be represented in the discretization chosen. In the true atmosphere perturbations can occur on much finer scales, and, strictly speaking, the averaging kernel is a continuous function. An averaging kernel in a coarse discretization will not allow to restore the averaging kernel on any finer grid.

If a joint retrieval of profiles of multiple different quantities is made, the above refers to the diagonal blocks of the averaging kernel matrix which refer to the quantity under consideration. The presence of non-negligible off-diagonal blocks indicates a significant interference between the species introduced by the regularization scheme.

The averaging kernel of a fully converged Levenberg-Marquardt retrieval equals that of the respective retrieval without the $\lambda \mathbf{I}$ term because after convergence this term has no impact on the solution. If the iteration is ended prematurely in order to use the Levenberg-Marquardt method to regularize ill-posed inverse problems, then the averaging kernel has to be calculated as suggested by Ceccherini and Ridolfi (2010).

When $\mathbf{S}_{\mathrm{y,total}}$ is approximated by $\mathbf{S}_{\mathrm{y,noise}}$ in the retrieval, the same approximation must be used for the evaluation of the averaging kernel. That is to say, in this case the averaging kernel should be calculated involving $\mathbf{S}_{\mathrm{y,noise}}$ instead of $\mathbf{S}_{\mathrm{y,total}}$.

Conversely, the derivative of the retrieved state with respect to the a priori information is $\mathbf{I} - \mathbf{A}$. With this, the retrieval can be rewritten as

$$\hat{\boldsymbol{x}} = \mathbf{A}\boldsymbol{x} + (\mathbf{I} - \mathbf{A})\boldsymbol{x}_{\mathrm{a}}. \tag{26}$$

For the retrieval of column amounts, the sensitivity of the column to the true state values at different altitudes can be represented by the column averaging kernel $\boldsymbol{A}_{\mathrm{col}}$, which is, as opposed to the averaging kernel described above, not a square matrix but a row vector (Wunch et al., 2010, see, e.g.,)

$$\boldsymbol{A}_{\mathrm{col}} = \boldsymbol{h}^T \mathbf{A}, \tag{27}$$

where $\boldsymbol{h}^T$ is the column operator whose multiplication with the vertical profile yields the vertical column density and where $\mathbf{A}$ is the regular profile averaging kernel as described above, but referring to partial columns instead of concentrations. The formalism is not quite the same as that described by Eskes and Boersma (2003, their Section 2) for DOAS column retrievals. Both conceptions are similar in that the number of elements of the vector representing the retrieved state can be smaller than that of the a priori information. The interpretation of both conceptions of the column averaging kernel, however, is different. The former (Wunch et al., 2010) describes the dependence of the retrieved column on the true profile, while any deviation of its row sum from unity hints at some influence of the a priori profile $\boldsymbol{x}_{\mathrm{a}}$ which may be used to regularize the ill-posed retrieval. The latter (Eskes and Boersma, 2003) accounts only for the different weights of the involved layers due to their respective airmass factors[7]. In the comparison of DOAS vertical columns with columns obtained from integration over vertically resolved profiles, multiplication of the comparison profile with this averaging kernel instead of unweighted summation of the partial column amounts of the layers will remove air-mass-factor-related components from the difference.

---

[7]Here a notation-related *caveat* is in order. Eskes and Boersma (2003) use the symbol $\boldsymbol{x}_{\mathrm{a}}$ and the term 'a priori profile' to denote the linearization point in a linearized solution of the radiative transfer model, while Rodgers (2000) uses this symbol to denote a Bayesian prior. In this paper, we roughly follow Rodgers' convention.

Usually regularization will entail that the retrieved state $\hat{x}$ is a smoothed and possibly biased representation of the true state $x$. Rodgers (2000, p. 48) offers two possible interpretations of the retrieved state $\hat{x}$. It can either be conceived as a smoothed estimate of the true state, or it can be construed as an estimate of the smoothed true state. The choice of the interpretation has major impacts on the error budget, which are discussed below. All this is not to say that the effect of the prior information is restricted to smoothing. If the averaging kernel is asymmetric, the resulting profile shape can be distorted in a sense that the extrema of a profile can be shifted upward or downward (see, e.g., the HOCl profiles in Jackman et al., 2008, their Fig. 12) or bias the result (see, e.g., Bhartia et al., 2013). The averaging kernel matrix contains information on the dependence of the result on the true state and the a priori assumption, the vertical resolution, and the information displacement.

### 6.4.1   The retrieved state as a smoothed estimate of the truth

As stated above, a retrieval can be understood as a smoothed estimate of the truth or an estimate of the smoothed truth. In the first case, any deviation between the estimate and the truth which is caused by the regularization of the retrieval has to be included in the error budget. Rodgers (2000) calls this error component 'smoothing error' and has suggested the following formalism to estimate it (Rodgers, 1990):

$$\mathbf{S}_{x,\text{smoothing}} = (\mathbf{I} - \mathbf{A})\mathbf{S}_a(\mathbf{I} - \mathbf{A})^T. \tag{28}$$

While in principle this formulation is a direct consequence of generalized Gaussian error propagation, the inclusion of the smoothing error in the reported error budget has been critically discussed by von Clarmann (2014). His main argument refers to the fact that this estimate does not refer to the difference between the retrieved and the true state but only to the difference between the estimate and the true state as represented on the grid on which $\mathbf{S}_a$ has been evaluated. This leads to the undesirable effect that a smoothing error evaluated on a coarse grid will be smaller than a smoothing error evaluated on a fine grid. Further, a smoothing error evaluated on a coarse grid and then propagated onto a fine grid, will be smaller than the smoothing error evaluated directly on the fine grid, although the interpolation between the grids is a linear operation, which is another undesirable outcome.

Since interpolation of profiles to other grids is a standard operation, it is not advisable to include the smoothing error in the error budget without a *caveat*. Instead, the averaging kernels should be communicated to the user, allowing them to evaluate the smoothing error on the final working grid.

In this context it should be mentioned that error estimates according to Eq. (10) include a smoothing error component and should not be used to calculate the error budget because the data user might not be aware of related problems and might, when interpolating profiles on a finer grid, propagate these error estimates to the finer grid.

Further, Rodgers (2000) points out that Eq. (28) will only yield a meaningful smoothing error if $\mathbf{S}_a$ is not just a constraint matrix chosen *ad hoc* to regularize the inversion but a real statistical description of the variability of the actual states around the mean state used as $x_a$. This criterion should even be more rigorous: The maxim of the most specific reference class has to be applied (Hempel, 1965). For example, to calculate the smoothing error of a tropical ozone profile retrieval we cannot use a global ozone climatology, although the particular tropical profile can be conceived as a member of the distribution formed

by the global average and the global variance. To calculate the smoothing error of a tropical ozone retrieval, we must not use an ozone climatology built from a whole year of global – including polar – ozone data, because this would over-estimate the ozone variability. The most specific reference class will be a homogeneous reference class whose internal variability is, as far as known, purely random.

Not all applications of a retrieval scheme of the type Eq. (6) use a climatological mean profile as a priori. For example, for the upcoming TEMPO mission, actual ozone measurements have been tested to be used as a priori (see, e.g., Johnson et al., 2018). In such applications, the $\mathbf{S}_a$ matrix contains the estimated uncertainties of the individual ozone measurements used instead of the climatological variability. The standard approach of maximum a posteriori retrievals with climatological prior is based on the assumption that a climatology based on data collected in the past will also be appropriate for the actual case. Hume (1748)

was the first to show that this assumption cannot conclusively be inferred from anything. The use of actual measurement data of the same part of the atmosphere from independent sources as prior information dispenses with this assumption and reduces the maximum a posteriori retrieval to a sort of optimal average of two independent measurements. The smoothing error evaluated for this kind of retrieval scheme represents the propagated uncertainties of the measurement(s) used as prior information.

A particular problem is the evaluation of the smoothing error difference (occasionally, perhaps more adequately, called

'smoothing difference error') of a pair of measurements. For this purpose, Rodgers and Connor (2003) suggest that the retrieved profiles should first be transformed to the same a priori profile $\boldsymbol{x}_c$, using the general transformation scheme Eq. (10.48) of Rodgers (2000)

$$
\begin{aligned}
\hat{\boldsymbol{x}}_{\text{new}} \quad = \quad & \left( \hat{\mathbf{S}}_x^{-1} - \mathbf{S}_{a,\text{old}}^{-1} + \mathbf{S}_{a,\text{new}}^{-1} \right)^{-1} \\
& \left[ \hat{\mathbf{S}}_x^{-1} \hat{\boldsymbol{x}}_{\text{old}} - \mathbf{S}_{a,\text{old}}^{-1} \boldsymbol{x}_{a,\text{old}} + \mathbf{S}_{a,\text{new}}^{-1} \boldsymbol{x}_{a,\text{new}} \right],
\end{aligned}
\tag{29}
$$

where profile and covariance matrix $\boldsymbol{x}_{a,\text{old}}$ and $\mathbf{S}_{a,\text{old}}$ represent the initially used prior information to be removed, and where $\boldsymbol{x}_{a,\text{new}}$ and $\mathbf{S}_{a,\text{new}}$ represent the new prior information be included instead. In the given application, the old prior information is that used for the retrieval, and the new one, $\boldsymbol{x}_{a,\text{new}}$, is the prior information $\boldsymbol{x}_c$, valid for the comparison profile. This transformation is possible within linear theory and adequate if and only if one result is in the linear domain of the other. If the profiles are provided on different grids, a transformation of the profiles, covariance matrices, and averaging kernels to a

common grid must precede the above transformation (see, e.g., Stiller et al. 2012 or Eckert et al. 2014 for sample applications, or Keppens et al. 2019 for a summary of methods.) Then the smoothing error difference is evaluated, where $\mathbf{A}_1$ and $\mathbf{A}_2$ are the averaging kernels of the retrieved profiles $\boldsymbol{x}_1$ and $\boldsymbol{x}_2$, all after application of the transformations outlined before.

$$
\mathbf{S}_{x_1 - x_2;\text{smoothing}} = (\mathbf{A}_1 - \mathbf{A}_2) \mathbf{S}_c (\mathbf{A}_1 - \mathbf{A}_2)^T
\tag{30}
$$

Here $\mathbf{S}_c$ is the a priori covariance matrix describing the variability of the atmospheric state around a priori profile $\boldsymbol{x}_c$ valid for

the comparison profile. Rodgers and Connor (2003), however, do not specify what type of $\boldsymbol{x}_c$ profile is adequate. The common a priori profile $\boldsymbol{x}_c$ can by no means be freely chosen. Here the maxim of the most specific reference class Hempel (1965) becomes important again. When large samples of collocated measurements are compared, the appropriate reference class for

instrument 1 is not necessarily the appropriate reference class for instrument 2, and vice versa, because both instruments might typically sample different parts of the atmosphere. The adequate sample with which to build the statistics needed to evaluate the smoothing error of the difference is that which is representative for the actual collocations of both measurements.

The criticism of the smoothing error as formulated above (After Eq. 28) does not apply to the smoothing error difference.

### 6.4.2 The retrieved state as an estimate of the smoothed truth

With the interpretation of the retrieved state as an estimate of the smoothed truth, we accept that measurements can only provide a finite-resolution representation of the truth and do not consider this as an error component of the measurement (not to mention the philosophical problems associated with what an infinitely resolved atmospheric state shall be; see von Clarmann 2014 for a critical discussion). The only important thing to consider is to avoid comparison of apples and oranges: Differences of atmospheric state variables are only meaningful if the data contain the same amount of the same a priori information and have the same vertical resolution. This is not typically given when two measurements are compared, and a part of the observed differences is thus due to related artefacts.

If the contrast in resolution is large enough to consider the better resolved measurement as both practically ideal compared to the other one and practically free of a priori information, then it is common practice to apply the averaging kernel of the coarser resolved measurement to the better resolved measurement (see Section 6.4.3 for concepts of altitude resolution).

$$\hat{\boldsymbol{x}}_{1,\text{smoothed}} = \mathbf{A}_2 \hat{\boldsymbol{x}}_{1,\text{original}} + (\mathbf{I} - \mathbf{A}_2) \boldsymbol{x}_{\text{a},2} \tag{31}$$

Here index 1 refers to the better resolved measurement and index 2 to the coarser resolved one. The other indices are self-explanatory. To our best knowledge, this approach was first suggested by Connor et al. (1994). This approach is also commonly applied when measurements are compared to model data. In this case the averaging kernel of the measurement is applied to the modeled atmospheric state. Within linear theory and the assumption in force that the better resolved data set contains no sizeable amount of a priori information, the Connor et al. approach indeed solves the problem that the original datasets are not directly comparable due to different vertical resolutions. The problem of interpolability of averaging kernels is discussed in Arosio et al. (2018).

Problems occur when linear theory is no longer adequate to describe the problem. For example, B. Funke has , during the preparation of Funke et al. (2017), encountered the following difficulty: When the MIPAS averaging kernels were applied to modeled Nitric Oxide (NO) distributions, the discrepancies between the modeled and the measured NO distributions were found to be larger than in the comparison without application of MIPAS averaging kernels. The following reason has been identified. The measured and the modeled NO distributions were so different that the application of the MIPAS averaging kernel to the modeled NO distributions was no longer justified. That is to say, the Connor et al. method is only valid if the data sets to be compared are similar enough to justify the assumption of linear theory. What would have been needed were MIPAS averaging kernels calculated for the modeled NO distribution. The latter approach, i.e., to calculate dedicated averaging kernels for model atmospheres, has been chosen, e.g., by Schneider et al. (2017) and references therein.

In this context another caveat is in order: Averaging kernels usually depend on the units in which the atmospheric state is expressed. For example, averaging kernels evaluated for volume mixing ratios must not be applied to number density profiles. Some authors prefer to use so-called 'fractional averaging kernels' instead, which refer to the relative instead of the absolute change of the state variable and are thus unit-independent (Keppens et al., 2015). However, again the caveat applies that these can be calculated only within linear theory, with the assumption in force that the retrieved profile is sufficiently close to the true profile.

### 6.4.3 Altitude resolution

Often the full information contained in the averaging kernel is summarized in simpler terms. The most important simple diagnostics that partially describe the content of the averaging kernel are vertical resolution, information displacement, and measurement response. We first discuss the concept of vertical resolution.

Vertical resolution of the retrieval, not to be confused with the vertical sampling implied by the tangent altitude increments or the instantaneous field of view of a limb sounder, describes the ability to distinguish separate features in a vertical profile. It cannot be better than the width of the vertical retrieval grid on which the results are presented. The latter should thus be chosen not to limit the resolution of the measurement. All information on the vertical resolution is included in the averaging kernel matrix. Contrary to common belief, a wide field of view or an observation geometry other than limb or with coarse vertical sampling do not *per se* degrade the vertical resolution of the measurements. The altitude resolution of the retrieval is determined only by the vertical grid and the regularization. It goes without saying, however, that a wide field of view or any sub-optimal observation geometry often forces the retrieval scientist to use a stronger regularization to get useful results, which, in turn, will degrade the altitude resolution. Thus, the field of view geometry or sampling have an indirect influence on the vertical resolution of the retrieval, which is fully accounted for by the averaging kernel matrix and does not need extra treatment. Vertical oversampling in limb sounding, i.e. the use of a tangent altitude spacing finer than the width of the instantaneous field of view of the instrument still allows a useful vertical resolution finer than the field of view (Roscoe and Hill, 2002). A measurement mode of this type has been employed, e.g., for MIPAS for the measurements recorded after 2004 (Fischer et al., 2008), and is standard for sub-millimeter and microwave measurements such as MLS (Barath et al., 1993; Waters et al., 2006) or ODIN/SMR(Urban et al., 2005).

Rodgers (2000) reports four measures of the vertical resolution, all based on the averaging kernel matrix. Two of these measures are commonly used. The first is the full width at half maximum of the respective row of the averaging kernel matrix, or, in the case of a retrieval of multiple quantities, the part of the row associated with diagonal block associated with the quantity of interest. The second is the reciprocal data density, which is the local grid width divided by the respective diagonal value of the averaging kernel matrix (Purser and Huang, 1993). In less than well-behaved retrievals or at the extreme ends of the profiles, where the maximum of the averaging kernel does not coincide with its nominal altitude, the latter provides better intelligible results.

The Backus and Gilbert (1970) spread (shown here in a generalized variant introduced by Rodgers 2000)

$$s(z) = 12 \int (z - z')^2 A^2(z, z') dz' / (\int A(z, z') dz')^2, \tag{32}$$

where $z$ is altitude and $A$ the respective element of the averaging kernel matrix $\mathbf{A}$, was found by Keppens et al. (2015) to be most informative under certain circumstances. Obviously in the case of the retrieval of different state variables the summation here and all similar applications should only be performed inside the diagonal sub-blocks, corresponding to each retrieval quantity. Among other reasons discussed towards the end of the section, going outside the sub-blocks could mean that even different units would be mixed.

A drawback of the Backus-Gilbert spread is that it depends largely on the grid on which the retrieval is performed. The averaging kernel of a retrieval performed on a finer vertical grid will have more pronounced side-lobes which are simply not resolved by an averaging kernel evaluated on a coarser grid. If we conceive the coarse-grid averaging kernel as a superposition of fine-grid averaging kernels, the side-lobes cancel out. The Backus-Gilbert spread is very sensitive to such side-lobes and will thus inadequately 'punish' the fine-grid retrieval by giving large weight to these side-lobes and thus assigning a large 'spread' to them. It thus does not seem suitable for a largely grid-independent measure of the vertical resolution.

Obviously, the altitude resolution can be altitude dependent. Usually, the averaging kernel matrix is evaluated on the grid on which the retrieval is performed, because the Jacobians needed are often a by-product of the retrieval. The disadvantage of this approach, however, is that the averaging kernel does not represent any subgrid smoothing effects. Averaging kernels evaluated on a finer grid, which, by the way, are no longer square, can in principle be provided if the related Jacobians are made available, but this is hardly ever done. The ideal averaging kernel is the identity matrix. This averaging kernel matrix corresponds to a maximum likelihood retrieval where the weight of prior information is zero. Here the altitude resolution is equal to the gridwidth of the retrieval. In agreement with our intuition, the altitude resolution cannot be better than the width of the grid on which the retrieval is performed.

It is a common misconception that the averaging kernel characterizes the vertical resolution of the estimated profiles $\hat{x}$. Instead, the averaging kernel characterises the vertical resolution of the difference between the retrieved profile and the a priori. If the a priori profile is highly structured and thus resolves fine scales, these structures are propagated onto the result $\hat{x}$.

As is often the case, precision and resolution share a trade space in remote sounding retrievals. We see from Eq. (19) (with Eq. 7 inserted) that weaker regularization will increase the impact of measurement noise. Conversely, weaker regularization will, according to Eq. (25), push the averaging kernel towards the identity matrix, which is associated with the optimally obtainable resolution of a profile at a given discretization, but can result in noise which makes the retrieval useless.

In the context of altitude resolution, a cautionary note is in order. The altitude resolution is neither identical to the grid width nor with the information smearing. In a regularized retrieval the vertical resolution is coarser than the retrieval grid. Only in an unconstrained maximum likelihood retrieval is the vertical resolution equal with the gridwidth. Conversely, the vertical resolution of measurements that are sensitive to a very small air parcel is only limited by the vertical grid, and the sampling theorem (Shannon, 1948) applies. That is to say, *in situ* measurements from a cruising aicraft have no altitude resolution in the sense as defined here although the measurements may be practically point measurements due to the small vertical extent

of the air parcel probed. In remote sensing the radiative transfer equation, which is integrated over all altitudes relevant to the retrieved profile, acts as an anti-aliasing filter, and the sampling theorem is of no concern.

Another concept closely related to the concept of altitude resolution is that of the degrees of freedom of the signal. This number is calculated as the trace of the averaging kernel matrix (Rodgers, 2000).

### 6.4.4 Information displacement

Ideally the maximum, the mean, and the median of the averaging kernel coincide with the nominal altitude but "it ain't necessarily so" (George and Ira Gershwin, 1935). Any displacement reflects the fact that the interpretation of the retrieved profile without consideration of the averaging kernel is, mildly speaking, misleading. This problem can often be remedied by comparing the retrieved profile not with any reference profile, but with a reference profile to which the averaging kernel matrix of the remotely sensed profile has been applied according to Eq. (31). Again, the caveat that this method is valid only within linear theory applies. A measure of the information displacement is the centroid offset of the averaging kernel (see, e.g., Keppens et al. 2015 and references therein).

An example of the importance of this issue is found in Jackman et al. (2008) who compared modeled HOCl distributions to those measured by MIPAS (von Clarmann et al., 2006). Maximum concentrations are displaced by more than $5\,\mathrm{km}$ before consideration of the averaging kernels.

### 6.4.5 Regularization bias and measurement response

If the a priori profile $\boldsymbol{x}_\mathrm{a}$ deviates from the expectation value of the true profiles $\langle\boldsymbol{x}\rangle$, the regularized retrievals may be biased. The component of the bias related to regularization, $\langle\hat{\boldsymbol{x}}-\boldsymbol{x}\rangle_\mathrm{regul.}$ is

$$\langle\hat{\boldsymbol{x}}-\boldsymbol{x}\rangle_\mathrm{regul.} = \langle(\mathbf{I}-\mathbf{A})(\boldsymbol{x}_a-\boldsymbol{x})\rangle \tag{33}$$

We have to distinguish two cases: Firstly, smoothing can cause biases, because, e.g., a sharp maximum of the true profile will always be reproduced too low, and the wings of the maximum will be reproduced too high. This typically occurs when a smoothing constraint as discussed after Eq. (5) is used in combination with an unstructured a priori profile. This type of bias, however, is only relevant if the retrieval is conceived as a smoothed representation of the truth as discussed in Section 6.4.1. If the retrieval is conceived as an estimate of the smoothed truth as discussed in Section 6.4.2, this type of bias is of no concern.

Secondly, regularization can cause a bias by pushing the results systematically towards higher or lower values. Any such effect besides mere smoothing is characterized by the measurement response function $q$ (occasionally also called 'vertical sensitivity'), a concept which was first applied to atmospheric measurements by Eriksson (2000) and Baron et al. (2002). It is defined as the sum over the row of the averaging kernel matrix.

$$q_i = \sum_{j=1}^{n} a_{i,j} \tag{34}$$

In the case of a multi-species profile retrieval, the sum is calculated over the subblock or the averaging kernel matrix referring to the profile under assessment. If the regularization of a retrieval provides a smoothed version of the truth, without

systematically pushing results towards greater or smaller values, the sum of the elements over each row of the averaging kernel should be unity. Any deviation of the row-sums from unity thus hints at an influence of the constraint that is beyond pure smoothing. The measurement response function is retrieval-unit-dependent.

Even if the averaging kernel matrix is far from unity, a measurement response function close to unity indicates that the retrieval is, putting measurement errors aside for a moment, a smoothed but unbiased representation of the true profile. Conversely, values of the measurement response function deviating by an appreciable amount from unity indicate a large influence of the prior information not only on the profile shape but also on the integrated values. Interpretation of the measurement response, however, requires some caution. Any non-zero $\langle \boldsymbol{x}_{\mathrm{a}} - \boldsymbol{x} \rangle$ will cause a bias in this case.

The row sum of the averaging kernel, which makes up the measurement response, consists of summands which refer to a perturbation by the same amount in each layer, where, again, the 'sameness' is unit-dependent. Such a perturbation can be fully realistic in one layer and fully unrealistic in other layers, depending on the retrieval-units. The evaluation of the measurement response is particularly problematic in cases where the profile values cover a wide dynamic range. This is the case, e.g., for the $H_2O$ mixing ratios or when the retrieval units are number density. For example, a certain perturbation in terms of number density of a certain trace gas at lower altitudes can correspond to merely moderate changes in the mixing ratio, while the same perturbation in terms of number density at high altitudes where air density is low will correspond to unrealistic mixing ratios. Thus, averaging kernels evaluated in units of number density can make the dependence of the result on values at higher altitudes appear large although the underlying perturbation in terms number density does nott represent any realistic condition. These unrealistic large contributions from higher layers inhibit the correct interpretation of the measurement response.

### 6.4.6 Regularization crosstalk

The discussion of the averaging kernel matrix and smoothing error was focused on the retrieval of single quantities so far, e.g., vertical profiles of a single state variable. Often, however, multiple different state variables are jointly retrieved in one leap. In this case the regularization constraining one state variable can affect the result of the other and vice versa. More specifically, the smoothing error of one variable can propagate onto the result of the other variable and thus give rise to regularization crosstalk. If the full (multi-variable) averaging kernel matrices are stored, the resulting parameter errors can be evaluated using Eq. 28. The regularized joint fit lies between the extremes of treating the other variable as a known parameter during the retrieval and the unconstrained joint-fit of both quantities.

### 6.4.7 Implicit Regularization via Coarse Discretization

In order to avoid problems due to formal regularization, often regularization by means of a coarse discretisation is used (e.g., von Clarmann et al., 2015). The averaging kernel of a preceding regularized retrieval on a fine grid provides the information how the suitable coarse grid can be defined.

The retrieval of vertical column densities in cases when no sufficient information on the vertical distribution of the state variable is available pushes this rationale to extremes (see Section 5.1). The column amount, however, can depend on the assumed shape of the vertical profile of the constituent used for the radiative transfer calculation, because a certain amount of a

gas at one altitude can affect radiative transfer differently than the same amount of a gas at another altitude. The causes can be, depending on the measurement principle, pressure broadening of lines, temperature dependence of the absorption coefficients and the source function, and others. This issue has been discussed for infrared emission spectroscopy by Blom et al. (1994). For DOAS measurements, see, e.g., Eskes and Boersma (2003) and references therein. To assess related effects, the column averaging kernel (Eq. 27) is an adequate tool.

### 6.4.8  Related issues

In the context of averaging kernels and vertical resolution a few further remarks are in order.

– Time series of state values at a given altitude are particularly problematic when the averaging kernel is time-dependent in itself. Here it may help to remove the prior information from the data along with resampling on a coarser grid in order to achieve $\mathbf{A}_{\text{coarse}} = \mathbf{I}$ as suggested by von Clarmann et al. (2015). The information on the adequate gridwidth is contained in the averaging kernel of the original retrieval. A new regularization matrix is built which emulates the interpolation scheme assciated with the desired coarse grid representation. The transformation itself is performed using Eq. 10.48 of Rodgers (2000). The immediate outcome of this transformation is still represented on the fine grid, but all values between the coarse gridpoints are fully determined by the interpolation scheme. Thus, it is sufficient to report the values at the coase gridpoints along with the regularization scheme. The new representation is neither considered as generally superior over the old one which involves explicit regularization nor is it meant to supersede it. It is meant only as an alternative representation where time-variant averaging kernels cause problems.

– While averaging kernels of maximum likelihood retrievals are unity on the native grid on which the retrieval has been performed, any interpolation to finer grids will entail non-unity averaging kernels.

– Averaging optimal estimates will not usually create optimal averages. This is particularly true when the prior information is the same for each retrieval, e.g., a climatological data set. This is because the weight of the prior information will be too large in the average (see, e.g., Ceccherini et al., 2014).

– Even if the prior information can be conceived as the frequency distribution of true states, any deviation of the assumed frequency distribution from the true one is an additional error source which is not typically considered in estimated error budgets.

### 6.5  Unknown error components

Error estimation will never be perfect, not only because the input variables of error estimation are uncertain in themselves, but also because there always are error sources that those responsible for the error estimation may not be aware of. Povey and Grainger (2015) propose "to present multiple self-consistent realisations of a data set as a means of depicting unquantified uncertainties." It is obvious that such ensemble techniques are well suited to investigate the non-linear interaction of multiple known error sources, to obtain sensitivity information if the data processors contributing to the ensemble consider different

types of known uncertainties, or to identify the spread of results which may result from different numerical implementations. These authors, however, fall short of telling us how such ensemble techniques should provide information on the effect of unknown error sources. The problem is that none of the data processors contributing to the ensemble has the unknown mechanism implemented, and the unknown uncertainty will cause an unknown bias of the ensemble mean rather than scatter of the ensemble.

The only way known to us to gain confidence that all relevant error sources have been considered is to compare multiple independent measurements based on different measurement systems where we can fairly safely assume that they are not all affected by the same type of systematic effect. If the discrepancies between the results of different instruments can be explained by the combined error budgets, we have reason to believe that the error budgets of the instruments under comparison are fairly complete (e.g., Rodgers and Connor, 2003; von Clarmann, 2006). For at least three independent measurements the random components of the error can be pinpointed quite safely (Laeng and von Clarmann, 2019, and references therein) (Loew et al., 2017, and references therein), and for a large number of independent instruments one can assume that even the bias of the mean will approach zero.

## 6.6 Natural variability

It goes without saying that natural variability in a sense that the atmospheric state at place $s_1$ and time $t_1$ differs from the one at $s_2$ and $t_2$ is not a genuine retrieval error. However, when in a validation context two independent measurements of the same state variable are compared and the measurements do not refer to exactly the same airmass, the spatial or temporal mismatch of the measurements along with natural variability will contribute to the difference. Often, natural variability is invoked as a universal excuse if validation studies hint at unexplained discrepancies. To allow a more quantitative assessment of the role of natural variability in validation, tools to assess the impact of less-than-perfect collocations is provided by, e.g., Sofieva et al. (2008), Verhoelst et al. (2015) or Laeng et al. (2019). The latter tool estimates the difference between two measurements that is explained by natural variability and is based on a parametrization of high-resolution model data. It saves the validation scientist from the need of dedicated model studies for each comparison. Also dense and precise high-resolution measurements can be used as so-called 'fiducial reference measurements'. The latter approach allows simultaneous evaluation of natural variability and validation of error estimates, as discussed in Staten and Reichler (2009) and Sofieva et al. (2014).

## 6.7 Drifts

Instrument drift we understand is a false trend in the derived state variables which is caused by an unstable instrument. At first order, a drift can be avoided if regular and frequent calibration are performed or if the self-calibrating measurement procedures are employed. However, higher order effects, e.g., related to the non-linearity of the calibration curve, can lead to noticeable drifts. Eckert et al. (2014), e.g., found drifts in MIPAS ozone even though regular calibration was performed. This was for the reason that, due to detector aging, the non-linearity of the detector sensitivity changed with time. Also the notorious aging problem of spacebased UV-measurements, degradation in the sense of a reduction in throughput due to intense solar radiation can cause drifts. Most often the cause are the coatings of optical elements.

Whenever *ex ante* drift estimates are available, they should of course be communicated to the data user. Since, however, drifts usually can be determined only reliably towards the end of a mission, it does not make sense to require drift estimates in data characterization papers, which are typically written in the early phase of a mission. A deeper discussion of drifts is found, e.g., in Hubert et al. (2016).

5      Spaceborne UV measurements are typically affected by particularly severe instrumental degradation, i.e., loss of throughput. This is usually caused by optical coatings degrading when exposed to UV radiation. If a tangent altitude normalization approach or another self-calibration approach is used, this degradation is not necessarily a big problem, but the signal to noise ratio will decrease over time.

     The SBUV/2 instruments use an on-board calibration system to track relative spectral and temporal changes in diffuser 10   reflectivity using a mercury lamp (e.g., DeLand et al., 2012). Since the solar diffuser is the only additional optical element between radiance and irradiance measurements, this system enables an accurate throughput change correction to be derived from SBUV/2 solar measurements. This correction is applied in the Level 1 processing and is not included in the error covariances.

     Intrinsically self-calibrating measurement geometries such as solar and stellar occultation or regular calibration measurements using internal sources at first order remove this error. This does, however, not apply to drifts of the shape of the nonlinear 15   detector response function as discussed above. To date, these drifts are not evaluated as part of the routine error analysis of the Level-2 product but they are assessed by careful comparison with other instruments. While it is not easily possible to get absolute drift estimates from this, at least the relative drifts between instruments can be estimated (e.g., Eckert et al., 2014; Laeng et al., 2017; Hubert et al., 2012; Rahpoe et al., 2015; DeLand et al., 2012). It is important to note that relative drifts between instruments may have causes beyond time-dependent calibration changes (e.g., a drift in tangent height registration as 20   shown in Livesey et al. 2018, Bourassa et al. 2018, or Kramarova et al. 2018.).

## 6.8   Combination of error components

Within linear theory, errors of different sources combine additively and follow Gaussian error propagation. We have $k$ covariance matrices of the dimension $n \times n$ representing the errors of $k$ different sources $\mathbf{S}_{x,1}$, $\mathbf{S}_{x,2}$, ... $\mathbf{S}_{x,k}$ and get

$$\mathbf{S}_{x,\text{total}} = \left( \begin{array}{cccc} \mathbf{I}_n & \mathbf{I}_n & \dots & \mathbf{I}_n \end{array} \right) \times \tag{35}$$

$$\times \left( \begin{array}{cccc} \mathbf{S}_1 & \mathbf{C}_{1,2} & \dots & \mathbf{C}_{1,k} \\ \mathbf{C}_{2,1} & \mathbf{S}_2 & \dots & \mathbf{C}_{2,k} \\ \vdots & \vdots & \ddots & \vdots \\ \mathbf{C}_{k,1} & \mathbf{C}_{k,2} & \dots & \mathbf{S}_k \end{array} \right) \left( \begin{array}{c} \mathbf{I}_n \\ \mathbf{I}_n \\ \vdots \\ \mathbf{I}_n \end{array} \right)^T ,$$

where $\mathbf{I}_n$ is an $n \times n$ identity matrix and where the $\mathbf{C}$ matrices represent covariances among the error sources. For independent error sources, these covariances are zero and combination of errors comes down to summing up the error covariance matrices. Beyond linear theory, the interaction of various error sources is best studied by means of ensemble sensitivity studies (see, e.g., Kulawik et al., 2019)

Some data providers publish total error estimates. This practice is also endorsed by Joint Committee for Guides in Metrology (JCGM) (2008a). There are, however, compelling arguments in favour of publishing the individual error components. Specifically, depending on the application of the data, the same type of error can act as random or systematic error. For example, in trend estimation constant biases of the target gas will fully cancel out. Conversely, if, e.g., the total chlorine budget is calculated, the systematic (i.e., time-independent) error components of the parent chlorine species can be fully uncorrelated among the species and thus have to be treated like fully uncorrelated random errors when the error of the total chlorine budget is estimated. In other words, the extent to which error components are 'systematic' is domain-dependent. An error which is systematic in time can be random in the altitude, species, or some other domain. Thus, the data user may be better helped by being given access to the individual error components and some advice on systematicity in the various domains.

## 7  Recommendations

The goal of the TUNER effort has always been to bring the atmospheric remote sensing community together to enable better science. While a great deal of work has been performed over many decades, certain questions about the intercomparability of different data sets continue to linger and can only be answered if the data provided satisfy the conditions of adequacy described in this paper. While TUNER is not the first attempt at achieving this lofty goal (and may not be the last), we believe that the TUNER group is well-suited to this task. With the aim of establishing a consensus on error reporting, the TUNER group is comprised of remote sensing retrieval experts representing instruments with well over a century of combined operational time and experience. Comprising both data providers and data users, the TUNER consortium aims to "practice what they preach" in the hopes that data from past, present, and future instruments may finally be used in a consistent and intercomparable fashion.

Based on the framework and consensus terminology outlined above, and in response to the conditions of adequacy formulated in Section 2, recommendations have been developed on how uncertainties shall be assessed and data characterization shall be reported. These recommendations may seem less specific than the reader might expect, but one-size-fits-all recommendations were found to be inadequate for the variety of instruments under consideration. In the following, we state the general principles that we consider to be useful. Further, we formulate recommendations with respect to the evaluation and reporting of random errors, systematic errors, and further diagnostic data. The respective conditions of adequacy which led to a particular recommendation are listed in brackets (see Section 2). When appropriate, the recommendation is followed by an example or a short discussion in order to elucidate the rationale behind the recommendation.

**R 1.** The language and notation used to describe the error budget must be clearly defined.

This can be accomplished either by explicit definitions of all terms and symbols used or by reference to any available document that lays down a self-consistent terminology. We hope that this paper serves that purpose and that the terminology and notation introduced here will be found useful[8] [CoA 1, CoA 3, CoA 4].

---

[8]In the scientific community, it is often desirable to have a citeable source regarding notation and terminology so as to be consistent. The authors do not want to dictate what language to use and thus do not provide such a recommendation about the notation and terminology in this paper. The decision is left to the reader.

**R 2.** Every effort should be made to make the error budget as complete as possible in the sense that all sizeable sources of uncertainty are included, either via linear mapping, sensitivity studies, or whatever is appropriate for the particular case under assessment.

The choice of which error estimation scheme is adequate depends on the instrument and the specific retrieval scheme. Thus, no 'one-size-fits-all' error estimation scheme is recommended here. The responsibility for judging which treatment of uncertainties is adequate lies with the retrieval scientist, because only they can judge which error sources and error propagation mechanisms are relevant for a particular instrument or data product. An overview of the most commonly used retrieval schemes is given is Sections 4 and 5. Error sources are discussed in Section 6 [CoA 1, CoA 5, CoA 3].

**R 3.** Substantive contributions from each relevant error component should be reported separately

. The reason for this recommendation is that an estimated error component due to one particular error source can be of random characteristic in one application and of systematic characteristic in another application. For example, errors due to uncertain strengths of spectral lines are random if, say, the chlorine budget is calculated from multiple chlorine-containing constituents, each having its own uncertainty due to spectroscopic data. Conversely, in the analysis of a time series of one species the estimated errors due to erroneous line intensities act as a systematic error. The data user is able to consider the relevant error components only if the error contributions are reported separately. If, in addition, the total error is reported, it should include the systematic and the random components. Some error sources can contribute both to the random and the systematic error components [CoA 5, CoA 4].

**R 4.** For each error source, it is often necessary to know if the resulting error components are independent between two subsets of data within a certain domain (time, space, species, etc.).

For example, the error component due to tangent altitude uncertainties can be correlated between different species retrieved from the same measurement. The error component due to spectroscopic data may be correlated in the altitude domain but uncorrelated between different species, etc. We recommend that data providers describe the correlation within each relevant domain either qualitatively or quantitatively, wherever possible. The need of this is illustrated by the example already described under Recommendation R 3. Another example are quasi-systematic errors which are random in the long run only but can be highly correlated on shorter time scales [CoA 5; CoA 3].

**R 5.** When instrument groups make the error components available, they should also indicate which of them contribute primarily to the random error and which contribute primarily to the systematic error.

Classification and combination of errors is most helpful to the data user if it is made by their systematic vs. random nature rather than by origin [CoA 5; CoA 3]. This is important, e.g., in the context of validation. If estimated errors are reported as aggregated parameter errors, and some of them are of systematic nature while the others are of random nature, the data user will not be able to judge which fraction of the bias or the standard deviation of the differences between two measurement systems is explained by the systematic or random error, respectively. On the face of it, this recommendation looks redundant with

Recommendation R 4 applied to the time domain but it is not. Components of the error budget may be strongly autocorrelated in the time domain but still lead to zero bias and thus contribute to the random error only. Again it should be kept in mind that some error sources can contribute both to the random and the systematic error components.

**R 6.** The meaning of the reported uncertainties shall be clarified.

Do they refer to $\pm 1\sigma$, $\pm 2\sigma$, etc. or to a specified confidence limit, such as 95% or 99%? Note that generalized Gaussian error propagation will usually produce error estimates in terms of variances, while Monte-Carlo-type sensitivity studies enable the confidence limits to be directly estimated. If the one is transformed to the other, the assumed underlying distribution shall be reported [CoA 1; CoA 4].

**R 7.** For all error components, the assumed ingoing uncertainties shall be reported in the relevant documentation. It should also be reported which correlation characteristics were assumed (e.g., scalar perturbation of a profile, individual perturbation of its elements, or consideration of its full covariance matrix)

Without reporting assumptions on ingoing errors, error propagation would not be traceable. With this information at hand, a data user can re-scale error estimates if there is some doubt about the assumption on ingoing uncertainties (e.g., the $\Delta b$ in Eq. 23 or $\mathbf{S}_b$ in Eq. 24) or if in a comparison study the error estimates of one instrument are more optimistic or pessimistic than those of the other[CoA 4].

**R 8.** If the retrieval uses prior information in the sense of Eq. 4 or Eq. 6, the a priori profiles must be reported.

This allows the data user to apply Eq. (31) or variants of it. Also for column retrievals where an a priori profile is scaled to obtain the best fit and then integrated over altitude to render the column, the a priori profile should be reported, because the column can depend on the assumed profile shape[CoA 5, CoA 4].

**R 9.** In addition to the error budget, averaging kernels (Eq. 25) should be reported. The averaging kernels should be based on a discretization fine enough that all expected relevant variation can be represented. For retrieval of vertical columns the column averaging kernels (Eq. 27) are the respective diagnostic.

If a certain retrieval scheme does not give direct access to averaging kernels (e.g., onion peeling) then averaging kernels shall be determined by sensitivity studies based on perturbations of the profile. For retrieval approaches using truncated singular value decomposition or related approaches, the final altitude resolution shall be expressed as averaging kernels. For global fit maximum likelihood retrievals (no regularization) the averaging kernels are by definition unity, but only in the native retrieval grid. In such cases, regridding of data will give rise to non-unity averaging kernels. At the very least, the original grid and the interpolation scheme shall be reported. The data provider should calculate the averaging kernels on the final grid on which the data are provided to the user. To avoid any misinterpretation of the averaging kernel and taking the averaging kernel matrix for its transpose, it should be indicated which index refers to the columns and which to the rows of the averaging kernel matrix [CoA 1, CoA 5, CoA 2, CoA 3].

**R 10.** The space to which the averaging kernel applies (e.g., linear/logarithmic, mixing ratio/density, absolute/relative, etc.) shall be reported.

This is particularly important when data are reported in a form that differs from that of the retrieval state vector. E.g., the averaging kernels resulting from a retrieval of the logarithms of mixing ratios must not be applied to the mixing ratios themselves. It is thus of utmost importance to communicate to the data user to which quantities the averaging kernels refer. If the averaging kernel made available to the data user underwent some transformation, the user should also be informed in which space the averaging kernel was initially calculated [CoA 1, CoA 5, CoA 3, CoA 4].

**R 11.** The smoothing error should not be included in the error budget. Instead the data users should be provided with the averaging kernel matrices calculated on a sufficiently fine grid allowing them to evaluate the smoothing error on their working grid to which they will transform the data.

Error propagation of the smoothing error in the context of interpolation to finer grids will usually fail to produce the full smoothing error on the fine grid (von Clarmann, 2014). Separate reporting of the smoothing error allows the data user to propagate all errors except the smoothing error through the interpolation and to evaluate the smoothing error directly on the fine grid, if desired. A caveat on the peculiarities related to the interpolation of the smoothing error is adequate if the smoothing error is reported. If the data are understood to be a representation of the smoothed state of the atmosphere (Section 6.4.2), the smoothing error is not needed and averaging kernels along with the prior information are sufficient. For example, in data assimilation, where the averaging kernel is part of the observation operator, inclusion of the smoothing error into the observation error covariance matrix would be redundant and lead to incorrect double counting of the smoothing effect[CoA 2].

**R 12.** The discretization must be specified. If the retrieval is reported as state value on any vertical grid, the applicable interpolation rule must be reported. If an altitude-resolved retrieval is performed in any other space than state value over altitude, pressure, or likewise (e.g., if eigenvectors or similar are used, see Section 5.1), then the final result should in addition be presented as vertical profiles and also all diagnostic data (error estimates, averaging kernels) should be transformed to the respective representation

While these alternative representations certainly have their advantages, the data producer is in a better position than the data user to provide the diagnostic data for a profile representation [CoA 5, CoA 2, CoA 3].

**R 13.** Retrieval scientists should judge whether evaluation of error budgets and averaging kernels for a limited number of representative cases is adequate. If averaging kernels are only provided for a few representative cases, one might still consider to show at least the vertical resolution profiles for each profile

Communication of a complete error budget for each profile, broken down to all components with all correlation information, along with averaging kernels and a priori information used, is not always technically feasible and often creates unnecessary data traffic. [CoA 6].

The following recommendations R 14–17 are applicable to the case when only representative diagnostic data are available.

In this context we would like to mention that there exist methods to convey the information content of a measurement at drastically reduced data volume (Migliorini et al., 2008). Such methods are particularly convenient in the context of data assimilation.

**R 14.** If representative error estimates are reported instead of error estimates for each single profile or data point, it is of utmost importance to tell the data user if the nature of each error component is chiefly additive (i.e., independent of the actual state value reported) or is chiefly relative (i.e., a scaling factor). For the first type, the estimated errors shall be reported in the same units as the state variable (e.g., Kelvin, ppmv, molec./cm$^3$); for the second type, estimated errors shall be reported as percentage errors.

With this information, the data user can adjust the error estimates to the particular scientific study. For example, measurement noise often leads to an additive error component, i.e., the estimated error is approximately of the same size, regardless how large the mixing ratio of the target gas is. Conversely, errors representing spectroscopic uncertainties are often multiplicative. That is to say, larger profile values have larger errors [CoA 3, CoA 6].

**R 15.** If certain estimated errors or other characterization data are known or suspected to depend systematically on time, latitude, or other parameters, this dependence should be reported, particularly if only representative errors are reported.

For example, in infrared emission spectroscopy the precision of concentration retrievals is usually worse for a colder atmosphere. With this information a data user who is using a retrieval of a particular cold day which is not well represented by the sample error estimates is warned that the actual precision may be worse than the reported one[CoA 3].

**R 16.** If, for application to mean profiles, mean averaging kernels are provided in conjunction with mean profiles instead of individual ones, then the correlation profiles between the averaging kernels and the retrieved profiles shall be provided

The reason is that the mean averaging kernel applied to the mean profile does not equal the mean of individual averaging kernels applied to individual profiles (von Clarmann and Glatthor, 2019). With the correlation profiles available, this difference can be corrected [CoA 6].

**R 17.** If, in order to reduce the data volume of profile data characterization, only standard deviations are reported for the individual profiles instead of the full covariance matrices, then a representative random error correlation pattern in the altitude domain (correlation matrix) shall be made available.

With this, the user can approximate individual covariance matrices [CoA 5, CoA 5, CoA 3, CoA 6].

**R 18.** The error estimates should explain observed differences between measurements of the same airmass.

The final criterion of adequacy of error reporting is whether discrepancies between measurements of the same atmospheric state variable by independent measurement systems can be explained by the error estimates. This practical and empirical criterion of completeness of the error budget does not require knowledge of the unknowable true value of the measurand [CoA 1, CoA 5,]. In this context we distinguish between random and systematic errors.

1. We consider random error estimation schemes as adequate if a combination of the deduced error and the less-than-perfect spatial or temporal coincidences between two data sets and natural variability together explain the observed standard deviation of the differences between two data sets. If predicted random errors fail to explain observed differences, they should be reassessed. Methods to find out which of the compared data sets has an inadequate random error estimate have been described in, e.g., Fioletov et al. (2006); Sofieva et al. (2014); Laeng and von Clarmann (2019) [CoA 5].

2. We consider estimates of the systematic errors to be adequate if they, along with sampling biases and after accounting for different vertical/horizontal/temporal resolutions and content of a priori information, explain the observed biases between independent instruments [CoA 5].

We consider it undesirable and a source of confusion to still report over-optimistic or pessimistic *ex-ante* error budgets without a related *caveat* if validation studies show that there is strong indication that the actual errors are significantly larger or smaller.

On the face of it, the list of recommendations appears quite weak, leaving a lot of freedom to the data provider. This is, however, not the case. Recommendation R 2, that the error budget should be as complete as possible, along with Recommendation R 18, which gives a criterion for the completeness of the error budget quickly make the apparent freedom disappear.

Admittedly, these recommendations will not guarantee perfect compliance with the conditions of adequacy, but due to the competing needs of rigor versus practicability the problem seems overconstrained. In other words, you 'can't always get what you want' (Jagger and Richards, 1969). However, we are still confident that they help to unify uncertainty reporting in the community of remote sensing of atmospheric composition and temperature. These recommendations have been developed from the perspective of mainly satellite-borne limb sounding and occultation observations but some of these concepts are equally applicable to other types of remote sensing missions.

## 8   Discussion and Outlook

In this paper we have discussed conventional (as opposed to machine learning and artificial-intelligence based approaches) error estimation methods for Bayesian and non-Bayesian retrieval methods. The choice of the retrieval method is a dilemma. If likelihood-based methods are chosen, the retrieval lacks a probabilistic interpretation and *ad hoc* constraints will imply a bias, at least if the retrieval is conceived as a smoothed estimate of the true state. This horn of the dilemma is avoided by Bayesian methods, which use probabilistic constraints. Adherents of likelihood-based methods, however, will point out the second horn of the dilemma, which is, that it is never warranted that the a priori statistics chosen indeed represents the true background state. Further, they will raise the concern that Bayesian methods, even if based on the true background statistics, may render bias-free estimates in the long run, but may be off the true atmospheric state in a single case. The decision for the acceptance of the one or the other horn of the dilemma is a philosophical one and in most cases it cannot be based on scientific grounds. The only recommendation we can offer in this respect is a plea for mutual tolerance. Regardless which approach is chosen, the data characterization has to be consistent with the retrieval method chosen. This paper tries to provide the scientific basis for this.

This paper is mainly addressed to providers of Level-2 data, i.e., data on atmospheric state variables. Some data users, however, prefer to work directly with Level-1 data, i.e., with measured radiances or transmissions. For example, the direct data assimilation of measured signals is sometimes preferred over the assimilation of retrieved state variables (e.g., Andersson et al., 1994). The radiative transfer forward model is in this case included in the assimilation scheme. The advantage of this method is that it avoids all problems related to a priori knowledge and regularization. We have touched this approach only upon passing in this paper and do not want do delve deeper into this. The only caveat we wish to add is that the observation error covariance matrix should not include measurement noise only but also contributions by uncertain parameters not assimilated (Section 5.2). This will typically lead to non-sparse observation error covariance matrices which may be the source of some further headache.

In some fields of remote sensing of the atmosphere, retrieval methods based on artificial intelligence, neural networks and machine learning are explored (Lary et al., 2016). A precondition for unification of error reporting of classical and artificial-intelligence based retrieval schemes seems to be semantic connectibility. The glossary by Stanford University (2020) is considered as an important first step. With respect to the data characterization of retrieval products generated with such algorithms, two cases have to be distinguished. The first case is that a neural network is used as a surrogate radiative transfer forward model, while the retrieval still follows the concepts presented in this paper. In this case, the error estimation and data characterization strategies discussed in Section 6 are still applicable, and the approximative nature of the neural-network based radiative transfer calculation can simply be conceived as a further source of forward modelling error. The second case is that machine learning algorithms are directly used for the retrieval. In this case, complete data characterization appears to be more challenging to us. Sensitivity studies or supervised learning of uncertainty prediction may be two possible pathways towards data characterization of artificial-intelligence based retrievals. In either case it seems important to us that the data user is provided with the same full data characterization as required for the conventional retrieval schemes.

But even with the conventional retrieval and error estimation schemes there is a lot of homework to do. We hope that this review paper has identified the most relevant problems in this field and provides a conceptual framework to adequately characterize remotely sensed atmospheric temperature and composition data.

Table 1: Satellite Data Processors, Limb Geometry (Emission, Occultation and Scattering):

| Instrument | Processor | Geometric Decomposition | Regularization[1] $\mathbf{R}$ | Reference |
|---|---|---|---|---|
| ACE-FTS | Version 3.6/3.6 | global fit | 0 | Boone et al. (2005) |
| | | | | Boone et al. (2013) |
| ATMOS | Version1-2 | onion peeling | 0 | Norton and Rinsland (1991) |
| | Version 3 | global fit | 0 | Irion et al. (2002) |
| CLAES | Chahine | Chahine's relaxation | | Kumer and Mergenthaler (1991) |
| | Onion Peeling | onion peeling | $\mathbf{S}_a^{-1}$ | Gordley and Marshall (1992) |
| CRISTA | UWuppertal | onion peeling | 0 | Offermann et al. (1999) |
| GOMOS | ALGOM2s (FMI) | other | $\mathbf{L}_2^T \gamma \mathbf{L}_2$ | Kyrölä et al. (2010) |
| | | | | Sofieva et al. (2010) |
| | | | | Sofieva et al. (2017) |
| HALOE | NASA | onion peeling | 0 | Thompson and Gordley (2009) |
| HIRDLS | HIRDLS | global fit | $\mathbf{S}_a^{-1}$ | Gille et al. (2008) |
| ILAS | Version 1 | onion-peeling | 0 | Yokota et al. (2002) |
| ILAS-2 | Version 1.4 | onion-peeling | 0 | Nakajima (2006) |
| ISAMS | | onion-peeling | $\mathbf{S}_a^{-1}$ | Marks and Rodgers (1993) |
| | | sequential estimation[2] | $\mathbf{S}_a^{-1}$ | Dudhia and Livesey (1996) |
| LIMS | NASA | onion peeling | Twomey (1977) | Gille and Russell III (1984) |
| | | | | Bailey et al. (1996) |
| MIPAS | UBologna | geofit | $\mathbf{S}_a^{-1}$ | Dinelli et al. (2010) |
| | ESA | global fit | $\mathbf{S}_a^{-1}$ | Raspollini et al. (2013) |
| | IMK/IAA | global fit[3] | $\mathbf{L}_1^T \gamma \mathbf{L}_1$ | von Clarmann et al. (2003, 2009) |
| | UOxford | global fit[4] | $\mathbf{S}_a^{-1}$ | Dudhia (2019) |
| MLS-AURA | NASA | geofit[5] | $\mathbf{L}_1^T \gamma \mathbf{L}_1$ | Livesey et al. (2003) |
| MLS-UARS | NASA | global fit | $\mathbf{S}_a^{-1}$ | Livesey et al. (2003) |
| Odin-SMR | Chalmers | global fit | $\mathbf{S}_a^{-1}$ | Urban et al. (2005) |
| OMPS-LP | IUP Bremen | global fit | $\mathbf{L}_0^T \gamma_1 \mathbf{L}_0 + \mathbf{L}_1^T \gamma_2 \mathbf{L}_1$ | Arosio et al. (2018) |
| | NASA | global fit | $\mathbf{S}_a^{-1}$ | Rault et al. (2013) |
| | USask | geofit | $\mathbf{L}_2^T \gamma \mathbf{L}_2$ | Zawada et al. (2018) |
| OSIRIS | Version 5 | global fit | 0 | Degenstein et al. (2009) |

Table 1: Satellite Data Processors, Limb Geometry (continued:

| Instrument | Processor | Geometric Decomposition | Regularization[1] $\mathbf{R}$ | Reference |
|---|---|---|---|---|
| | | | | Bourassa et al. (2012) |
| SABER | NASA | onion peeling interleave | 0 | Russell III et al. (1999) Rong et al. (2009) |
| SAGE I | NASA | Chahine[6] | 0 | Chu and McCormick (1979) |
| SAGE II | NASA | global fit[7] | 0 | Damadeo et al. (2013) |
| SAGE III | NASA | global fit | 0 | Wofsy et al. (2002) |
| SAMS | | sequential estimation | $\mathbf{S}_a^{-1}$ | Rodgers et al. (1984) |
| SCIAMACHY | IMK | global fit | $\mathbf{S}_a^{-1} + \mathbf{L}_1^T \gamma \mathbf{L}_1$ | Bender et al. (2013) Bender et al. (2017) |
| | IUP Bremen (limb scatter) | global fit | $\mathbf{L}_0^T \gamma_1 \mathbf{L}_0 + \mathbf{L}_1^T \gamma_2 \mathbf{L}_1$ | Rozanov et al. (2011b) Rozanov et al. (2011a) |
| | IUP Bremen (occultation) | onion peeling | 0 | Noël et al. (2016) Noël et al. (2018) |
| | | global fit | $\mathbf{L}_0^T \gamma_1 \mathbf{L}_0 + \mathbf{L}_1^T \gamma_2 \mathbf{L}_1$ | Azam et al. (2012) |
| SMILES | JAXA | global fit | $\mathbf{S}_a^{-1}$ | Takahashi et al. (2010) |
| | NICT | global fit | $\mathbf{S}_a^{-1}$ | Baron et al. (2011) |
| SOFIE | GATS | onion peeling interleave | 0 | Gordley et al. (2009) |

(1) For processors with multiple data products, the actual regularization may vary depending on the retrieved atmospheric parameter.

(2) sequential estimation using a Kalman filter

(3) under consideration of horizontal gradients

5  (4) sequential estimation in the spectral domain

(5) subsets of orbits are used.

(6) SAGE team's best guess as original documentation was lost.

(7) onion peeling for $H_2O$.

Table 2: Satellite Data Processors: Nadir sounders

| Instrument | Processor | Column or Profile Retrieval | Regularization[1] $\mathbf{R}$ | Reference |
|---|---|---|---|---|
| AIRS | NASA v6 | profiles | PCA | Susskind et al. (2014) |
| | NUCAPS | profiles | PCA | Susskind et al. (2003) |
| | CLIMCAPS | profiles | $\mathbf{S}_a^{-1}$ | Smith and Barnet (2019) |
| | MUSES | profiles | $\mathbf{L}_1^T \gamma_1 \mathbf{L}_1 /$ $\mathbf{L}_0^T \gamma_2 \mathbf{L}_0$ or $\mathbf{S}_a^{-1}$ | Fu et al. (2018) |
| CrIS | NUCAPS | profiles | PCA | Susskind et al. (2003) |
| | CLIMCAPS | profiles | $\mathbf{S}_a^{-1}$ | Smith and Barnet (2019) |
| | CFPR | profiles | $\mathbf{S}_a^{-1}$ | Shephard and Cady-Pereira (2015) |
| | MUSES | profiles | $\mathbf{L}_1^T \gamma_1 \mathbf{L}_1 /$ $\mathbf{L}_0^T \gamma_2 \mathbf{L}_0$ or $\mathbf{S}_a^{-1}$ | Fu et al. (2016) |
| GOME | AMC-DOAS | columns | 0 | Noël et al. (1999) |
| | GODFIT(BIRA) | columns | 0 | Lerot et al. (2010) |
| | DOAS | columns | 0 | Richter et al. (1998) |
| | OPERA (KNMI) | profiles | $\mathbf{S}_a^{-1}$ | van Peet et al. (2014) |
| | RAL | profiles | $\mathbf{S}_a^{-1}$ | Siddans (2003) |
| | WFDOAS | columns | 0 | Weber et al. (2018) |
| GOME-2 | AMC-DOAS | columns | 0 | Noël et al. (2008) |
| | GODFIT(BIRA) | columns | 0 | Lerot et al. (2010) |
| | DOAS | columns | 0 | Vrekoussis et al. (2009) |
| | | | | Boersma et al. (2018) |
| | OPERA (KNMI) | profiles | $\mathbf{S}_a^{-1}$ | van Peet et al. (2014) |
| | RAL | profiles | $\mathbf{S}_a^{-1}$ | Siddans (2003) |
| GOSAT | ACOS | $XCO_2$ | $\mathbf{S}_a^{-1}$ | O'Dell et al. (2012) |
| | BESD | $XCO_2$ | $\mathbf{S}_a^{-1}$ | Heymann et al. (2015) |
| | NIES | columns/profiles | $\mathbf{S}_a^{-1}$ | Yoshida et al. (2013) |
| | RemoTeC | profiles | $\mathbf{L}_1^T \gamma_1 \mathbf{L}_1 /$ $\mathbf{L}_0^T \gamma_2 \mathbf{L}_0$ | Butz et al. (2011) |
| | UOL | $XCO_2$ | $\mathbf{S}_a^{-1}$ | Cogan et al. (2012) |
| IASI | ASIMUT-ALVL (BIRA) | profiles | $\mathbf{S}_a^{-1}$ | De Wachter et al. (2017) |

Table 2: Satellite Data Processors: Nadir sounders (continued)

| Instrument | Processor | Column or Profile Retrieval | Regularization[1] $\mathbf{R}$ | Reference |
|---|---|---|---|---|
| | EUMETSAT | profiles | $\mathbf{S}_a^{-1}$ | August et al. (2012) |
| | FORLI(ULB/LATMOS) | profiles | $\mathbf{S}_a^{-1}$ | Hurtmans et al. (2012) |
| | MUSICA (IMK) | profiles | $\mathbf{S}_a^{-1}$ | Borger et al. (2018) |
| | RAL | profiles | $\mathbf{S}_a^{-1}$ | Siddans et al. (2017) |
| | $\delta$-IASI | profiles | $\mathbf{S}_a^{-1}$ | Liuzzi et al. (2016) |
| MOPITT | V8 | profiles | $\mathbf{S}_a^{-1}$ | Deeter et al. (2019) |
| OCO-2 | FOCAL | $CO_2$ profiles | $\mathbf{S}_a^{-1}$ | Reuter et al. (2017b, a) |
| | NASA/V9 | $XCO_2$ | $\mathbf{S}_a^{-1}$ | O'Dell et al. (2018) |
| | UOL | $XCO_2$ | $\mathbf{S}_a^{-1}$ | Boesch et al. (2011) |
| | RemTeC | profiles | $\mathbf{L}_1^T \gamma \mathbf{L}_1$ | Wu et al. (2018) |
| OMI | DOAS | columns | 0 | Theys et al. (2015) |
| | | | | Boersma et al. (2018) |
| | GODFIT (BIRA) | columns | 0 | Lerot et al. (2010) |
| | Harvard-SAO V3 | columns | 0 | González Abad et al. (2015) |
| | MUSES | profiles | $\mathbf{L}_1^T \gamma_1 \mathbf{L}_1 / \mathbf{L}_0^T \gamma_2 \mathbf{L}_0$ or $\mathbf{S}_a^{-1}$ | Fu et al. (2018) |
| | NASA-GSFC | columns | 0 | Bhartia and Wellemeyer (2002) |
| | OPERA (KNMI) | profiles | $\mathbf{S}_a^{-1}$ | van Peet et al. (2014) |
| | RAL | profiles | $\mathbf{S}_a^{-1}$ | Siddans (2003) |
| OMPS-NM | NASA-GSFC | columns | 0 | Bhartia and Wellemeyer (2002) |
| OMPS-NP | NASA-GSFC | profiles | $\mathbf{S}_a^{-1}$ | Bhartia et al. (2013) |
| SBUV | NASA | profiles | $\mathbf{S}_a^{-1}$ | Bhartia et al. (2013) |
| SCIAMACHY | AMC-DOAS | columns | 0 | Noël et al. (2004) |
| | BESD | profiles | $\mathbf{S}_a^{-1}$ | Reuter et al. (2010) |
| | | | | Reuter et al. (2011) |
| | DOAS | columns | 0 | Afe et al. (2004) |
| | | | | Boersma et al. (2018) |
| | OPERA (KNMI) | profiles | $\mathbf{S}_a^{-1}$ | van Peet et al. (2014) |
| | RAL | $O_3$ profiles | $\mathbf{S}_a^{-1}$ | Siddans (2003) |
| | WMF-DOAS | columns | 0 | Schneising et al. (2012) |

Table 2: Satellite Data Processors: Nadir sounders (continued)

| Instrument | Processor | Column or Profile Retrieval | Regularization[1] $\mathbf{R}$ | Reference |
|---|---|---|---|---|
| TES | v7 | profiles | $\mathbf{L}_1^T \gamma_1 \mathbf{L}_1/$ $\mathbf{L}_2^T \gamma_2 \mathbf{L}_2$ or $\mathbf{S}_a^{-1}$ | Bowman et al. (2006) |
| TROPOMI | DOAS | columns | 0 | Theys et al. (2017) |
| | GODFIT (BIRA) | columns | 0 | Lerot et al. (2010) |
| | RemoTeC | profiles | $\mathbf{L}_1^T \gamma_1 \mathbf{L}_1/$ $\mathbf{L}_0^T \gamma_2 \mathbf{L}_0$ | Hu et al. (2018) |
| | SICOR | columns | 0 | Borsdorff et al. (2018) |
| | WMF-DOAS | columns | 0 | Schneising et al. (2019) Weber et al. (2018) |

(1) For processors with multiple data products, the actual regularization may vary depending on the retrieved atmospheric parameter, and whether it is a column or profile.

*Code and data availability.* N/A

*Author contributions.* TvC, DAD and NJL organized the project. TvC, NJL and RD wrote major parts of the text. All authors contributed to the discussion of the paper and particularly the recommendations

*Competing interests.* TvC, CvS and GPS are associate editors of AMT but have not been involved in the evaluation of this paper.

5  *Acknowledgements.* The World Meteorological Organization (WMO) has provided travel support through the Stratosphere-troposphere Processes And their Role in Climate (SPARC) project who have selected the TUNER project as a SPARC activity. The International Space Sciene Institute (ISSI) has funded two International Team meetings in Berne at their venue. Part of this work was carried out at the Jet Propulsion Laboratory, California Institute of Technology, under contract with NASA. The authors would like to thank Andreas Richter for providing information on UV-VIS nadir sounding products. One of the reviewers stood out as particularly knowledgeable, thorough, thoughtful and
10  constructive. The authors highly appreciate their suggestions.

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
