# Peer review of "Estimating and Reporting Uncertainties in Remotely Sensed Atmospheric Composition and Temperature"

_Atmospheric Measurement Techniques, 2019_

## Referee Comment (RC1) · Anonymous Referee #1 · 3 Dec 2019

In their paper Thomas von Clarmann and co-authors provide a list of recommendations on how to report on errors, based on the activities of the TUNER project. To my opinion this is a very important and timely topic, and I acknowledge the effort made by the author team to write a dedicated paper discussing this point. I found the paper interesting, but also had two major reservations and a substantial number of comments, as detailed below, which require a major revision of the paper before publication.

Major general comments:

My first major reservation: the purpose of the paper is the formulation of the list of recommendations for a more uniform and complete error reporting in level-2 satellite

data products (see the last line of the abstract and section 7). However, the bulk of the material presented in the paper is basically a review of real-world implementations of optimal-estimation based (and related) profile retrievals. As such the authors could consider to split the document into two papers, a review of profile retrievals and a shorter more focussed paper about unified error reporting.

Several sections of the paper are providing useful functional background information for section 7. But for quite some sections I could not find the link with the final recommendations. Examples are section 5.4 and also parts of 4, 5.1 and 6.4 (e.g. 6.4.3 to 6.4.6), 6.7. Because of these sections the paper is very long.

Section 5.4 is a review of (profile) retrieval approaches, but contains a lot of material which is not directly relevant for the paper. Personally, I would propose to shorten this section, keeping possiby the tables (and references) and keeping those remarks which are important in the context of error reporting. This review-like section also leads to a very long list of references. It would be good to mention only those references that bring new information to the discussion how to present the retrieval errors.

In general, there is quite a big conceptual step (gap) between sections 3-6 and the summarising recommendations in section 7. Ideally all recommendations should be complemented by motivations, examples and explanatory information in the sections preceding section 7. The link between the recommendations and the rest of the text (which reads as a review of retrieval methods and theory) is often unclear to me. I would suggest that the authors go through sections 3 to 6 (see list of subsections mentioned above) and remove discussions which are not functional to motivate the requirements presented in section 7.

After reading the first sections of the paper it was not fully clear to me what is really the problem which is addressed? In what sense are retrieval products not comparable? Please provide (generic) examples of retrieval products which miss information which makes a direct comparison between retrievals, or comparisons with independent data

difficult or impossible. In what sense is there a need for a new set of recommendations, e.g. what is missing after the work of e.g. QA4EO or the GUM?

The final set of recommendations are focussing on profile retrievals. But the tables include also total column retrieval examples (e.g. DOAS). I think this is a missed opportunity, and I would encourage the authors to formulate explicitly what their recommendations are for column retrieval products (some recommendations are generic, but several parts of section 7 explicitly refer to profiles).

Arguably the atmospheric composition data assimilation community is the main user of satellite retrieval products. This community and their needs are basically not discussed in the paper. More generally the users of the data do not receive much attention, and the requirements are discussed from a L2 data provider point of view. This is my second major reservation. Some parts of the text refer to the validation activities, but this is not presented in a very structured way. The needs and feedback from the validation and assimilation communities on existing L2 satellite products would be an important starting point to discuss requirements for satellite data products. Some assimilation users would prefer to work directly with the level-1b data, an option which is also not discussed.

The recommendations in section 7 are not always formulated as a recommendation, but leave room for interpretation and implementation. I sometimes found the CoA points in section 2 even more clear and explicit than the recommendation points. It may be useful to split the list in section 7 in actual (strong) recommendations and related discussion points. Sometimes it is not so clear what is actually recommended by the authors, e.g. due to a trade-off between completeness and data volume, or aspects are left to the retrieval teams to decide (e.g. point 1, 2, 3, 4, 16, 18).

I was expecting recommendations also regarding the naming (see section 3). The authors discuss in particular "error" versus "uncertainty", but do not really provide a clear guidance on what to use. Also, the consistency or inconsistency with the GUM

activity are not clear to me after reading the section. The reader is referred to a paper in preparation. Retrieval datafiles contain parameters labelled as "precision", "accuracy", "trueness" etc. and different guidelines exist from different space agencies and for different application areas. It would be useful if the authors can discuss naming conventions also in this paper and express a clear opinion/recommendation.

Machine learning approaches are getting more and more popular and deserve some special attention. Several machine learning implementations for retrievals are limited on the error information they provide. It would be useful to have some targeted recommendations for these approaches as well.

Detailed comments:

Abstract: The abstract reads like an introduction. I would encourage the authors to summarise (shorten) the first part and expand on the last sentence with a summary of the content and main results of the paper.

Introduction

l6: "reduction"? Should this be "deduction"?

l16: "The project ... is a consortium of". Please modify

l24: "atmospheric composition and temperature profiles". What about other profiles, e.g. water vapour? Is the paper limited to profiles, or are single property (column) retrievals also included?

l37: "are do not need to be", please correct.

Section 2:

l82, CoA 1: "and/or error estimation schemes". Would it not be better to say "and/or retrieval schemes"?

CoA 2: "independent of the vertical grid". But I assume at this point that error covariances are specified on a specific grid used in the retrieval ?!

CoA 5: "and different amounts of prior information". Do you mean "and different sources of prior information"?

p3, l10: "but we consider it unrealistic to assign quality indicators for 'fitness for purpose' for all conceivable applications." This is an interesting remark. It would be useful to expand on this: explain how it is discussed by QA4EO and which parts are unrealistic.

Sec 3.1: please introduce the acronym "Joint Committee for Guides in Metrology (JCGM)" just once, and use only the acronym "JCGM" in the rest of the paper.

sec 3.1, l34: "actually claimed that there are conceptual differences between error analysis and un- certainty estimation." For readers who did not follow this debate it is hard to follow this section. It would be helpful to add a few sentences to list the claimed conceptual differences between these two terms.

Section 4: I find it useful to include a section with the theoretical background and notation. In fact, using this notation could be a recommendation (Section 7, point 1).

eq. 2: "can only be approximated" What does this refer to? The ill-posed or underdetermined nature of many inverse problems?

l70: "macrorcopic"

l77: What is the approximation which turns "f" into "F". Are these real-life uncertainties in f? Is F now a matrix or still a non-linear function?

l87: "overdetermined case (m > n)". Whether or not the inverse problem is overdetermined also depends on F, and not only on the size of the vectors. Add "and not ill-posed". (This is discussed on next page

p5, l7: "In most real-world applications, only measurement noise is considered here, while other measurement uncertainties like calibration errors are neglected at this

stage." Remove "here" and "at this stage".

p5, l21-44: This is an interesting historical note, but not essential for this paper and may be removed.

eq. 5: What is L1? What are its properties?

Sec.5:

l22: mention the loss of information

Sec 5.4. This section is basically a review of retrieval approaches: why is it relevant for this paper to include such a review? See my general remark above.

Sec 6, p10, point 2: Model errors: It would seem logical to me to split this into RT model errors and inputs used by the forward and inverse models, e.g. influence of atmospheric aspects like surface characterisation, aerosols and clouds, other meteorological variables (humidity, temperature).

Sec 6, p10, point 3: "errors caused by decomposing the inverse problem". Does this deserve a separate section?

Sec 6.1.1, l37: "cheerful" ...

Sec 6.1.3, l33: "measurments"

Sec 6.2.1: "If a complete model is available but not used .., the effect of the missing processes can be assessed via sensitivity analyses based on the complete model ...". This sounds like a recommendation (could be part of section 7).

p15, l3-7: "The OCO-2 team is currently working on ..". I could not understand this paragraph. I suggest to either explain the approach in more detail or omit.

l24: "retrived"

p16, l40: "the derivative". I do not understand how to take such a derivative.

Sec 6.3: The parameter errors are often very relevant and could be discussed more extensively. For these parameters often simplifying assumptions are made (e.g. climatologies) or they are taken from elsewhere (e.g. actual weather model output) or they may be derived in the retrieval itself (or previous step in the retrieval). All these choices will lead to different characteristics for the related errors, often introducing quasi-systematic error correlations.

6.3: Why is this section called "parameter errors" instead of something like "Inverse model decomposition errors"

Sec 6.4.1 and 6.4.2: I'm happy that the authors include these two "interpretations". This is a subtle point, often not understood by satellite data users.

p16, l87: "the undesirable effect that a smoothing error evaluated on a coarse grid will be smaller than a smoothing error evaluated on a fine grid." I do not really understand why this is undesirable. This property seems to make sense to me: more layers allow more detail to be resolved (and smoothed away by the retrieval process).

p18, l18: "also commonly applied when measurements are compared to model data". It would be good to mention explicitly the data assimilation application here.

p19, l10: "reasons reasons"

Sec 6.4.3: I was wondering if this section (on altitude resolution) is needed as background to section 7.

Section 7

Point 2: A bit weak, it leaves a lot of room for different approaches.

Point 3: Does this have repercussions for the data volume? Especially when each component has its own covariance matrix?

Point 4: Again leaves much freedom. What about proposing a 1 sigma as default?

Point 6: "error components available, they should also indicate how they contribute to the random and/or systematic error" What about the total error: should this consist of a random and a systematic part? What does "indicate" mean in practice? Please be more specific.

Point 7: It is difficult to understand what is meant here. What is the domain of a subset of a component of a source of error? It would be good to provide an example. What is the difference between an error source and an error component?

Point 9: "assumed ingoing uncertainties shall be reported". What is meant by "reported"? Does this refer to the ATBD, a journal paper or to the L2 datafiles themselves?

Point 10: Sometimes $(I−A)x\_a = 0$ even though the retrieval still needs/depends on a-priori information. Should the a-priori be reported also in this case?

Point 10: What are "similar operations"? Please be more explicit.

Point 11: I do not understand why it is crucial to have the results as vertical profiles (as opposed to desirable). The vertical profile retrievals are linked to the real physical world through the averaging kernels, as specified in Eq. 30. Ignoring this link leads to all the smoothing error considerations (and problems) as discussed extensively by the authors. Especially when the kernel is very different from the unity matrix I, the interpretation of the retrieved profile as a real profile becomes troublesome. The retrieval at a given altitude then contains physical information from (depends on concentrations in) many other layers, as specified by the averaging kernel matrix. The kernels will always have altitude on one axis, even if presented in eigenvalue space, and relate the retrieval to real physical profiles. Please explain why this strong statement ("should be presented") is made.

Point 12: "Ideally the data provider calculates the averaging kernels on the final grid". What is proposed here? It sounds like a commitment of the retrieval team (data provider) to provide support to all users with a grid which differs from the retrieval grid.

This would imply a major commitment. Or would this imply that each retrieval product should be accompanied by software to do the interpolation (extrapolation is also very likely!) to different grids.

Point 13: "This is particularly important when data are reported in a form that differs from that of the retrieval state vector". This may not be very clear to the reader. Please provide an example. Why is it important in this case?

Point 15: "If the data are understood to be a representation of the smoothed state of the atmosphere, the smoothing error is not needed and averaging kernels along with the prior information are sufficient". I suggest that the authors explicitly mention applications here, e.g. model-satellite comparisons and data assimilation.

Point 16: "Communication of a complete error budget ... is not always technically feasible and often creates unnecessary data traffic." I would suggest that the authors include a reference to the work of S. Migliorini, DOI: 10.1175/2007MWR2236.1. This paper describes how the data volume can be reduced drastically (explicit a-priori profile and error covariance no longer needed) while preserving the full error information, to support data assimilation applications. Do the authors consider this a possible alternative for storing the retrieved profiles, see e.g. point 10, 11?

Point 18: This important point distinguishes random and systematic errors, related to real-world validation activities. I agree that this is the ultimate test for the errors provided.

In practice there will be a difficult to quantify group of contributions to the error budget which are quasi-random, quasi-systematic. Error terms related to input parameters (climatologies, estimates of auxiliary information on the surface, clouds, aerosols impact on trace gas retrievals, temperature/humidity profile information, measurements from other space instruments, the a-priori and other model information) may average out over long time periods (e.g. a year) but are typically (strongly) correlated in space and/or time. Are there any general recommendations that can be made for this group

of error contributions? Sometimes such contributions are presented to users as "random" and sometimes as "systematic" by the retrieval teams. It would be good if the authors discuss this random/systematic distinction in more detail and, where possible, provide clear recommendations how to deal with this.

———————————————————

---

## Referee Comment (RC2) · Anonymous Referee #2 · 3 Dec 2019

I have to confess that I am still puzzling what was the real intention of the authors in submitting this long and, to some extent, verbose report for publication to AMT.

Although I appreciate the effort in contributing to simplify the exchange of L2 data and explain their error characteristics, in its present version the paper seems just an occasion for the many authors to recount and self-reference what they did in the area of inverse/retrieval algorithms for the sounding of atmospheric parameters.

The title seems to open to a wide tutorial, however at the end of the abstract they say the goal of the paper is just to provide a list of *recommendations which shall help to unify retrieval error reporting*. In section 3, it seems that the authors want to redefine terminology about errors. Do we have to call the root mean square error, simply uncertainty? And the variance, precision? Or whatsoever? Do we have to stick to new definitions issued by JCGM and BIPM? Is it a problem of terminology or contents? Or simply, do authors want to set up a sort of protocol for exchanging L2 products? By the way, in the end, I count 6 CoAs and 18 (with subpoints) recommendations, for a total of 24 and more. To me, more than 3 recommendations are effective as no recommendations at all. In effect, 24 recommendations are normally much more than the degrees of freedom or pieces of information conveyed by common retrievals.

Looking deep inside the paper, I can see interesting aspects about trying to define a common paradigm to interpret data coming from a large variety of satellite data processors. However, this objective is somewhat lost among unnecessary details of retrieval schemes, methodological issues and what I could call a silent but insistent criticism to Optimal Estimation. Furthermore, I think that the format of the present study is much more adequate for a report.

Concerning *retrieval error reporting*, the canonical Theory of Statistics has been teaching us (e.g. Kendall and Stuart Vol I, II, III, *The Advanced Theory of Statistics, Fourth Edition, 1979*) for so many years that the performance of a given statistic or estimator, say $\hat{x}$ , is measured in terms of its mean square error or deviation from the *true value*, which can be decomposed in variance and bias, namely

$$E[(\hat{x} - x)^2] = E[(\hat{x} - \bar{x})^2] + E[(x - \bar{x})^2]$$

For the assessment of the root mean square error and its reporting, the consolidated usage is today to share and/or distribute.

1. Estimated state (of course) and related retrieval covariance matrix
2. Background (state and covariance)
3. Averaging Kernels

Based on the above items, the performance of any estimator (bias and variance) can be unambiguously quantified. From what I can see, in the end, the above three ingredients are what authors agree with to be the basic items to share. In this respect, a potential list of recommendations, included that of authors, could be made and explained in one-two pages.

I have also to say, that authors' recommendation list itself is largely independent of the bulk of the present paper.

General Remarks

The paper is lacking a correct definition and assessment of bias. Authors seem to identify the random component of the root mean square error as the error or uncertainty of a given retrieval system. What about the bias? What's the strategy they want to set up to estimate it and eventually share with end users?

I have found a bit confusing the question about grid-independent retrievals, which for me is a non-sense, since normally one works with a discretized state vector. Apart from forward model (FM), the bias depends

on the given constraints, which are normally grid-dependent, in the sense that their definition and use is contingent to the way the state vector has been discretized. In effect, for a regularized estimator the bias depends solely on the constraints (again apart from FM biases). This basic aspect has been largely overlooked in the paper, and in fact their recommendations are not consistent with a correct sharing of the root mean square error.

On the same line, their CoA2 is inconsistent with the idea of root mean square error. Furthermore, I am not sure if it can be implemented, in practice. To streamline my personal thinking, let's suppose $W$ is a suitable interpolation/extrapolation operator, which transforms a given estimator $\hat{x}_{n1}$, defined on a grid with $n_1$ layers, into a new one, say $\hat{x}_{n2}$, defined on a grid with $n_2$ layers, we have

$$\hat{x}_{n2} = W\hat{x}_{n1},$$

with $W$ a matrix of size $n_2 \times n_1$. CoA2 requires that, using authors' language,

$$WS_{x,noise,n1}W^T = S_{x,noise,n2},$$

where, $S_{x,noise,XX}$ is the error covariance directly retrieved on the grid with $XX$ layers. However, I cannot see how the above condition can be met for any choice of $W$ and $n_1 \geq n_2$ or $n_1 \leq n_2$. Atmospheric state vectors are not band-limited signals, therefore a mere extrapolation/interpolation of a given retrieval from a coarser to a finer grid will not show finer structures of the underlying state. Hence, the above condition would normally not be met.

Condition CoA2 seems to have been set up just to criticize the concept of smoothing error, which is the way Rodgers considers for the bias. Since the bias of the individual, single, retrieval depends on the true value, which is normally not know, Rodgers considers the variability of the true value (variance-covariance) in order to have at least an estimates of the interval in which the bias is expected to range. However, the variability and/or stochastic behaviour of the state vector, which is correctly considered in OE, is overlooked by authors. They say, "natural variability is not a genuine retrieval error". It seems to me that authors purposely mislead statistical error with mistake. Natural variability is what makes our weather to be forecastable, but not exactly predictable. This is why we need statistics to address natural variability.

Taking into account the natural variability of the state vector, it is possible to perform an assessment of the estimator's bias, e.g., through the (unfortunately named) *smoothing error*, whose meaning has been, in fact, completely mislead by authors (see also later when dealing with the smoothing error).

Finally, because of the many issues addressed in the paper, in the end it looks like a confusing revision of Rodgers 2000; a sort of *pout-pourri* of about everything is known today on atmospheric inverse problems: Twomey, Tikhonov, Rodgers, LS, ML. Furthermore, the estimator described in Eq. (4) in the text is not rigorously derived from any basic principle of statistics, it is just copied from OE and rewritten by substituting $S_a^{-1}$ with $R$.

Specific Comments

Pag. 3. At best, CoA2 is only consistent with the variance component of the estimated error. What they want to do with the bias is not clear. Stand as is, I have doubt CoA2 is effective and can really work.

Page 4. Section 3.1 This is confusing. Please state exactly why uncertainty cannot be used or why it sounds ambiguous if referred to the root mean square error of an estimator.

Page 6, Eq (3), I cannot see any point why the unconstrained Least Squares solution should be called "Maximum Likelihood". This is a misconception. The assumption of Normal pdf is what really qualify the estimator (3). The reason of using ML because it yields LS under normality is untenable; it is like saying that

a meteorologist is using Einstein General Gravity (EGG) theory when forecasting the atmosphere with the Newton dynamical equations, because EGG retrieves Newton in the limit of low velocity. Why do authors not qualify the bias and variance of the estimator? Why the reader has to wait until section 6, just to see the variance alone of the estimator.

Page 7, Eq. (4). This is the worst part of the paper. Equation (4) is the OE estimator where $S_a^{-1}$ has been substituted with $R$. In force of this unjustified and ad-hoc substitution, authors claim that the estimator (4) becomes more flexible and powerful than the OE shown in Eq. (6). Also, in this case the variance of the estimator has been presented to the reader in instalments; first Eq. (7) and then an incredible jump to go to Eq. (18). In addition,

a. The bias of the estimator is not qualified/assessed/quantified in any part of the document
b. What is the reason to change $S_a^{-1}$ with $R$? What are the expected improvements?
c. Why has the Tikhonov-Twomey regularization $\gamma$-parameter disappeared? That is why not $\gamma R$?
d. What's the role of $x_a$, and why not $x_0$ as in Eq. (3)?
e. With $R$ set to any of the suggested matrices, 0-1-2 order difference matrices, Eq. (4) is dimensionally inconsistent. The authors seek a protocol-independent of constraints and other assumptions, but they propose to use an estimator, which is dimensionally inconsistent and depending on the units used for the state vector. In which way do they achieve dimensional consistency between the two terms in the squared brackets?

It would be much fairer to say "Equation (4), as well as Eq. (3) (e.g. global fit), has been normally in use for the retrievals from satellite-borne limb sounding and occultation observations. It is here considered because still now many satellite processors rely on it. Or something similar. The description of the various estimators, LS, TT, OE should be as much as neutral and respond to the need to just explain their error characteristics.

Page 7, line 30. What do you exactly mean with *smoothed?* What is a *smoothed profile?* How smoothing is quantified, and why this is a good property. In comparison to estimator (3), estimate (4) is biased and the bias structure is determined by $R$, which is grid dependent. So, how the estimated errors can be propagated according to CoA2? What is the solution proposed by authors: just forget about bias?

Page 8. Eq (6). Now that the authors have invented $R$, they can say *our estimator* retrieves the OE estimate if we put $R = S_a^{-1}$, unbelievable! By the way, to me, to $R = S_a^{-1}$, is the only possible choice, if we want to reach dimensional consistency.

Page 8, paragraph beginning at line 8. This comment seems to stay here just to add some references. By the way, it is not appropriate for Eq. (6). This is a comment to be added soon after Eq. (5). It does not apply to Eq. (6), in fact, OE elegantly solve the problem of high dynamic range of the state vector, because $S_a$ has the right dimension to properly scale the state vector. As shown in many papers, OE can be solved for the scaled variable $\tilde{x} = S_a^{-1/2}x$, which is equalized to a standardized variate, at each layer.

Page 8, Eq. (7) and discussion after. Here it seems that an *essential role in error estimation* is played by the variance of the estimator alone, and the bias? Once again, how the bias of estimator (4) is qualified/assessed/quantified?

Section 5. All is said in this section is today overcome by *Simultaneous Retrieval.* Section 5 is out-of-date and should be totally removed.

Section 5.4.5 Still Onion Peeling?

Section 5.4.6 See point above. I recommend a CoA0: Please forget about ad-hoc and non-optimal methods!

Sections 6.1 to 6.3 can be summarized under a very short section entitled "Instrument Noise and Forward Model bias"

Section 6.4. Authors here simply miss the important point that the Averaging Kernels matrix, A, qualifies and serves to assess the bias error, at least the part coming from the background constraint. In fact, if we take expectations on both side of Eq. (25) all random components associated to the instruments are averaged to zero, and we remain with the expectation value, $E(\hat{x})$. Systematic component, originating from the forward model, can be dealt with appropriate transforms of the radiance vector, e.g., *random projections*.

Section 6.4.1. All the verbose premise of the paper points straight to this criticism of the smoothing error. However, the only thing which is fairly criticisable here is the word *smoothing. In fact, s*mooth, smoother and similar terms should be banned from the context of error assessment and analysis. If Rodgers had said the retrieval can be regarded as a **biased estimate** of the true state, then everything would have gone to the right place. In effect, the smoothing error is the missing bias term to be added to the variance in order to have an estimate of the root mean square error, $E[(\hat{x} - x)^2]$. In principle, there is no need to interpolate/extrapolate to different grids a given state vector for the purpose of comparison. For visual inspection, one can just plot the given estimators and confidence intervals on the same plot, using the proper pressure-altitude grid. Why the quest of plotting differences?

Pag. 27 and 28. Eq. s (28) and (29) can be left to more elaborated comparisons. There is no need to cover this aspect in the present paper.

Pag. 28. Eq. (30). What do you mean "better resolved"? Please, quantify. The paper is aiming at providing recommendation, this cannot be given in terms of ambiguous qualitative terms.

Pag. 6.4.3 From section 6.4.3 on, until section 7, the paper appears to be unnecessary long.

Section 7. As said at the beginning 18 recommendations are too many to be useful.

Table 1 and Table 2. I do not understand the scope of these two tables. If authors want to provide a list of official L2 data providers, the list is too long since it should show only Agencies. If the authors want to provide a list of the many scientists dealing with Satellite Data Processors, it is too short.

---

## Author Comment (AC1) · 16 Jan 2020

**Review #1:**

The authors thank the reviewer for their careful and thorough reading of our manuscript and for their constructive comments.

(1.0) **Review: In their paper Thomas von Clarmann and co-authors provide a list of recommendations on how to report on errors, based on the activities of the TUNER project. To my opinion this is a very important and timely topic, and I acknowledge the effort made by the author team to write a dedicated paper discussing this point. I found the paper interesting, but also had two major reservations and a substantial number of comments, as detailed below, which require a major revision of the paper before publication.**

**Reply:** We thank the reviewer for this encouraging overall appreciation. However, we suspect that it remained somehow unnoticed that the paper was submitted as a review paper. This, we hope, justifies several sections which might not be necessary in an original research paper, and may make some of the criticism obsolete.

**Planned Action:** We will metion in the introduction that this paper is a review paper.

(1.2) **Review: Major general comments:**
**My first major reservation: the purpose of the paper is the formulation of the list of recommendations for a more uniform and complete error reporting in level-2 satellite data products (see the last line of the abstract and section 7). However, the bulk of the material presented in the paper is basically a review of real-world implementations of optimal-estimation based (and related) profile retrievals. As such the authors could consider to split the document into two papers, a review of profile retrievals and a shorter more focussed paper about unified error reporting.**

**Reply:** Retrieval and error estimation are intertwined. The relevance of error sources can depend on the retrieval scheme chosen. We have thus intended to lay down the entire framework in which the error estimation takes place. It is often the real-world implementation which gives rise to some errors in the retrieval. Thus we think that the detailed description of the real-world implementations and the error reporting issue should not be torn apart. The consideration of various specific retrieval implementations makes the difference between our paper and existing literature. We concede that the link between these sections and the recommendations should be made clearer.

**Planned Action:** Instead of splitting the paper we will try to make the logical connection between the sections of the paper clearer.

In order to strengthen the link between the retrieval section and the error reporting section, additional section references will be included.

On page 3, l1, we will add: "[We then systemize and discuss the various sources of retrieval error] **and, if applicable, their dependence on the retreval scheme chosen**[(Section 6)]"

On page 18,Section 6.1.1., l. 15, we will add: "[The situation is different in a decomposed retrieval] **(Section 5.4)**"

On page 18, Section 6.1.1., l22, we will make a direct link to Section 5.5: "[In the context of error propagation in the the Levenberg-Marquardt algorithm] **(Section 5.5)**[, it is important to distinguish two different applications]"

On page 21, Section 6.2.1, l30, we will add "[by the radiative transfer model $\vec{F}$ in use] **(Section 4)**"

On page 24, Section 6.3, l4-5, we will add "[... is already known] **(see, Section 5.4)**".

On page 24, Section 6.3.1, l23, we will add "[In onion peeling] **(Section 5.4.5)** [the ray...].

On page 25, Section 6.4: There is already a link to Eq. (4).

(1.3) **Review: Several sections of the paper are providing useful functional background information for section 7. But for quite some sections I could not find the link with the final recommendations. Examples are section 5.4 and also parts of 4, 5.1 and 6.4 (e.g. 6.4.3 to 6.4.6), 6.7. Because of these sections the paper is very long.**

**Reply:** Section 4: Eqs 1-4 lay down the notational basis for the rest of the paper. The text on maximum likelihood approaches seems important to us because the current literature is strongly biased towards Bayesian methods. Our intention has been to get the community together and to form a common framework.

Section 5.1 Discretization: Here we introduce concepts where the result is represented in other ways than value over vertical coordinate. Without introducing this, Recommendation 11 would be incomprehensible.

Section 5.4 Decomposition: Without decomposition of the retrieval problem there would be no parameter errors at all, and different decomposition approaches require different ways to estimate the retrieval errors. Thus we consider this section as essential.

Section 6.4.3 Altitude resolution: We consider to add an amendent to Rec. 16: "If averaging kernels are only provided for a few representative cases, one might still consider to show the vertical resolution profiles for each profile."

Section 6.4.6 Regularization crosstalk: This is an often overlooked error component and may be essential for a complete error budget.

Section 6.7 Drifts: This section is indeed not referred to from anywhere else in the paper, but without it, we are pretty sure, we will be blamed to have forgotten this issue.

**Planned Action:** The links between these sections and the recommendations will be made clearer.

(1.4) **Review: Section 5.4 is a review of (profile) retrieval approaches, but contains a lot of material which is not directly relevant for the paper. Personally, I would propose to shorten this section, keeping possiby the tables (and references) and keeping those remarks which are important in the context of error reporting. This review-like section also leads to a very long list of references. It would be good to mention only those references that bring new information to the discussion how to present the retrieval errors.**

**Reply:** Well, this paper is indeed meant as a review paper. It is for this reason that we touch more topics than directly necessary for the recommendations and aim for a comprehensive list of references. Even if not all the sections are necessary to infer the recommendations, the information they provide is often still necessary to comply with the recommendations, i.e., to provide the data characterization as requested in the recommendations.

**Planned Action:** We will try to find a place in the paper where examples can be included which demonstrate the relevance of the material presented.

(1.5) **Review: In general, there is quite a big conceptual step (gap) between sections 3-6 and the summarising recommendations in section 7. Ideally all recommendations should be complemented by motivations, examples and explanatory information in the sections preceding section 7. The link between the recommendations and the rest of the text (which reads as a review of retrieval methods and theory) is often unclear to me.**

**Reply:** As stated above, the paper has indeed been submitted as a review paper. We think that retrieval theory and error estimation are fairly intertwined issues. Any change in the retrieval setup needs to be reflected by the error estimation approach.

**Planned Action:** The link between the recommendations and the theoretical parts will be made clearer.

(1.6) **Review: I would suggest that the authors go through sections 3 to 6 (see list of subsections mentioned above) and remove discussions which are not functional to motivate the requirements presented in section 7.**

**Reply:** We think that a section is justified not only if it motivates the recommendations but also if it provides the technical information needed to understand and to provide what the recommendations require.

**Planned Action:** Again: The link between the recommendations and the theoretical parts will be made clearer and more specific.

(1.7) **Review: After reading the first sections of the paper it was not fully clear to me what is really the problem which is addressed? In what sense are retrieval products not comparable? Please provide (generic) examples of retrieval products which miss information which makes a direct comparison between retrievals, or comparisons with independent data difficult or impossible.**

**Reply:** We agree that generic examples will be useful.

**Planned Action:** Examples will be included.

(1.8) **Review: In what sense is there a need for a new set of recommendations, e.g. what is missing after the work of e.g. QA4EO or the GUM?**

**Reply:** Regarding the treatment of uncertainty, QA4EO basically recommends to apply the GUM recommendations. GUM does not take constrained retrievals using prior information into account. It does not take the problems into account which are caused by the real-world retrieval schemes. Broadly speaking, it is not specific enough for our purpose. limitations of the GUM for remote sensing application were identified already in, e.g., Povey and Grainger, AMT, 2015, section 2.3, where they state "These conventions [from GUM] apply equally to satellite remote sensing data but represent an impractical ideal that does not help an analyst fully represent their understanding of the uncertainty in their data. This is due to the simplistic treatment of systematic errors."

**Planned Action:** We will expand on this in the paper.

(1.9) **Review: The final set of recommendations are focussing on profile retrievals. But the tables include also total column retrieval examples (e.g. DOAS). I think this is a missed opportunity, and I would encourage the authors to formulate explicitly what their recommendations are for column retrieval products (some recommendations are generic, but several parts of section 7 explicitly refer to profiles).**

**Reply:** We agree.

**Planned Action:** For those recommendations which refer to profiles only, a respectice recommendation for column retrievals will be included.

(1.10) **Review: Arguably the atmospheric composition data assimilation community is the main user of satellite retrieval products. This**

**community and their needs are basically not discussed in the paper.**

**Reply:** The quantities needed by the data assimilation community are basically those we recommend to provide: Covariance matrices of the data uncertainties and averaging kernels. The former are needed for their observation error covariance matrix; the latter are needed for their observation operator.

**Planned Action:** We will metion the need of correct diagnostics for data assimilation.

**(1.11) Review: More generally the users of the data do not receive much attention, and the requirements are discussed from a L2 data provider point of view.**

**Reply:** This paper is indeed addressed to data providers. A tutorial paper addressed to data users is in preparation. There the correct use of the data characterization will be demonstrated. We agree, however, that the relevance of the correct use of the error estimates, averaging kernels, etc. in quantitative applications like data comparison, time series and trends, data assimilation etc. deserve to be mentioned.

**Planned Action:** We will add to the intro: "This review paper, the first 'foundational' paper from the TUNER team, is mainly addressed to the providers of remotely sensed data. A paper addressed to the data users, guiding them through the correct use of the uncertainty information, is currently being written (Livesey et al., in preparation)" In addition, we will mention some typical applications of satellite data along with their specific needs with respective to data characterization.

**(1.12) Review: This is my second major reservation. Some parts of the text refer to the validation activities, but this is not presented in a very structured way. The needs and feedback from the validation and assimilation communities on existing L2 satellite products would be an important starting point to discuss requirements for satellite data products.**

**Reply:** While validation and assimilation are not meant to be the main content of the paper, we agree that the relevance of correct data characterization is essential for these purposes.

**Planned Action:** We will mention the relevance of correct data characterization for validation and assimilation, and we will highlight the relevance of validation studies for the assessment of the adequacy of error estimates.

**(1.13) Review: Some assimilation users would prefer to work directly with the level-1b data, an option which is also not discussed.**

**Reply:** The problem of the assimilation of L1B data is that all errors (parameter errors etc.) have to be mapped into the radiance space to get the observation error covariance matrix right. In spectral measurements this will typically lead to fully correlated non-sparse covariance matrices, which are, to our knowledge, not favoured by the assimilation community. While direct radiance assimilation is successfully applied to nadir sounders, we are not aware of any application where, say, tangent altitude offsets are correctly dealt with in high-resolution limb sounding data. There is certainly a lot to say with respect to this issue, but we think that a deeper discussion of related issues is beyond the scope of this paper, which is already quite long.

**Planned Action:** We will mention in Section 5.2 that in direct radiance assimilation it is important to consider a measurement covariance matrix which does not only contain noise but the mapping into the radiance space of all uncertainties of parameters which are not assimilated.

(1.14) **Review: The recommendations in section 7 are not always formulated as a recommendation, but leave room for interpretation and implementation. I sometimes found the CoA points in section 2 even more clear and explicit than the recommendation points. It may be useful to split the list in section 7 in actual (strong) recommendations and related discussion points. Sometimes it is not so clear what is actually recommended by the authors, e.g. due to a trade-off between completeness and data volume, or aspects are left to the retrieval teams to decide (e.g. point 1, 2, 3, 4, 16, 18).**

**Reply:** The general problem is that the specific decision depends on the instrument and the retrieval approach chosen. The decision is under responsibility of the retrieval scientist. However, we provide criteria to judge if the decision was correct.

**Planned Action:** We will make clear that the choices are not *ad libitum* choices but must comply with our general rationale that the error estimates must explain observed inter-instrument differences.

(1.15) **Review: I was expecting recommendations also regarding the naming (see section 3). The authors discuss in particular "error" versus "uncertainty", but do not really provide a clear guidance on what to use..**

**Reply:** Whatever naming we would suggest, it would always be in conflict with a part of the community. We consider it as a major progress if awareness of the ambiguities of language is created, and if authors clearly define the language they use. Usually no language is *per se* better than another language, and we do not want to stipulate conventions. We want to restrict the recommendations

to the objectively necessary.

**Planned Action:** We will make clearer in the paper that no language is *per se* better than another language, but authors should clearly define the language they use.

(1.16) **Review: Also, the consistency or inconsistency with the GUM activity are not clear to me after reading the section. The reader is referred to a paper in preparation.**

**Reply:** The criticism of GUM is controversial among the TUNER community. Thus we found it appropriate to defer these issues to a paper which is authored only by those who wholeheartedly endorse this criticism. We agree with most of the technical recommendations in GUM, although we think that these are not specific enough for our purpose. We do not all agree with their construal of the concept of 'error', and we do not all agree that we can dispense with the concept of the 'true value'.

**Planned Action:** We will check if we can make the related paragraph clearer without becoming biased towards or against some of the concepts under dispute.

(1.17) **Review: Retrieval datafiles contain parameters labelled as "precision", "accuracy", "trueness" etc. and different guidelines exist from different space agencies and for different application areas. It would be useful if the authors can discuss naming conventions also in this paper and express a clear opinion/recommendation.**

**Reply:** We have intentionally avoided this. One naming convention might not be more adequate than another one. We do not want to make prescriptions with respect to conventions but we have tried to restrict our recommendations to those that can be inferred from the conditions of adequacy.

**Planned Action:** In the paper we will make the rationale outlined above explicit.

(1.18) **Review: Machine learning approaches are getting more and more popular and deserve some special attention. Several machine learning implementations for retrievals are limited on the error information they provide. It would be useful to have some targeted recommendations for these approaches as well.**

**Reply:** There are indeed some relevant issues here. If in machine learning the machine is trained with retrieved data (from a conventional retrieval) then all uncertainties of the latter propagate on the regression parameters created by the machine-learning scheme. A variant of machine learning is supervised learning with neural networks. There are two distinct approaches to the use of neural

networks in remote sensing. They can be used for the forward modelling of radiative transfer. In this case all the error analysis we describe is still feasible and valid. Or they can be used directly for the retrieval. Then the error estimation schemes presented in our paper are not easily applicable.

**Planned Action:** We will try to add a short paragraph on these issues but we do not want to add too much length to the paper.

(1.19) **Review: Detailed comments:**
**Abstract: The abstract reads like an introduction. I would encourage the authors to summarise (shorten) the first part and expand on the last sentence with a summary of the content and main results of the paper.**

**Reply:** We think that the abstract summarizes the main parts of the paper and the general ideas behind them.

**Planned Action:** none

(1.20) **Review: Introduction l6: "reduction"? Should this be "deduction"?**

**Reply:** We think "reduction" is correct here (in the sense of the technical term 'data reduction').

**Planned Action:** This issue may become obsolete because major parts of the introduction will be rewritten anyway.

(1.21) **Review: l16: "The project ... is a consortium of". Please modify**

**Reply:** Thanks for spotting.

**Planned Action:** The entire introduction will be rewritten.

(1.22) **Review: l24: "atmospheric composition and temperature profiles". What about other profiles, e.g. water vapour?**

**Reply:** We consider water vapour as a constituent of the atmosphere and think that it is thus covered by "composition".

**Planned Action:** none

(1.23) **Review: Is the paper limited to profiles, or are single property (column) retrievals also included?**

**Reply:** We do not see a fundamental difference. A column amount can be conceived as a profile containing one element. Some parts of the paper refer directly to column retrievals, e.g., the column averaging kernel (Eq. 26).

**Planned Action:** Particularly in the recommendations section more weight will be given to column retrievals.

(1.24) **Review: l37: "are do not need to be", please correct.**

**Reply:** Thanks for spotting!

**Planned Action:** This will be corrected.

(1.25) **Review: Section 2:**
**l82, CoA 1: "and/or error estimation schemes". Would it not be better to say "and/or retrieval schemes"?**

**Reply:** yes, indeed.

**Planned Action:** This will be reworded: "[The error estimates should be intercomparable among different instruments], retrieval schemes, [and/or error estimation schemes.]"

(1.26) **Review: CoA 2: "independent of the vertical grid". But I assume at this point that error covariances are specified on a specific grid used in the retrieval ?!**

**Reply:** Yes, they are. But generalized Gaussian error estimation applied to the resampling on other grids will produce the correct covariance matrices also on other grids. This is not usually true for the smoothing error, thus our critical position with respect to the latter. Needless to say that interpolation to a finer grid causes a sort of smoothing error but this is not what we understood should be included in the noise error. It is another category of error. Thus it cannot be expected to be rendered by the propagation of noise to a finer grid.

**Planned Action:** none

(1.27) **Review: CoA 5: "and different amounts of prior information". Do you mean "and different sources of prior information"?**

**Reply:** Actually we need both: amounts (weight) of the prior and the values of the prior itself.

**Planned Action:** We will reword this : "[...different amounts of] possibly different [prior information]".

(1.28) **Review: p3, l10: "but we consider it unrealistic to assign quality indicators for 'fitness for purpose' for all conceivable applications." This is an interesting remark. It would be useful to expand on this: explain how it is discussed by QA4EO and which parts are unrealistic.**

**Reply:** There is an almost infinite number of possible applications and purposes. Thus it is impossible to provide fitness-for-purpose indicators for all, and we consider it not useful to select a few of them on an ad hoc basis.

**Planned Action:** none

(1.29) **Review: Sec 3.1: please introduce the acronym "Joint Committee for Guides in Metrology (JCGM)" just once, and use only the acronym "JCGM" in the rest of the paper.**

**Reply:** Agreed.

**Planned Action:** This will be corrected.

(1.30) **Review: sec 3.1, l34: "actually claimed that there are conceptual differences between error analysis and uncertainty estimation." For readers who did not follow this debate it is hard to follow this section. It would be helpful to add a few sentences to list the claimed conceptual differences between these two terms.**

**Reply:** In wide parts of the literature - from Gauss to Rodgers and beyond - 'error' is used also as a statistical estimate of the absolute difference between the estimate and the true value. There are concepts which allow to estimate this quantity without knowing the true value (we say 'estimate', not 'know'!). GUM seems to ignore this connotation and it appears (however, this is not quite clear in their documents) that they refer to error only as the actual difference between the estimate and the true value.

**Planned Action:** We will try to find a clearer wording for this part.

(1.31) **Review: Section 4: I find it useful to include a section with the theoretical background and notation. In fact, using this notation could be a recommendation (Section 7, point 1).**

**Reply:** We are happy that our section on theory and notation is appreciated. In our recommendation #1 we write " We hope that this paper serves that [clearly defined language and notation] purpose". But since no notation is *per se* better than any other one, we do not feel to be in a position to dictate which convention others should use, as long as everything is clearly defined.

**Planned Action:** We will make this recommendation clearer by adding "[that

purpose] and that the terminology and notation introduced here will be found useful."

(1.32) **Review: eq. 2: "can only be approximated" What does this refer to? The ill-posed or underdetermined nature of many inverse problems?**

**Reply:** yes, exactly. And beyond this, large rank of the matrix to be inverted, which will impose some practical limitations.

**Planned Action:** We will add "[only be approximated] due to the over- or underdetermined or otherwise ill-posed nature of the problem and the large rank of the matrix to be inverted."

(1.33) **Review: l70: "macrorcopic"**

**Reply:** Thanks for spotting.

**Planned Action:** This will be corrected.

(1.34) **Review: l77: What is the approximation which turns "f" into "F". Are these real-life uncertainties in f? Is F now a matrix or still a non-linear function?**

**Reply:** $\vec{f}$ represents the (unknown) true radiative transfer function, and $\vec{F}$ the model we use. $\vec{F}$ typically is a nonlinear vectorial function.

**Planned Action:** We will add: "$\vec{F}$ is a vector-valued non-linear function and deviates from $\vec{f}$ in that it involves numerical approximations and may not include the full physics of radiative transfer." We will correct **F** to $\vec{F}$ in the text.

(1.35) **Review: l87: "overdetermined case (m ¿ n)". Whether or not the inverse problem is overdetermined also depends on F, and not only on the size of the vectors. Add "and not ill-posed". (This is discussed on next page)**

**Reply:** We are afraid that here different terminologies clash. According to the conditions of well-posedness by Hamadard, every inhomogeneous over-determined problem without collinear equations is by definition ill-posed because it does not have an exact solution. Thus 'overdetermined and not ill-posed' is usually an unsatisfiable condition. We use the convention endorsed, e.g., by James E. Gentle, Numerical Linear Algebra for Applications in Statistics, DOI https://doi.org/10.1007/978-1-4612-0623-1, Springer-Verlag New York, Inc. 1998, Print ISBN 978-1-4612-6842-0, Online ISBN 978-1-4612-0623-1, Series Print ISSN 1431-8784, who states in Chapter 3, page 94: "However, many of the linear systems that occur in scientific applications are overdetermined; that is, there are more equations than there are variables, resulting in a nonsquare coefficient matrix." The Wikipedia article entitled "Overdetermined System" (retrieved 16 Jan 2020) they even state implicitly that collinearity ot the equations is not in conflict with over-determinedness.

**Planned Action:** none

(1.36) **Review: p5, l7: "In most real-world applications, only measurement noise is considered here, while other measurement uncertainties like calibration errors are neglected at this stage." Remove "here" and "at this stage"**

**Reply:** We have added these words with intention. Otherwise the reader might think that we claim that other error sources are omitted also in the error propagation.

**Planned Action:** none

(1.37) **Review: p5, l21-44: This is an interesting historical note, but not essential for this paper and may be removed.**

**Reply:** Since this paper is intended to be a review paper, we think that it is appropriate to put the methods in their historical context. Furthermore, there seems to be quite some confusion about what maximum likelihood is, how it can be justified, and which dubitable assumptions it avoids, and the pro/contra likelihood discussion seems often to be based on half-truths. Thus we think it would be useful to guide the interested reader to the original literature.

**Planned Action:** none

(1.38) **Review: eq. 5: What is L1? What are its properties?**

**Reply:** Some information will be added.

**Planned Action:** This will be reworded: "With the $(n-1) \times n$ first order differences matrix $\mathbf{L}_1$ and $\gamma$ a scaling parameter to control the strength of the regularization, the choice of
$$\mathbf{R} = \gamma \mathbf{L}_1 \mathbf{L}_1^T, \tag{1}$$
renders fields of profiles..."

(1.39) **Review: Sec.5: l22: mention the loss of information**

**Reply:** We think that the loss of information is included in "and limits the spatial resolution of the solution"?

**Planned Action:** none

(1.40) **Review: Sec 5.4. This section is basically a review of retrieval approaches: why is it relevant for this paper to include such a review? See my general remark above.**

**Reply:** We consider it as relevant, because any error propagation scheme can only be understood in the context of the related retrieval scheme. What in one kind of decomposition is accounted for by the error propagation of noise needs explicit evaluation of the related parameter error in another kind of decomposition.
Further, we recall that this paper has indeed been submitted as a review paper and we suspect that this information has been lost somewhere in the system.

**Planned Action:** The introduction will be rewritten to make the purpose of the paper clearer.

(1.41) **Review: Sec 6, p10, point 2: Model errors: It would seem logical to me to split this into RT model errors and inputs used by the forward and inverse models, e.g. influence of atmospheric aspects like surface characterisation, aerosols and clouds, other meteorological variables (humidity, temperature).**

**Reply:** According to our systematics, the latter are not model errors but parameter errors. Otherwise the reviewer's suggestion and the way we organize this section are very close. "Incomplete Models" and "Numerical Issues" together cover the RT model errors, and the inputs used by the RT model (as far as they are not parameter errors) are the uncertainties in the "model constants". If we combined both RT subsubsections into one subsection, we would need the paragraph caption which is not allowed according to AMT formatting standards. Thus we are forced to keep the hierarchy of sections flat.

**Planned Action:** none

(1.42) **Review: Sec 6, p10, point 3: "errors caused by decomposing the inverse problem". Does this deserve a separate section?**

**Reply:** We think so, because it has major implication on error estimation and reporting. In a joint retrieval of species $A$ and $B$, there is no parameter error due to uncertainties in $A$, but in a sequential retrieval (first $A$ then $B$), there is.

**Planned Action:** We will edit the text to make the logical flow better visible.

(1.43) **Review: Sec 6.1.1, l37: "cheerful" ...**

**Reply:** That's what we felt...

**Planned Action:** None by now.

(1.44) **Review: Sec 6.1.3, l33: "measurments"**

**Reply:** Thanks for spotting.

**Planned Action:** This will be corrected.

(1.45) **Review: Sec 6.2.1: "If a complete model is available but not used .., the effect of the missing processes can be assessed via sensitivity analyses based on the complete model ...". This sounds like a recommendation (could be part of section 7).**

**Reply:** We think that this is implicitly included in the completeness requirement in the recommendations. Our recommendations only say what we want but not how it should be achieved. It is the content of the preceding sections to present and discuss methods how this can be achieved.

**Planned Action:** While we are reluctant to add further recommendations, we consider to mention this as an example along with the respective recommendation.

(1.46) **Review: p15, l3-7: "The OCO-2 team is currently working on ..". I could not understand this paragraph. I suggest to either explain the approach in more detail or omit.**

**Reply:** We agree that this is hard to understand.

**Planned Action:** This part will be rewritten.

(1.47) **Review: l24: "retrived"**

**Reply:** Thanks for spotting.

**Planned Action:** This will be corrected.

(1.48) **Review: p16, l40: "the derivative". I do not understand how to take such a derivative.**

**Reply:** $A$ is the derivative of the retrieved state with respect to the true state. With a *tertium non datur* assumption (this is here that we disregard the dependence of the solution with respect to noise and other uncertainties, which

are addressed elsewhere), then **I**-**A** is the derivative of the retrieved state with respect to the prior information. Thus it is not necessary to differentiate $\hat{\hat{x}}$ with respect to **I**-**A** explicitly.

**Planned Action:** We are somewhat reluctant to add much length to the paper with respect to this but we will make reference to the literature where the use of **I**-**A** is introduced. Probably the Rodgers book, his Section 3.2.

(1.49) **Review: Sec 6.3: The parameter errors are often very relevant and could be discussed more extensively. For these parameters often simplifying assumptions are made (e.g. climatologies) or they are taken from elsewhere (e.g. actual weather model output) or they may be derived in the retrieval itself (or previous step in the retrieval). All these choices will lead to different characteristics for the related errors, often introducing quasi-systematic error correlations.**

**Reply:** We agree.
**Planned Action:** We will expand on this.

(1.50) **Review: 6.3: Why is this section called "parameter errors" instead of something like "Inverse model decomposition errors"**

**Reply:** Because these errors are not always caused by the decomposition. Sometimes just prior assumptions or external information are used. We call them parameter errors because they are related to the parameters of the forward model as discussed in Section 5.3. We concede that the first line of 6.3 is misleading in the way it is written.

**Planned Action:** We will reword the first lines of Section 6.3.

(1.51) **Review: Sec 6.4.1 and 6.4.2: I'm happy that the authors include these two "interpretations". This is a subtle point, often not understood by satellite data users.**

**Reply:** We are glad for appreciation, particularly because reviewer #2 finds this unnecessary.

**Planned Action:** none

(1.52) **Review: p16, l87: "the undesirable effect that a smoothing error evaluated on a coarse grid will be smaller than a smoothing error evaluated on a fine grid." I do not really understand why this is undesirable. This property seems to make sense to me: more layers allow more detail to be resolved (and smoothed away by the retrieval process).**

**Reply:** Yes, but if there is a profile on a fine grid, there should be no difference in the error estimate between (case 1) the profile has been retrieved directly on the fine grid, with a certain constraint limiting the vertical resolution to a certain value and (case 2) the profile has been retrieved on a coarser grid, and the resampled to the fine grid. In both cases the difference from the truth can be the same but the (propagated) smoothing errors will be different. This seems absurd to us.

**Planned Action:** none, since this is discussed in detail in the literature referenced.

(1.53) **Review: p18, l18: "also commonly applied when measurements are compared to model data". It would be good to mention explicitly the data assimilation application here.**

**Reply:** Agreed.

**Planned Action:** We will add: "In data assimilation the averaging kernel has to be included in the observation operator."

(1.54) **Review: p19, l10: "reasons reasons"**

**Reply:** Thanks for spotting.

**Planned Action:** This will be corrected.

(1.55) **Review: Sec 6.4.3: I was wondering if this section (on altitude resolution) is needed as background to section 7.**

**Reply:** We agree that the old version of the recommendations does not make any use of the concept of altitude resolution; we think, however, that at least the altitude resolution should be reported for each single profile if only representative averaging kernels are provided.

**Planned Action:** We consider to make an amendment to R16 or R17 that, if AKs are presented only for individual cases, a vertical resolution profile for each single profile still is useful.

(1.56) **Review: Section 7 Point 2: A bit weak, it leaves a lot of room for different approaches.**

**Reply:** This is intentional. We will accept any method as long as the resulting errors explain differences encountered between independent measurement systems.

**Planned Action:** none

(1.57) **Review: Point 3: Does this have repercussions for the data volume? Especially when each component has its own covariance matrix?**

**Reply:** Yes, it certainly has. This is why we write "The ideal approach ...".

**Planned Action:** We will add: "It is the responsibility of the data provider to judge to which degree simplifications are justified."

(1.58) **Review: Point 4: Again leaves much freedom. What about proposing a 1 sigma as default?**

**Reply:** This is controversial. This default is not better than any other default, and in order not to upset researchers following other conventions, we refrain as much as possible from stipulating conventions and limit ourselves to recommendations which can in some way be inferred from agreed conditions of adequacy.

**Planned Action:** We will add a statement to the intro on our rationale not to stipulate conventions.

(1.59) **Review: Point 6: "error components available, they should also indicate how they contribute to the random and/or systematic error" What about the total error: should this consist of a random and a systematic part?**

**Reply:** Yes, the total error consists of both components. We insist on reporting them separately, because, depending on the application, only one or the other component may be important. E.g., for time series or trend analysis only the random error component is important, and additive systematic errors are irrelevant. Conversely, in monthly zonal means of densely sampled data the standard error of the mean, representing the random error, goes down to almost zero, and the systematic part of the error will survive.

**Planned Action:** We will add that, if a total error is reported, it should include both the random and systematic error components.

(1.60) **Review: What does "indicate" mean in practice? Please be more specific.**

**Reply:** We mean "report" or "describe".

**Planned Action:** We will consider to reword this.

(1.61) **Review: Point 7: It is difficult to understand what is meant here. What is the domain of a subset of a component of a source of error? It would be good to provide an example.**

**Reply:** We agree. The recommendations are very generic. This is intentional, since they should be applicable to different measurement systems and retrieval schemes. Examples can help to clarify what we mean in practice.

**Planned Action:** Illustrative examples will be added to the recommendations.

(1.62) **Review: What is the difference between an error source and an error component?**

**Reply:** In an ozone retrieval, the retrieved ozone mixing ratio may depend on temperature. The temperature uncertainty is the error source and that part of the ozone error which is caused by the temperature uncertainty is the error component. We agree that this terminology needs to be defined somewhere but we find that Section 4 "Retrieval Theory and Notation" is the better place to do this.

**Planned Action:** we will add in Section 4: "[is the measurement noise mapped into the retrieved atmospheric state.] **In other words, $S_{x,noise}$ is the error component in $\vec{x}$ due to the error source $S_{y,noise}$.**

(1.63) **Review: Point 9: "assumed ingoing uncertainties shall be reported". What is meant by "reported"? Does this refer to the ATBD, a journal paper or to the L2 datafiles themselves?**

**Reply:** At the place where the respective resulting errors are reported. There are many ways to make error analysis traceble in this respect. The important thing is that the information can easily be found. If the source of these data is accessible, a link may do.

**Planned Action:** [For all error components, the assumed ingoing uncertainties shall be reported] **in the relevant documentation**[, otherwise error propagation would not be traceable.]

(1.64) **Review: Point 10: Sometimes (I-A)$x_a$ = 0 even though the retrieval still needs/depends on a-priori information. Should the a-priori be reported also in this case?**

**Reply:** We do not quite understand how the retrieval can depend on a priori information when (I-A)$x_a$ = 0. Does the reviewer mean cases when the quantity the a priori refers to is not an element of x? According to our terminology this would not be "a priori" in the narrow sense but parameter. Or does the reviewer talk about implicit a priori information imposed by a coarse grid?

**Planned Action:** The text will be reworded in a way that it will become clear that a priori information is meant only in a narrow technical sense and does refer only to elements of the state vector actually retrieved.

(1.65) **Review: Point 10: What are "similar operations"? Please be more explicit.**

**Reply:** we mean variants of Eq. 30 which may be formally different but follow the same rationale.

**Planned Action:** "or to perform similar operations" will be replaced by "or variants of it"

(1.66) **Review: Point 11: I do not understand why it is crucial to have the results as vertical profiles (as opposed to desirable). The vertical profile retrievals are linked to the real physical world through the averaging kernels, as specified in Eq. 30. Ignoring this link leads to all the smoothing error considerations (and problems) as discussed extensively by the authors. Especially when the kernel is very different from the unity matrix I, the interpretation of the retrieved profile as a real profile becomes troublesome. The retrieval at a given altitude then contains physical information from (depends on concentrations in) many other layers, as specified by the averaging kernel matrix. The kernels will always have altitude on one axis, even if presented in eigenvalue space, and relate the retrieval to real physical profiles. Please explain why this strong statement ("should be presented") is made.**

**Reply:** We do not mean "exclusively represented as vertical profile". There is nothing wrong with presenting the data in a different way, and we agree that for certain applications this can even be advantageous. But when data from different sources are combined in one study, this happens almost always on the basis of a vertical profile representation. We just want to make sure that in this case the data characterization is available. We do not want to allow the data provider the excuse "I do not need averaging kernels because in my representation they are irrelevant and everything else is the business of the data user". We think that the data provider is in a better position than the data user to provide the averaging kernels in the profile space.

**Planned Action:** We will change the text to: "[...then the final result should] in addition [be presented as vertical profiles and also all diagnostic data (error estimates, averaging kernels) should be transformed to an altitude-dependent...]".

(1.67) **Review: Point 12: "Ideally the data provider calculates the averaging kernels on the final grid". What is proposed here? It sounds**

like a commitment of the retrieval team (data provider) to provide support to all users with a grid which differs from the retrieval grid. This would imply a major commitment. Or would this imply that each retrieval product should be accompanied by software to do the interpolation (extrapolation is also very likely!) to different grids.

**Reply:** This is not meant. We are talking about the final grid on which the data producer distributes the data. Sometimes data are retrieved on some specific grid (e.g., related to the tangent altitudes) and then resampled on a uniform grid. In this case the user is not helped much with an averaging kernel which refers to the original retrieval grid.

**Planned Action:** The text will be clarified: "[...on the final grid] on which the data are provided to the user."

(1.68) **Review: Point 13: "This is particularly important when data are reported in a form that differs from that of the retrieval state vector". This may not be very clear to the reader. Please provide an example. Why is it important in this case?**

**Reply:** E.g. the application of log averaging kernels to vmr profiles gives a mess.

**Planned Action:** We will add: "E.g., the averaging kernels resulting from a retrieval of the logarithms of mixing ratios must not be applied to mixing ratios. It is thus of utmost importance to communicate to the data user to which quantities the averaging kernels refer."

(1.69) **Review: Point 15: "If the data are understood to be a representation of the smoothed state of the atmosphere, the smoothing error is not needed and averaging kernels along with the prior information are sufficient". I suggest that the authors explicitly mention applications here, e.g. model-satellite comparisons and data assimilation.**

**Reply:** We agree.

**Planned Action:** We will explicitly mention applications.

(1.70) **Review: Point 16: "Communication of a complete error budget ... is not always technically feasible and often creates unnecessary data traffic." I would suggest that the authors include a reference to the work of S. Migliorini, DOI: 10.1175/2007MWR2236.1. This paper describes how the data volume can be reduced drastically (explicit a-priori profile and error covariance no longer needed) while preserving the full error information, to support data assimilation applications. Do the authors consider this a possible alternative for storing the retrieved profiles, see e.g. point 10, 11?**

**Reply:** We will mention this possibility. However, we find it hard enough to prevent data users from simply ignoring the diagnostic data because they seem to be too complicated, and a data-reduced representation of the matrices may make the problem even worse.

**Planned Action:** We will reference this paper and mention that it might provide (at least partly) a solution to the data traffic problem.

(1.71) **Review: Point 18: This important point distinguishes random and systematic errors, related to real-world validation activities. I agree that this is the ultimate test for the errors provided. In practice there will be a difficult to quantify group of contributions to the error budget which are quasi-random, quasi-systematic. Error terms related to input parameters (climatologies, estimates of auxiliary information on the surface, clouds, aerosols impact on trace gas retrievals, temperature/humidity profile information, measurements from other space instruments, the a-priori and other model information) may average out over long time periods (e.g. a year) but are typically (strongly) correlated in space and/or time. Are there any general recommendations that can be made for this group of error contributions? Sometimes such contributions are presented to users as "random" and sometimes as "systematic" by the retrieval teams. It would be good if the authors discuss this random/systematic distinction in more detail and, where possible, provide clear recommendations how to deal with this.**

**Reply:** Again, it is the responsibility of the data providers to make a sensible distinction here. The final criterion is that the error estimates can be confirmed by the validation of the standard deviation of the differences and the bias. In particular situations it may even be appropriate to split the contribution of one error source into a systematic and a random component. Errors which are random in longer time-scales but systematic in shorter timescales are exactly what we mean with 'errors correlated in certain domains'. In cases like those mentioned by the reviewers, it must be reported that errors are autocorrelated in the time domain.

**Planned Action:** We will add this example to R7.

---

## Author Comment (AC2) · 16 Jan 2020

The authors thank the reviewer for his comments.

**Review #2:**

(2.1) **Review: I have to confess that I am still puzzling what was the real intention of the authors in submitting this long and, to some extent, verbose report for publication to AMT.**

**Reply:** The intention of this paper is to summarize retrieval approaches actually in use in satellite remote sensing, to systematize them in a common framework and notation, to discuss the implications of related choices on error propagation and to infer related recommendations on unified error reporting.

**Planned Action:** We will state this intention clearer in the introduction

(2.2) **Review: Although I appreciate the effort in contributing to simplify the exchange of L2 data and explain their error characteristics, in its present version the paper seems just an occasion for the many authors to recount and selfreference what they did in the area of inverse/retrieval algorithms for the sounding of atmospheric parameters.**

**Reply:** The purpose of the article is to cover all the satellites for remote sensing of atmospheric compositions over the past 20 years from the all frequency ranges, microwave, infrared, NIR and UV/VIS, as a review paper. This implies numerous references, and since the list of authors includes many scientists from many different groups working in this field, it appears quite natural to us that self-referencing is inavoidable.

**Planned Action:** We will add more references from scientists who are not involved in this paper in order to make the article more balanced.

(2.3) **Review: The title seems to open to a wide tutorial, however at the end of the abstract they say the goal of the paper is just to provide a list of recommendations which shall help to unify retrieval error reporting.**

**Reply:** We are afraid that this is misreading; we do **not** say that the goal of the paper is just to provide a list of recommendations. We do mention that we provide some recommendations, but the rest of the abstract summarizes the problem areas tackled in this overview paper. Only in the section on conditions of adequacy we indeed mention the "ultimate goal of presenting a list of recommendations". The attribute "ultimate" makes clear that this is by no means the only goal.

**Planned Action:** We will address this more clearly in the abstract, introduction, and top of the recommendations.

(2.4) **Review: In section 3, it seems that the authors want to redefine terminology about errors. Do we have to call the root mean square error, simply uncertainty? And the variance, precision? Or whatsoever? Do we have to stick to new definitions issued by JCGM and BIPM? Is it a problem of terminology or contents? Or simply, do authors want to set up a sort of protocol for exchanging L2 products?**

**Reply:** We want to avoid quibbling about words. The reviewer is free to call the quantities mentioned as they like, as long as the terms are clearly defined somewhere. However, the terminology we use is applicable also to single measurements, while we have problems to assign a meaning to the terms 'bias' or the 'root mean square error' in the case of a single measurement.

**Planned Action:** none

(2.5) **Review: By the way, in the end, I count 6 CoAs and 18 (with subpoints) recommendations, for a total of 24 and more. To me, more than 3 recommendations are effective as no recommendations at all. In effect, 24 recommendations are normally much more than the degrees of freedom or pieces of information conveyed by common retrievals.**

**Reply:** We do not understand how it is logically justified to calculate the sum of the CoAs and the recommendations. We thought that sums can only be calculated of items of the same category. We also do not understand what the logical link between the number of recommendations and the degrees of freedom of a retrieval is. We do not see how are these quantities connected. To us, these quantities seem imcommensurable.
We would have preferred less recommendations but condensing them makes them less specific, and finally we would end up with some vague truisms which would not be helpful at all.

**Planned Action:** none

(2.6) **Review: Looking deep inside the paper, I can see interesting aspects about trying to define a common paradigm to interpret data coming from a large variety of satellite data processors. However, this objective is somewhat lost among unnecessary details of retrieval schemes, methodological issues [...]**

**Reply:** The retrieval schemes and methodical issues belong to the core content of the review paper. Without understanding the underlying simplifying assumptions of a retrieval scheme, it is difficult, perhaps even impossible, to provide a

reliable error estimate.

**Planned Action:** The introduction will be rewritten to make the purpose of the paper clearer.

**(2.7) Review: [...] and what I could call a silent but insistent criticism to Optimal Estimation.**

**Reply:** We neither endorse nor dispraise any particular method but we describe the methods which are currently in use, or whose data products are currently in use. For each method we discuss the underlying assumptions.

**Planned Action:** We will make an explicit statement that the superiority of either maximum-likelihood based or optimal-estimation based retrievals cannot be decided on scientific grounds but is a purely philosophical question.

**(2.8) Review: Furthermore, I think that the format of the present study is much more adequate for a report.**

**Reply:** Reports typically report technical information related to one instrument, processor, etc. We present, in a unified notation, an overview of all methods we are aware of. Thus we think that this paper serves well as an overview paper for the TUNER special issue, because it provides a framework the other papers of the special issue can refer to.

**Planned Action:** none

**(2.9) Review: Concerning retrieval error reporting, the canonical Theory of Statistics has been teaching us (e.g. Kendall and Stuart Vol I, II, III, The Advanced Theory of Statistics, Fourth Edition, 1979) for so many years that the performance of a given statistic or estimator, say $\hat{x}$ , is measured in terms of its mean square error or deviation from the true value, which can be decomposed in variance and bias, namely**
$$E[(\hat{x} - x)^2] = E[(\hat{x} - \bar{x})^2] + E[(x - \overline{(x)})^2]$$
**For the assessment of the root mean square error and its reporting, the consolidated usage is today to share and/or distribute.**
**1. Estimated state (of course) and related retrieval covariance matrix**
**2. Background (state and covariance)**
**3. Averaging Kernels**
**Based on the above items, the performance of any estimator (bias and variance) can be unambiguously quantified. From what I can see, in the end, the above three ingredients are what authors agree with to be the basic items to share. In this respect, a potential list of recommendations, included that of authors, could be made and**

**explained in onetwo pages.**

**Reply:** First a side remark: The fact that canonical theory of statistics relates the performance of a statistic estimator to the true value strengthens our position against GUM. We do agree that the errors of an ensemble of retrievals can be decomposed into the mean square error and the bias, and we use this concept ourselves in order to validate the error estimates. We concede that our list of recommendations is three pages, but it covers issues not mentioned by the reviewer (correlations in other domains; data traffic, and others).

**Planned Action:** none

(2.10) **Review: I have also to say, that authors' recommendation list itself is largely independent of the bulk of the present paper.**

**Reply:** We admit that not all parts of the paper are needed to derive the recommendations, but the information contained in the bulk of the paper is needed to provide the quantities requested by the recommendations.

**Planned Action:** The relation between the recommendations and the rest of the paper will be made clearer.

(2.11) **Review: General Remarks**
**The paper is lacking a correct definition and assessment of bias. Authors seem to identify the random component of the root mean square error as the error or uncertainty of a given retrieval system. What about the bias? What's the strategy they want to set up to estimate it and eventually share with end users?**

**Reply:** It is not true that we identify the random component of the root mean square error as the error or uncertainty of a given retrieval system. We conceive the error or uncertainty as a quantity which is composed of a random part (corresponding to what the reviewer calls root mean square error) and a systematic part (corresponding to what the reviewer calls bias). We state explicitly that the systematic error estimates can be tested using the bias between collocated measurements of independent measurement systems, and that the random part can be tested using the standard deviation of the difference between collocated data from different measurement systems. The bias is commonly defined as a mean difference between the measured value and the true value unless explicitly specified differently. In our paper, when we use the term 'bias' with any other meaning than the mean difference between the measured and the true value, we state explicitly what the mean difference refers to.

**Planned Action:** We will add a paragraph on the bias.

(2.12) **Review: I have found a bit confusing the question about gridin-**

dependent retrievals, which for me is a nonsense, since normally one works with a discretized state vector. Apart from forward model (FM), the bias depends on the given constraints, which are normally griddependent, in the sense that their definition and use is contingent to the way the state vector has been discretized. In effect, for a regularized estimator the bias depends solely on the constraints (again apart from FM biases).[...]

**Reply:** Another contribution to the bias can be calibration issues. The role of the constraint is discussed in Section 6.

**Planned Action:** We will mention that the choice of a prior which is not the expectation of an ensemble the actual measurement is taken from will cause a bias.

(2.13) **Review: This basic aspect has been largely overlooked in the paper, and in fact their recommendations are not consistent with a correct sharing of the root mean square error.**

**Reply:** We recommed that the averaging kernels and priors used shall be communicated to the users. The users can then evaluate the smoothing error on the final grid they use, after evaluating the additional averaging kernel component entailed by the interpolation. Sharing the total error will cause inadequate error estimates after resampling and respective generalized Gaussian error propagation.

**Planned Action:** As said above, the discussion of biases will be expanded.

(2.14) **Review: On the same line, their CoA2 is inconsistent with the idea of root mean square error.**

**Reply:** The intention is to avoid that data users interpolate the smoothing error on a finer grid. Instead they should be provided with all information they need to directly evaluate it on the grid of their choice. Any possible inconsistence with the root mean square error comes only from conceiving the retrieved state as a smoothed estimate of the truth, a conception we do not explicitly endorse. Conceiving the retrieved state as an estimate of the smoothed truth removes this inconsistency.

**Planned Action:** none

(2.15) **Review: Furthermore, I am not sure if it can be implemented, in practice.**

**Reply:** For noise alone, CoA2 can be implemented. It is only the combined noise and smoothing error which causes the problem.

**Planned Action:** none

(2.16) **Review: To streamline my personal thinking, let's suppose $W$ is a suitable interpolation/extrapolation operator, which transforms a given estimator $\hat{x}_{n1}$, defined on a grid with $n_1$ layers, into a new one, say $\hat{x}_{n2}$, defined on a grid with $n_2$ layers, we have**

$$\hat{x}_{n2} = W\hat{x}_{n1},$$

**with $W$ a matrix of size $n_1 \times n_1$. CoA2 requires that, using authors' language,**

$$WS_{x,noise,n1}W^T = S_{x,noise,n2},$$

**where, $S_{x,noise,XX}$ is the error covariance directly retrieved on the grid with XX layers. However, I cannot see how the above condition can be met for any choice of $W$ and $n_1 \leq n_2$ or $W$ and $n_1 \geq n_2$. Atmospheric state vectors are not bandlimited signals, therefore a mere extrapolation/interpolation of a given retrieval from a coarser to a finer grid will not show finer structures of the underlying state. Hence, the above condition would normally not be met.**

**Reply:** We agree with everything above except that the additional error is not part of the noise but of the smoothing error, which, we suggest, should be evaluated newly on the finer grid. The example presented by the reviewer shows perfectly why we insist that noise and smoothing error should be reported separately. For noise alone, CoA2 is fulfilled by using generalized Gaussian error propagation. And again, conceiving the retrieval as an estimate of the smoothed truth removes this inconsistency.

**Planned Action:** none

(2.17) **Review: Condition CoA2 seems to have been set up just to criticize the concept of smoothing error, which is the way Rodgers considers for the bias. Since the bias of the individual, single, retrieval depends on the true value, which is normally not know, Rodgers considers the variability of the true value (variancecovariance) in order to have at least an estimates of the interval in which the bias is expected to range. However, the variability and/or stochastic behaviour of the state vector, which is correctly considered in OE, is overlooked by authors.**

**Reply:** We do not agree. We do not criticize the concept of the smoothing error in general (except for the ambiguity of the underlying interpretations of probability, which we criticize in a very careful and moderate wording). The central point of our criticism is the inclusion of the smoothing error in the total

error, which will lead to inconsistent results after resampling of profiles.

**Planned Action:** none

(2.18) **Review: They say, "natural variability is not a genuine retrieval error". It seems to me that authors purposely mislead statistical error with mistake. Natural variability is what makes our weather to be forecastable, but not exactly predictable. This is why we need statistics to address natural variability.**
**Taking into account the natural variability of the state vector, it is possible to perform an assessment of the estimator's bias, e.g., through the (unfortunately named) smoothing error, whose meaning has been, in fact, completely mislead by authors (see also later when dealing with the smoothing error).**

**Reply:** To us, natural variability explains that the atmospheric state at one time and one place is different from the state at another place and another time. Due to this natural variability we cannot expect that two instruments measuring at different places and/or times will render the same result. Detected differences thus do not hint at any malfunction of one of the instruments or retrieval and thus are not genuine measurement errors. Still, these differences have to be considered in comparisons. The reviewer has torn this quotation out of a very different context in our paper. From the context of Section 6.6, where the quoted statement comes from, it should be very clear what we mean. We do not understand how the reviewer can, on the basis of this text, accuse us to "purposely mislead statistical error with mistake".

**Planned Action:** We will add "[...natural variability] in a sense that the atmospheric state at place $s_1$ and time $t_1$ differs from the one at $s_2$ and $t_2$." And we will give more weight in the text to the regularization bias.

(2.19) **Review: Finally, because of the many issues addressed in the paper, in the end it looks like a confusing revision of Rodgers 2000; a sort of poutpourri of about everything is known today on atmospheric inverse problems: Twomey, Tikhonov, Rodgers, LS, ML.**

**Reply:** Our intention is to cover all relevant (in the sense that data retrieved with these methods are still around) methods within a consistent framework and a common notation. This is a precondition for unified error reporting. While the book of Rodgers (2000) provides an excellent theoretical basis, we apply this theory (and other variants) to the real-world retrieval schemes and investigate which uncertainties are caused by the assumptions and approximations in place. We understand this as a systematic compilation rather than a potpourri. We first lay down the basic theory. Then we discuss how retrieval schemes used in the real world deviate from the idealized theory. Then we discuss all error sources and their relevance. We find that the content is clearly structured, and

goes beyond the content of the available literature in that it treats also the relevant real-world problems.

**Planned Action:** The introduction of the paper will be rewritten to make the purpose of the paper more evident.

(2.20) **Review: Furthermore, the estimator described in Eq. (4) in the text is not rigorously derived from any basic principle of statistics, it is just copied from OE and rewritten by substituting $S_a^{-1}$ with $R$**

**Reply:** From our introduction it should be clear that we do not only consider methods which have a probabilistic interpretation. T. von Clarmann and U. Grabowski, Atmos. Chem. Phys. 7:397-408 (2007), their appendix, have shown that there is even a probabilistic interpretation of Eq 4 with R defined as shown in Eq 5. We do not see what is wrong with putting a method in a more general context.

**Planned Action:** none

(2.21) **Review: Specific Comments**
**Pag. 3. At best, CoA2 is only consistent with the variance component of the estimated error. What they want to do with the bias is not clear. Stand as is, I have doubt CoA2 is effective and can really work.**

**Reply:** Resampling of profiles and associated error propagation works well for all error components (noise, instrumental calibration biases, forward model biases...) except those which depend on the sampling of the $\mathbf{S}_a$ matrix. Thus we insist that the latter should be evaluated on the final grid, using the respective sufficiently resolved covariance matrix.

**Planned Action:** none

(2.22) **Review: Page 4. Section 3.1 This is confusing. Please state exactly why uncertainty cannot be used or why it sounds ambiguous if referred to the root mean square error of an estimator.**

**Reply:** We do not say that 'uncertainty' shall not be used. We say that the claimed difference between 'uncertainty' and 'error' is controversial. And according to GUM (and with respect to this issue we agree with GUM), 'uncertainty' does NOT refer to the root mean square error of an estimator but includes also systematic effects.

**Planned Action:** none

(2.23) **Review: Page 6, Eq (3), I cannot see any point why the unconstrained Least Squares solution should be called "Maximum Likeli-**

hood". This is a misconception. The assumption of Normal pdf is what really qualify the estimator (3). The reason of using ML because it yields LS under normality is untenable; it is like saying that a meteorologist is using Einstein General Gravity (EGG) theory when forecasting the atmosphere with the Newton dynamical equations, because EGG retrieves Newton in the limit of low velocity.

**Reply:** The term 'maximum likelihood' in this context is used by Rodgers (2000) for a solution which is free of formal prior information. And this terminology is consistent with that of Fisher, who coined that term. If we search for a solution of which the probability that it reproduces the (noisy) measurement is largest, we get, by definition, the maximum likelihood solution. If we apply this principle to Gaussian noise, the maximum likelihood solution happens to be the least squares solution. We do not use the maximum likelihood solution because it yields LS under normality but we use least squares because ML plus normally distributed yields least squares. It is agreed - and even conceded by Fisher - that ML does not yield the solution of maximum posterior probability. But what is untenable about it? we do never claim that we consider only methods which have a probabilistic interpretation. And more generally speaking: We do not particularly endorse any of the methods we describe. In this paper, we just describe and characterize them.

**Planned Action:** none

(2.24) **Review: Why do authors not qualify the bias and variance of the estimator?**

**Reply:** Because we have organized the paper such that first the methods are presented, and in Section 6 error estimation is discussed. This seems justified to us, because a lot of the error propagation stuff can be treated in parallel for all the estimators, and touching this issue here would lead to redundancies which would make the paper even longer. Both bias and variance of the estimates depend on many more choices than the estimator alone.

**Planned Action:** none

(2.25) **Review: Why the reader has to wait until section 6, just to see the variance alone of the estimator.**

**Reply:** An estimator does not have a variance, only the estimate has one. There are a lot more sources which contribute to the variance of the estimate than measurement noise. Making an exception for this particular source of variance does not seem adequate to us.

**Planned Action:** none

(2.26) **Review: Page 7, Eq. (4). This is the worst part of the paper. Equation (4) is the OE estimator where $S_a^{-1}$ has been substituted with $R$. In force of this unjustified and adhoc substitution, authors claim that the estimator (4) becomes more flexible and powerful than the OE shown in Eq. (6).**

**Reply:** We do not make such a statement.

**Planned Action:** none

(2.27) **Review: Also, in this case the variance of the estimator has been presented to the reader in instalments; first Eq. (7) and then an incredible jump to go to Eq. (18).**

**Reply:** We find it quite natural to first present the methods and then discuss the error sources. This seems particularly adequate to us since Eq. 18 represents only one component (often not even the leading one!) of the random error.

**Planned Action:** none

(2.28) **Review: In addition, (a) The bias of the estimator is not qualified/assessed/quantified in any part of the document**

**Reply:** The bias caused by the regularization is only one component of the total bias. We do not see any good reason to give it an extra treatment by discussing it in Section 4 while all other bias-generating errors are discussed in Section 6. This would disrupt the logical structure of the paper and may even lead the readers astray because they may think that the bias caused by the regularization term is always the most important one.

**Planned Action:** We will rewrite Section 6.4.5 and will discuss the bias-generating properties of the retrieval approaches there.

(2.29) **Review: b. What is the reason to change $S_a^{-1}$ with $R$? What are the expected improvements?**

**Reply:** We do not claim in this paper that there are improvements. We simply want to systematize existing retrieval methods by presenting them in a common framework and notation.

**Planned Action:** none

(2.30) **Review: c. Why has the TikhonovTwomey regularization $\gamma$-parameter disappeared? That is why not $\gamma R$?**

**Reply:** Thanks for spotting!

**Planned Action:** The equation will be corrected. The text above the equation will be changed to: '[... first order differences matrix,] and $\gamma$ a scaling parameter to control the strength of the regularization".

(2.31) **Review: d. What's the role of $x_a$, and why not $x_0$ as in Eq. (3)?**

**Reply:** It makes a difference with respect to what the solution is smoothed. The solution of Eq (3) (in the linear case or after iteration in a well-behaved case) does not depend on $x_0$. Thus $x_0$ can be freely chosen. The solution of Eq (4) does depend on $x_a$, because the smoothing operator will not smooth the profile but the difference between the profile and the a priori. Thus, $x_a$ cannot be freely chosen.

**Planned Action:** We will point this issue out in the text

(2.32) **Review: e. With $R$ set to any of the suggested matrices, 012 order difference matrices, Eq. (4) is dimensionally inconsistent. The authors seek a protocolindependent of constraints and other assumptions, but they propose to use an estimator, which is dimensionally inconsistent and depending on the units used for the state vector. In which way do they achieve dimensional consistency between the two terms in the squared brackets?**

**Reply:** First of all, we do not propose anything, but we describe methods which are actually in use. And back to the question: By an appropriate definition of $\gamma$ (which has admittedly been missing in the discussion paper) dimensions can easily be included.

**Planned Action:** We will define $\gamma$ in the text and mention its units.

(2.33) **Review: It would be much fairer to say "Equation (4), as well as Eq. (3) (e.g. global fit), has been normally in use for the retrievals from satelliteborne limb sounding and occultation observations. It is here considered because still now many satellite processors rely on it. Or something similar. The description of the various estimators, LS, TT, OE should be as much as neutral and respond to the need to just explain their error characteristics.**

**Reply:** Eq (4) is the algebraic generalization. Both Tikhonov smoothing and optimal estimation are particular instanciations of it. We think that this is a fairly neutral way to present these methods. It shows how themethods are related.

**Planned Action:** none

(2.34) **Review: Page 7, line 30. What do you exactly mean with smoothed? What is a smoothed profile? How smoothing is quantified, and why this is a good property.**

**Reply:** A smoothed profile is a profile where the altitude-to-altitude differences of the profile values are reduced. The question why it is a good property is answered in the second part of the criticized sentence: "thus avoiding unphysical oscillations..."

**Planned Action:** We will modify the sentence as follows: "[...smoothed] in the sense of reduced altitude-to-altitude differences".

(2.35) **Review: In comparison to estimator (3), estimate (4) is biased and the bias structure is determined by $R$, which is grid dependent. So, how the estimated errors can be propagated according to CoA2? What is the solution proposed by authors: just forget about bias?**

**Reply:** If the retrieval is conceived as an estimate of the smoothed truth as discussed in Section 6.4.2 and if the measurement response as discussed in Section 6.4.5 is unity (as it typically is with first order differences Tikhonov regularization) then estimator (4) is bias-free. If $\mathbf{S}_a$ does not equal the (typically unknown) $\langle \vec{x}_{\text{true}} \rangle$, optimal estimation will have a bias. Thus, things are not as simple as they seem to be. We thus think that the bias discussion should not be touched upon passing in Section 3 but should be deferred to Section 6.4.5, where we have the content of Section 6.4.2 available, and where we can discuss the bias issue at more depth.

**Planned Action:** Section 6.4.5 will be rewritten to include the bias issue.

(2.36) **Review: Page 8. Eq (6). Now that the authors have invented $R$, they can say our estimator retrieves the OE estimate if we put $R = S_a^{-1}$, unbelievable!**

**Reply:** We find it quite natural that, when we generalize over formalisms and then specify again, we get the original specification back. We do not see what is wrong about this.

**Planned Action:** none

(2.37) **Review: By the way, to me, to $R = S_a^{-1}$, is the only possible choice, if we want to reach dimensional consistency.**

**Reply:** We disagree. With the correct units (which can be imported via $\gamma$), any $\mathbf{R}$ will be dimensionally consistent, regardless if it has a probabilistic interpretation or not.

**Planned Action:** none

(2.38) **Review: Page 8, paragraph beginning at line 8. This comment seems to stay here just to add some references.**

**Reply:** The fact that in the case of logarithmic retrievals the data characterization also refers to the logarithm of the state value is often overlooked and has already caused some confusion among data users. Thus, we find it appropriate to mention this issue.

**Planned Action:** none

(2.39) **Review: By the way, it is not appropriate for Eq. (6). This is a comment to be added soon after Eq. (5). It does not apply to Eq. (6), in fact, OE elegantly solve the problem of high dynamic range of the state vector, because $S_a$ has the right dimension to properly scale the state vector. As shown in many papers, OE can be solved for the scaled variable $\tilde{x} = S_a^{-1/2}x$, which is equalized to a standardized variate, at each layer.**

**Reply:** We disagree. The caveat regarding the Gaussian probability density function is relevant only if the estimate is given a probabilistic interpretation, i.e., in the context of Eq. (6). And the suggested method using $\tilde{x}$ as a retrieval variable does not solve the problem that, for a variable which mostly has small values but a large natural variability (i.e. large $x_a$), the wings of a Gaussian penetrate wide into the negative. That is to say, optimal estimation assigns positive probability densities to negative temperatures or mixing ratios.

**Planned Action:** We will better highlight the problem of positive probability densities to negative temperatures or mixing ratios.

(2.40) **Review: Page 8, Eq. (7) and discussion after. Here it seems that an essential role in error estimation is played by the variance of the estimator alone, and the bias? Once again, how the bias of estimator (4) is qualified/assessed/quantified?**

**Reply:** Here we neither discuss the variance nor the bias. Both variance and bias include more than only noise and regularization, respectively. Thus, the discussion of both is deferred to Section 6.

**Planned Action:** The bias will be discussed in Section 6.4.5.

(2.41) **Review: Section 5. All is said in this section is today overcome**

**by Simultaneous Retrieval. Section 5 is outofdate and should be totally removed.**

**Reply:** A scientist trying to figure out what the total error budget of HALOE or SOFIE data is, is not much helped by this statement. And for, e.g., infrared spectroscopic instruments with 30-40 data products, represented at tens of altitudes each, and – depending on the instrument type – more than 1000 profiles per day with overlapping lines of sight, simultaneous retrieval of everything is still beyond reach. And if, e.g., spectroscopic data of one species are inconsistent in different parts of the spectrum, simultaneous retrieval can even be worse than a sequential approach.

**Planned Action:** none

(2.42) **Review: Section 5.4.5 Still Onion Peeling?**

**Reply:** As said above: the users of, say, HALOE or SOFIE data are not helped very much by saying "the data providers should have used another retrieval method."

**Planned Action:** none

(2.43) **Review: Section 5.4.6 See point above. I recommend a CoA0: Please forget about adhoc and nonoptimal methods!**

**Reply:**
1. It is the purpose of this paper to get error estimation for existing data sets under control. We are not proposing a data analysis scheme for a future instrument. The reviewer seems to have misunderstood the conditions of adequacy. They are not about retrieval schemes, but for error propagation schemes for given (not necessarily favoured or endorsed) retrieval schemes.
2. Optimal methods are optimal only if a real $x_a$ and a real $S_a$ are available. These are often not available, and many "optimal estimation" retrievals are non-optimal retrievals in disguise. Some of the instruments covered by our study have made measurements of some species for the first time. Where to get the prior and its statistics from in this case? And finally: Who says that the prior which was valid until yesterday is still valid today? Remember the turkey that came to the gate of the enclosure everyday at 9:00 expecting to be fed. This went well until Thanksgiving. But according to the rules of inductive inference, on which optimal estimation is based, the turkey behaved fully rational!
3. Forgetting methods not favoured by the reviewer clashes with comment 2.33, where we are requested to be neutral? (cf 2.33)
4. We understand that science is the generation and aggregation of knowledge. Based on this assumption it is unclear to us how forgetting anything should advance science.

**Planned Action:** none

(2.44) **Review: Sections 6.1 to 6.3 can be summarized under a very short section entitled "Instrument Noise and Forward Model bias"**

**Reply:** First, we have organized Section 6 by causes of the errors and not by random versus systematic errors. This is because the same source of error can show up as the one or the other, according to the retrieval scheme. And second, we do not see how this reorganization should make the section shorter.

**Planned Action:** none

(2.45) **Review: Section 6.4. Authors here simply miss the important point that the Averaging Kernels matrix, A, qualifies and serves to assess the bias error, at least the part coming from the background constraint. In fact, if we take expectations on both side of Eq. (25) all random components associated to the instruments are averaged to zero, and we remain with the expectation value, $E(\hat{x})$. Systematic component, originating from the forward model, can be dealt with appropriate transforms of the radiance vector, e.g., random projections.**

**Reply:** The bias caused by the regularization is dealt with in Section 6.4.5.

**Planned Action:** Section 6.4.5 will be expanded and restructured.

(2.46) **Review: Section 6.4.1. All the verbose premise of the paper points straight to this criticism of the smoothing error. However, the only thing which is fairly criticisable here is the word smoothing. In fact, smooth, smoother and similar terms should be banned from the context of error assessment and analysis. If Rodgers had said the retrieval can be regarded as a biased estimate of the true state, then everything would have gone to the right place. In effect, the smoothing error is the missing bias term to be added to the variance in order to have an estimate of the root mean square error, $E[(\hat{x}-x)^2]$. In principle, there is no need to interpolate/extrapolate to different grids a given state vector for the purpose of comparison. For visual inspection, one can just plot the given estimators and confidence intervals on the same plot, using the proper pressurealtitude grid. Why the quest of plotting differences?**

**Reply:** We do not criticize the smoothing error as such but we criticise that it may be included in the total error and will thus be inappropriately propagated for resampled profiles. We find the claim that interpolation is unnecessary somewhat odd. Science does not only consist of plotting data. Some more quantitative approaches are required. Time series at one altitude, when the original data have a varying altitude grid, quantitative profile comparison as suggested by Rodgers and Connor (J. Geophys. Res. 108(D3):4116, doi10.1029/2002JD002299, 2003) and many more scientific applications need interpolation of the data to a common grid.

**Planned Action:** none

(2.47) **Review: Pag. 27 and 28. Eq. s (28) and (29) can be left to more elaborated comparisons. There is no need to cover this aspect in the present paper.**

**Reply:** Here we agree with reviewer #1 who finds this section paticularly important. Furthermore, these sections are important to understand when regularization can cause a bias and when not. Equations 28 and 29 are essential for these sections.

**Planned Action:** none

(2.48) **Review: Pag. 28. Eq. (30). What do you mean "better resolved"? Please, quantify. The paper is aiming at providing recommendation, this cannot be given in terms of ambiguous qualitative terms.**

**Reply:** This statement refers to situations only where the contrast in the resolution is large. Thus, this statement does not depend on the particular definition of vertical resolution, any of the resolution concepts introduced in Section 6.4.3 will do.

**Planned Action:** We will move Section 6.4.3 before Section 6.4.1. Then we will have the definitions of altitude resolution available and will make reference to these definitions.

(2.49) **Review: Pag. 6.4.3 From section 6.4.3 on, until section 7, the paper appears to be unnecessary long.**

**Reply:** At many instances the reviewer criticizes that the various consequences of regularization are not sufficiently discussed, and here, where these issues are dealt with, the paper is criticized to be too long. We are confused.

**Planned Action:** none

(2.50) **Review: Section 7. As said at the beginning 18 recommendations are too many to be useful.**

**Reply:** As said before, we would have preferred less recommendations but condensing them makes them less specific, and finally we would end up with some

vague truisms which would not be helpful at all.

**Planned Action:** none

(2.51) **Review: Table 1 and Table 2. I do not understand the scope of these two tables. If authors want to provide a list of official L2 data providers, the list is too long since it should show only Agencies. If the authors want to provide a list of the many scientists dealing with Satellite Data Processors, it is too short**

**Reply:** "official L2 providers" and "agencies" are no terms of scientific relevance. And including a "list of the many scientists dealing with Satellite Data Processors" would not be useful either. Our criterion is: we included data processors of which the data are distributed to the scientific community. We think that this is a sensible criterion.

**Planned Action:** none unless we are made aware of further data products that deserve to be included according to the criterion mentioned above.

---

## Referee Report (RR1)

**Review of "Estimating and Reporting Uncertainties in Remotely Sensed Atmospheric Composition and Temperature" by Thomas von Clarmann et al.**

On reading this first time I got the impression that the authors were biassed towards one kind of remote sounding - spectrally resolved limb sounding. It is not until the end of the Recommendations that I found the reason: "These recommendations have been developed from the perspective of mainly satellite-borne limb sounding and occultation observations but some of these concepts are equally applicable to other types of remote sensing missions." This limitation should be stated right at the start, at least in the introduction, if not in the abstract or even the title. Better still they should attempt to write a paper about generic remote sounding.

Another impression I got is that in places there is a bias against the use of regularisation, without mentioning the disadvantages of not using it. This seems to be irrelevant to the apparent aim of the paper, namely objectively developing a unified scheme for reporting the characteristics of remotely sensed data.

Done properly, the description of any information-conserving retrieval in terms of a linearisation point, an averaging kernel and an error covariance matrix contains all of the information of the measurement in a unified and standard way, such that the user does not need to know much about the instrument. It is simply a given linear function of the atmospheric state, about a given linearisation point, together with a given error covariance, valid in so far as the forward function is linear within the error bounds. As such it could be used as data for any kind of further retrieval or transformation that the user may wish to do. (It may be sensible to divide the error covariance into different independent sources in cases where there is temporal correlation between different measurements.) It is not important whether the retrieval is ML or MAP based.

The paper is much too long, though I am at a loss to recommend in detail how it might be shortened. A lot of attention to removing unnecessary detail would help. A couple of sections - Onion Peeling and Chahine's Method seem a bit obsolete. Are they really needed?

**Specific comments**

Page 3 line 8: In what way are the characterisation schemes described in Rodgers 2000 not of general applicability?

Page 3 line 27: It seems to me that this requirement is not possible. The requirement should be that means should be provided for transforming error estimated errors from one grid to another.

Page 4 after section 2: Should there be a requirement that spatial and temporal correlations in errors between retrievals should be described? This is mentioned in the recommendations at the end but I feel it should be in the conditions of adequacy.

Page 4 line 24: sentence starting "The interested reader" and ending on line 29. I got lost in the sentence. Use a list papers and then say a few words about each reference.

Section 3.1: This whole section reminds me of angels dancing on pinheads. Various committees are producing different varieties of camels, to mix metaphors. (This is a complaint about committees, not about this paper)

Page 6 line 3: "two different ways to evaluate this quantity." It seems to me that one of these ways is a way of validating the other.

Page 7 line 7: Insert "an explicit" after "because"

Page 9 line 12: "holds" is a bit vague. How about something like "has a Bayesian interpretation"

Page 10 section 5.1: You should remind the reader here that discretisation itself is a kind of regularisation, and if the discretisation is too fine, further regularisation will needed to deal with the ill-posedness of the inversion.

To say that in a maximum likelihood retrieval, the grid width is identical to the spatial resolution of the retrieval may be formally correct but can be misleading. A fine structure whose amplitude is less than the retrieval noise is not usefully resolved.

Page 12 line 18: "problems" problems is not quite the right word perhaps "considerations" would be more appropriate. These are not problems unless they are ignored, dealing with them is part of the process.

Page 15 line 2: Using layer values instead of level values should not make much difference unless the layers are made thicker. It is simply a different kind of representation with the same number of unknowns.

Page 18 line 7: Marquardt was aware that using  $\lambda I$  lead to various kinds of problems. He suggested using a scaling matrix **D** with different elements down the diagonal. Many variants of Levenberg-Marquardt are available in the literature.

Page 19 line 8: should "inaccurate instrument model" be included in this list or is it implied by item 1?

Page 20 bottom paragraph: Another problem with residual-based noise characterisation is that systematic spectroscopic errors will lead to systematic retrieval errors, whereas random noise will lead to random retrieval errors. The user needs to be able to distinguish these. Equation (19) applies separately to each independent source of error which can be described by a covariance matrix. Examination of the spectral residual is an important part of validation of the instrument error model.

Page 21 rest of 6.1.1: Again, problem (line 2) is not quite the right word. Propagated noise from preceding retrieval steps may be formally dealt with internally as a parameter error, but as far as the data user is concerned it is still noise and should be presented as such.

You need a clear definition *from the users point of view* of what is meant by noise and what is meant by parameter error. It seems to me that anything which is uncorrelated between successive measurements is noise (primarily as a result of detector noise), and anything which is correlated, or even constant, between successive measurements should be classified as a parameter error. From the user's point of view internal details of how you do the retrieval should not matter, but the experimenter needs a formal way of calculating it correctly, the analysis should treat the retrieval as a whole, however it works internally.

My overall impression of section 6 is that it is something of a ragbag, going into too much detail in the case of some kinds of instrument while ignoring other kinds of instrument almost completely. E.g. nadir sounders, radiometers, GPS occultation.

Page 23 line 29: I suggest you replace "but are not part of" by "but are not usually thought of as part of"

Page 23 bottom line: I do not see a list of "the most prominent auxiliary data uncertainties", but just a discussion of pointing.

Page 24 lines 20 and 21: this only applies for some kinds of instrument.

Page 28 lines 24-26: "one": there will normally be an infinite set of atmospheric states which are consistent with the pure measurement information, not just one. This is true regardless of the number of levels used in the retrieval representation. The atmosphere is effectively a continuum. The purpose of the grid is to reduce the dimensionality of the problem, so as to search for a solution in a finite a dimensional sub-space. Thus a grid is just another form of regularisation. Furthermore, there is no particular reason for choosing a sub-space defined by a grid, any nonsingular representation could be chosen. Even if a grid is chosen, the implied interpolation rule must be specified. A linear interpolation, for example, introduces non-physical gradient discontinuities.

Any proper retrieval method with its correct characterisation will conserve information, as long as the problem is not grossly nonlinear, hence there is no "degradation" of the data, and the grid id fine enough to represent all singular vectors of the weighting functions for which there is non-neglible signal to noise. The retrieval can always be adjusted to allow for a required change in (or removal of) prior data, if used. Think of the characterisation as the forward model for a new retrieval.

A retrieval method that does not use a priori can be distorted by large errors and degraded by resulting in retrieval noise so large that the error analysis no longer remains in the nearlinear region. It also discards information available in the measurement by using a too coarse grid.

Page 28 line 25: The meaning of resolution "with respect to the true state of the atmosphere" is unclear. We do not know what the true state of the atmosphere is. Resolution is a property of an instrument, including its noise characteristics, conceptually it is the spacing of two delta functions that can be distinguished from each other, and that depends critically on both the size of the delta functions and the noise of the instrument. Hence we need to know something about the variability of the atmosphere before we can make sensible judgements about resolution.

Page 29 line 29: you should define precisely what you mean by biased. There will only be bias if the prior state and covariance matrix, if that is what is used, are inadequate. Any bias would be towards the prior state only in regions where there is not much real information from the measurement.

It might be worth including somewhere the usual warning about averaging retrievals which include a priori information.

Page 30 line 1: These so-called distortions occur generally in areas where there is no real information in the measurement. The user should be aware of this.

Page 30 line 12: I am unhappy about the use of the word "criticised" in this context. The use of (28) is quite appropriate in its own context. This section is not "criticism" as you usually used in English but rather a discussion of the appropriate context. A discussion of retrieval grids and interpolation in the context of prior data can be found in Rodgers (2000) section 10.3.1. (Note the obvious error in the heading for section 10.3.1.3.)

Page 30 paragraph starting line 19: the interpretation of the matrix  $S_a$  is not relevant to error analysis. The retrieval is some more or less known function of the true state. All that is required for error analysis are the derivatives of this function with respect to quantities that might have error components.

Page 30 line 24-26: I agree completely.

Page 31 lines 1-2: For a priori you have to use what information you have - that's what prior information means. Like the measurement, it needs properly assessing and validating. The better your a priori the better your result will be.

Page 31 line 8: "true": perhaps "appropriate" would be a better word.

Page 31 lines 9-11: If they sample different parts of the atmosphere, you are not going to compare them.

Page 32 line 19 and 20: If profile retrievals are to be assimilated the proper way to use them is the characterisation of the retrieval as a smoothed version of the profile, with the averaging kernel and the appropriate error covariance. This includes all the information in the measurement, and nothing else. The a priori in that case is simply a linearisation point. However modern data assimilation systems use radiances not retrievals. It makes validation and allowance for temporally correlated errors easier.

Page 33 line 10: Please state what you mean by the vertical resolution of the instrument itself. If you mean the vertical spacing of observations for limb sounders then please remember that not all instruments are limb sounders.

Page 33 lines 11-15: The vertical resolution is limited primarily by the physics of the measurement, as expressed by the continuous weighting functions, together with the instrumental noise. A profile representation such as a vertical grid should be chosen not to limit the resolution of the measurement.

It is easy to be misled by the nature of a limb sounder to think that the resolution must be something to do with the measurement spacing, and hence grid spacing (I think this is what you mean, but it isn't clear). Different points in the measured spectrum will have different weighting functions, and consequently it is quite possible for there to be information on a finer vertical scale. Noise matters too, because it can hide the structure you are trying to resolve. To understand the useful resolution of a system you need to have some idea of the size of the structures you want to see.

Alternatively if you mean that you choose a grid spacing so the the problem is formally over-constrained, then you have to choose so that the problem is not unstable, and hence the choice of grid size becomes your method of regularisation. Noise and resolution are intimately linked by a trade-off.

Page 34 line 5: The Backus-Gilbert spread was developed in order to design a retrieval method which optimises resolution; this is why it was designed to suppress sidelobes. I do not understand why you think that a retrieval on a finer grid will produce more pronounced sidelobes.

Page 34 line 18: You can say this more simply by stating that the averaging kernel characterises the vertical resolution of the difference between the retrieved profile and the a priori.

Page 34 line 25: You could append "but can result in noise which makes the retrieval useless"

Page 35 line 1: Rodgers (2000) uses the term degrees of freedom **for** signal. This is associated with the degrees of freedom for noise, and is not particularly to do with retrieval. The sum of the two is the total number of degrees of freedom for the measurements, m.

Degrees of freedom for signal is a good guide to a suitable number of levels to use in a grid, or number of elements in some other representation.

Page 35 line 16: Please define what do you mean by bias and distort.

What normally happens is that in a part of the profile where the ML retrieval has low noise, an MAP retrieval on the same grid gives the same value and the same averaging kernel. Where the noise is large, the MAP retrieval moves smoothly to the a priori, while the ML gives only noise. Which you prefer depends on your application. You can have "bias" or noise.

Page 35 line 23:"Systematically": I presume here you are referring to ensembles of retrievals rather than a single retrieval. Perhaps you should give the usual warning about averaging retrievals containing a priori. I.e. the a priori component should be removed before averaging.

Page 35 line 25: This concept goes back to Backus and Gilbert who use the constraint that the sum of the rows of the average kernel matrix should be unity in developing their retrieval method.

Page 36 line 17: "fake": only if the retrieval method has been badly designed, for example by using an unsuitable prior covariance matrix.

Page 36 line 26: What is the case under discussion?

Page 36 line 29: This line appears to mean "in order to avoid problems due to regularisation, regularisation by means of a coarse discretisation is used".

Page 37 line 7: It is not clear how this would be done.

Page 37 line 15: The English is a bit peculiar here. How about "Even if the prior information can be conceived as the...." See also comment on Page 31 lines 1-2. Any deviation of your estimate of instrument noise covariance from reality is likewise a source of error.

Page 39 last sentence: I strongly agree; should this have been in the conditions of adequacy?

Page 42 line 16: Sensitivity studies don't necessarily have to be based on delta functions, this may sometimes lead to numerical problems. Any kind of suitable complete set of orthogonal functions will do.

Page 42 line 20: I suggest that "ideally the data provider" be replaced by "the data provider must".

Page 42 line 23: I suggest adding somewhere "For retrievals given on a grid, the implied interpolation scheme must be specified".

Page 42 line 31 onwards: If smoothing error is reported, the a priori covariance on which it is based should be given. However I am inclined to think that you should recommend that smoothing error should not be reported, for the reasons you give.

Page 43 line 13: Precisely what is meant by a "profile representation" must be defined somewhere. It not only includes values on a grid, it also includes an implied interpolation between the grid points. This interpolation must be specified.

Page 44 recommendation 18: This does not sound like a recommendation. Either construct a recommendation from this discussion, or separate it as an explicit discussion section.

Page 45 line 5: This mention of satellite limb soundings and occultation observations is important it should not be left as a throwaway comment at the end of the paper, it should be mentioned right at the start. Preferably the recommendations should have been developed from the perspective of remote sounding generically.

**Minor points**

Page 2 line 3 "shall" should be "should".

Page 3 line 20: The sentence starting "We refer to diagnostic metadata" is unclear. I assume it means something like "By diagnostic metadata we mean"

Page 4 line 3 Insert "proper" before combination.

Page 4 line 8: QA4EO should be spelled-out at its first use.

Page 5 line 32: This sentence is unclear, I assume it means "no particular terminology" rather than "not having a terminology".

Page 7 line 21: "m > n" does not necessarily imply over-determined. It is quite possible for the rank of the Jacobian to be less than n if weighting functions are not linearly independent. (I'm nitpicking)

Page 12 line 30: Limb sounding was first used to my knowledge for Mariner Mars in the 1965 in the form of radio occultation sounding

Page 24 line 20: Replace "makes the" by "computes a".

Page 24 line 22: Knowledge of radiative transfer

Page 24 section 6.2.1: Do we need a single subsection within a section? This comment also applies to section 6.4.1.

Page 25 line 23: Replace "along" by "according to" or "by".

Page 34 line 15: Delete comma after retrieval.

Page 37 line 30: Replace "exclude that they all are" by 'assume that they are not"

**Review of "Estimating and Reporting Uncertainties in Remotely Sensed Atmospheric Composition and Temperature" by Thomas von Clarmann et al.**

**Addendum**

Strictly, the averaging kernel and the weighting functions are continuous functions. The discrete versions, with an interpolation rule, are approximations.

In the near-linear case, the continuous averaging kernel is a linear combination of the continuous weighting functions. A fine enough grid should be chosen to approximate it well - the singular vectors of the weighting functions are a good guide. It can be approximated onto a coarse grid such as one used for a ML retrieval, but it cannot be restored by interpolation from that grid, as its fine structure is lost. For example, the fine grid averaging kernel for a coarse grid retrieval with linear interpolation is definitely not triangular, as the interpolation rule might imply.

Data users should be able to access the averaging kernel computed on a fine grid so they can evaluate smoothing error on whatever grid they need. This applies to any retrieval method, not just ML.

---

## Author Response (AR2)

**Referee #2, Review #1:**

**Review:**

**Scientific Significance: good**
**Scientific Quality: good**
**Presentation Quality: good**
**For final publication, the manuscript should be accepted as is.**

**Suggestions for revision or reasons for rejection (will be published if the paper is accepted for final publication):**
**The paper has been improved a lot, although I have to confess that I still do not like the way the paper has been structured. I am still not convinced about their choice not to include the smoothing-bias error in the total error. I would have preferred a more pragmatic approach. Having said that, I have also to confess that this is a very good piece of work and there could be people who could appreciate this work more than I do.**

**Reply:** The authors appreciate the tolerance of the reviewer who suggests acceptance of the manuscript although it does not comply with their personal preference.

**Action:** None.

**Referee #1, Review #2:**

We would like to note that a lot of issues mentioned by the reviewer in the first review and which are included here in quotes have already been solved in our revised version and answered in our initial rebuttal. We do not discuss these issues again here, except if the new review refers to these issues. Instead, we concentrate on the new comments.

**Review: Rating:**
**Scientific Significance: fair**
**Scientific Quality: good**
**Presentation Quality: fair**
**For final publication, the manuscript should be rejected.**

**Suggestions for revision or reasons for rejection (will be published if the paper is accepted for final publication)**
**After reading the paper a second time I must admit that I have even stronger reservations than the first time. Instead of my initial recommendation "major revision" I now disagree with publication in a form which resembles the present structure of the paper. In general I note that the revisions made in response to my comments have been minor, not addressing in full my major concerns.**

Let me clarify my judgement by reviewing my major general comments (between quotes) in my first review:

"My first major reservation: the purpose of the paper is the formulation of the list of recommendations for a more uniform and complete error reporting in level-2 satellite data products (see the last line of the abstract and section 7). However, the bulk of the material presented in the paper is basically a review of real-world implementations of optimal-estimation based (and related) profile retrievals. As such the authors could consider to split the document into two papers, a review of profile retrievals and a shorter more focussed paper about unified error reporting."

"Several sections of the paper are providing useful functional background information for section 7. But for quite some sections I could not find the link with the final recommendations. Examples are section 5.4 and also parts of 4, 5.1 and 6.4 (e.g. 6.4.3 to 6.4.6), 6.7. Because of these sections the paper is very long."

"Section 5.4 is a review of (profile) retrieval approaches, but contains a lot of material which is not directly relevant for the paper. Personally, I would propose to shorten this section, keeping possibly the tables (and references) and keeping those remarks which are important in the context of error reporting. This review-like section also leads to a very long list of references. It would be good to mention only those references that bring new information to the discussion how to present the retrieval errors."

Suddenly the authors claim that the paper is meant to be a review. However, this is completely unclear based on the title and abstract alone. The abstract contains several sentences which would fit better in the bulk of the paper. Only the last sentence of the abstract specifies what the paper will contribute to the existing literature, namely a list of recommendations. This is in contrast with some words that have now been added to the introduction about the paper being meant as a review. The link between the conditions section (2) and the recommendations section (7) on the one hand, and the rest of the paper on the other hand, is still not very strong.

The authors have responded to this major reservation, but by implementing minor fixes like cross-references and clarifications of links between the sections. This does not address my more fundamental observations that the review part of the paper is not serving the recommendations part at the end of the paper.

I would strongly urge the authors to re-think the purpose of the paper. I suggest:

- either a review of profile retrievals (focussing on the error formulation),
- or a shorter recommendation paper on how to present retrieval data products to users,
- or both (two papers).

**In it's present form I think the material presented in the paper is not fully compliant with either a review or a recommendation paper. So this would require major adjustments to the text and the logical structure of the paper. Currently the paper is very long because of the review nature.**

**Reply:** The paper was from the beginning meant as an overview paper of the TUNER special issue. As such it summarizes the existing relevant literature the TUNER work is based on and presents the recommendations developed in the TUNER activity. It is not true that we "suddenly claim that the paper is meant to be a review". For Review Papers in AMT it is mandatory to obtain approval by the Executive Editors prior to submission in order to make sure that the envisaged content fits in with an AMT review paper. We have taken this step on 17 September 2019 and have obtained the approval in an e-mail from Executive Editor Thomas Wagner. And during submission we have clicked "Review paper". There is nothing sudden about our claim that this paper is meant as a review paper.

During the correspondence with the Executive Editors, they were informed about the content of the paper, in particular, that it includes conditions of adequacy, a comprehensive methodical review-like part, and recommendations. The Executive Editors have approved this general outline and have recommended to submit the paper as a review paper. Most likely information about this fundamental decision was not available to the reviewer.

Beyond this more formal argument, we strongly disagree with the statement that the review part does not serve the recommendations part of the paper. Many of the error components at issue in the recommendations part are caused by the approximations and simplifications employed in real-world retrieval schemes. This provides a strong link between Section 5 and the recommendations in Section 7. We think it would be of very limited use to make recommendations as to which error estimates correlations, averaging kernels and so forth shall be reported without explaining to the data provider how to do it. Thus, the knowledge communicated in Section 6 is needed to follow the recommendations in Section 7. We have brought this argument already in our initial rebuttal. Unfortunately, the reviewer did not provide any feedback to our justification, so we do not know with what exactly the reviewer disagrees, and why.

**Action:** It has been made clear in the abstract that the paper contains also a review of common retrieval and error estimation schemes.

**Review:** "After reading the first sections of the paper it was not fully clear to me what is really the problem which is addressed? In what sense are retrieval products not comparable? Please provide (generic) examples of retrieval products which miss information which makes a direct comparison between retrievals, or comparisons with independent data difficult or impossible. In what sense is there a need for a new set of recommendations, e.g. what is missing after the work of e.g. QA4EO or the GUM?"

The authors address this first point with an extra sentence in the introduction, which is useful. However, I would like this to be discussed more systematically/methodologically in the main text with a clear link to the recommendations.

**Reply:** We agree that our work has to be put into the context of existing work, which we do in Section 3.1. Here we explain that existing work is either not targeted at indirect measurements (GUM) or is heavily targeted at formal procedures and workflows while staying extremely vague with respect to the technical and scientific content (QA4EO), at least where instrument-overarching documents are concerned. For example, the concept of the averaging kernels is not even mentioned in the QA4EO documentation, at least not at any detectable place. Their "guide to expression of uncertainty of measurements" is restricted to general rules of error propagation, without any consideration of problems specific to remote sensing of atmospheric composition or temperature, and without consideration of errors resulting from less than ideal, real world retrieval schemes.

**Action:** We have made our discussion of QA4EO more specific. Further we have added examples where non-intercomparable data characterization caused problems in the past.

**Review:** "The final set of recommendations are focussing on profile retrievals. But the tables include also total column retrieval examples (e.g. DOAS). I think this is a missed opportunity, and I would encourage the authors to formulate explicitly what their recommendations are for column retrieval products (some recommendations are generic, but several parts of section 7 explicitly refer to profiles)."

The authors clarify how the formalism (sec 6.4) includes column retrievals as a limiting case. In practice, total column products are organized differently than profile products, and error treatments also differ. Many satellite products are total columns products, so I really miss a more detailed discussion of the variability in total column approaches. The addition of specific recommendations how to harmonise column products would be very useful.

**Reply:** We do not quite understand which point the reviewer wants to make, because the comment is quite vague. In particular, we do not see why column retrievals should be so fundamentally different from profile retrievals that our framework is turned inapplicable. We are somewhat confused by this comment, for the following reasons:

1. The retrieval and error estimation schemes discussed in this paper are fully applicable to column retrievals which do not rely on the assumption of an optically thin atmosphere (e.g. mid IR, SWIR). Also what we write about the different error sources (various measurement errors, parameter errors, radiative transfer modelling errors) is equally applicable to DOAS retrievals assuming an optically thin atmosphere.

2. DOAS is mentioned as an example why our framework should be inapplicable to column retrievals. But DOAS is not limited to column retrievals. This method is used for profile retrievals as well.

3. It is often thought that the use of a differential signal where only the structured part of the measurement provides the information is a unique characteristic of DOAS type measurements. This is, however, not correct: To parametrize the background signal and to obtain the information only from the structured part of the measurement has been a standard approach in, e.g., mid-infrared solar occultation spectroscopy (e.g. ATMOS in the 1980s) and in mid-infrared limb emission spectroscopy since its beginnings (e.g. MIPAS). And we have not been able to identify any reason why our framework should not be applicable to the DOAS concept.

4. Similarly, it is often thought that the two-step retrieval is a unique characteristic of DOAS. Also this is not quite correct. Early IR solar occultation measurements used this approach as well. Beyond this, dividing data analysis in these two steps (step 1: from the measurement to the slant path column amount; step 2 from the slant path column amount to the profile or vertical column) is a mere technical but no logical difference. The Jacobian of the full problem can easily be calculated by application of the chain rule to the two steps. Thus, our formalism is applicable to the two-step retrieval as well.

5. In our paper, also the column averaging kernel has been discussed, as well as the XCO2 retrievals, and the "downwelling factor" approach. Thus, it is not true that the paper is only about profile retrievals and that column retrievals appear only in the tables.

The only point that we missed so far was the discussion of the relevance of the air mass factor and its dependence on the a priori profile. We have now included this issue and make reference to the related literature (Eskes and Boersma, 2003). We admit that this omission was inadequate but it could easily be repaired. We consider it as over-exaggerated to conceive our entire framework as inapplicable to column retrievals only because we missed the air-mass factor issue so far.

The claim of incommensurability of profile and column retrieval uncertainties would be counter-unificationist and would thus counteract the purpose of this paper and the TUNER activity.

**Action** We have included a section on DOAS retrievals. We also have added a section dealing with the parametrization of the background signal. Further we discuss now the Eskes and Boersma variant of the column averaging kernel. And finally we have added some examples of error sources in DOAS-type retrievals.

**Review: "Arguably the atmospheric composition data assimilation community is the main user of satellite retrieval products. This community and their needs are basically not discussed in the paper. More generally the users of the data do not receive much attention, and the requirements are discussed from a L2 data provider point of view. This is my second major reservation. Some parts of the text refer to the validation activities, but this is not presented in a very structured way. The needs and feedback from the validation and assimilation communities on existing L2 satellite products would be an important starting point to discuss requirements for satellite data products. Some assimilation users would prefer to work directly with the level-1b data, an option which is also not discussed."**

**I still find it hard to accept that recommendations are given on how to report the errors without referring to the user community. Different use applications have different needs. The authors mention that a companion paper is addressing this. The reporting is the interface between data provider and user, so both sides should be addressed. As example, there are also very practical considerations like file sizes: error information can easily become by far the largest part of the dataset. The role and activities of the space agencies (NASA / EUMETSAT / ESA / CNSA / JAXA) in unifying data products should also be mentioned.**

**Personally I think a generalised set of recommendations for the L2 retrieval teams that fit all (unspecified) applications of the data is of limited use.**

**Reply:** We disagree. We have boiled down experience of decades of cooperations with and support of data users (including assimilation and validation scientists) into our conditions of adequacy. What we then report is basically the how's and why's of error propagation from level 1 data (calibrated measurements of radiances or transmittances) to level 2 data (profiles of temperature or concentrations or column amounts of trace gases). What the data user does is a transformation of level 2 data to a higher order data product (e.g. differences between modeled and retrieved data, time series, averages, correlations and so forth). The error propagation from level 2 to these higher order products is

a topic in itself and not just another aspect of what we do here. The goal of this paper is to provide guidance how to provide the data user with all diagnostics needed to perform the error propagation from level 2 to higher level data products. How to use these diagnostics properly is beyond the scope of this paper and deserves a paper of its own. It remains unclear why recommendations fitting all (unspecified) applications of the data should be of limited use. We consider the stance that for different user communities or agencies different error propagation laws have to be applied as untenable.

Many of our recommendations aim at bringing down the size of the error data to a realistic size by reporting representative data instead of individual error estimates for each single measurement.

**Action:** None.

**Review: "The recommendations in section 7 are not always formulated as a recommendation, but leave room for interpretation and implementation. I sometimes found the CoA points in section 2 even more clear and explicit than the recommendation points. It may be useful to split the list in section 7 in actual (strong) recommendations and related discussion points. Sometimes it is not so clear what is actually recommended by the authors, e.g. due to a trade-off between completeness and data volume, or aspects are left to the retrieval teams to decide (e.g. point 1, 2, 3, 4, 16, 18)."**

**This is acknowledged by the authors, but the number of recommendations has not changed, and the authors have not made a division in recommendations and related discussion points. I still think that reducing the number would make the concluding recommendations more useful.**

**Reply:** The reviewer does not pinpoint one single recommendation as obsolete, inadequate or not useful. It is the mere number which is criticised. As already said in our initial reply, reducing the number of recommendations would make them less specific and more vague.

**Action:** The discussion points have been separated from the recommendations.

**Review: "I was expecting recommendations also regarding the naming (see section 3). The authors discuss in particular "error" versus "uncertainty", but do not really provide a clear guidance on what to use. Also, the consistency or inconsistency with the GUM activity are not clear to me after reading the section. The reader is referred to a paper in preparation. Retrieval data files contain parameters labelled as "precision", "accuracy", "trueness" etc. and different guidelines exist from different space agencies and for different application areas.**

It would be useful if the authors can discuss naming conventions also in this paper and express a clear opinion/recommendation."

The authors reply that this would always be in conflict with part of the community. I had hoped for a stronger recommendation. For instance in the data assimilation community there have been papers devoted to unified naming and notation. Could the work of Rodgers not serve as starting point, since much of the retrieval formalism was discussed in a systematic way by him? I think there is a need for recommendations on which word means what.

**Reply:** Our first recommendation reads "The language and notation used to describe the error budget must be clearly defined. This can be accomplished either by explicit definitions of all terms and symbols used or by reference to any available document that lays down a self-consistent terminology. **We hope that this paper serves that purpose and that the terminology and notation introduced here will be found useful.**" This is a recommendation as required, however in modest and polite words. However, who are we to dictate anybody which language to use? We consider quibbling about words as futile.

**Action:** We have added a footnote "In the scientific community, it is often desirable to have a citable source regarding notation and terminology so as to be consistent. The authors do not want to dictate what language to use and thus do not provide such a recommendation about the notation and terminology in this paper. That decision is left to the reader."

**Review:** "Machine learning approaches are getting more and more popular and deserve some special attention. Several machine learning implementations for retrievals are limited on the error information they provide. It would be useful to have some targeted recommendations for these approaches as well."

A discussion on machine learning is added in the final discussion section. The bulk of the paper addresses more traditional optimal estimation type profile retrievals based on full radiative transfer models. It is not clear to me (and likely to the reader) to what extent the recommendations are general and apply to important classes of alternative approaches as well, such as the various forms of machine learning which will become much more prominent in the future, or popular approaches such as DOAS."

**Reply:** Since our first revision we have distinguished two variants of ML/AI approaches: Their application to the forward problem, in combination with a conventional retrieval, or their direct application to the inverse problem. We make a clear statement (p.42 l.8-10 in the first revision) that in the first case

everything said on error estimation still applies. We then make suggestions how to deal with error estimation in the second case. And finally we state "In either case it seems important to us that the data user is provided with the same full data characterization as required for the conventional retrieval schemes." We suspect that the reviewer has missed this, because otherwise it would have been clear that the recommendations are general.

**Action:** None.

**Summary Reply:** The general content and ouline of our paper and its submission as a review paper have been approved by the Executive Editors of AMT. The reviewer's comments appear to be entirely based on personal preference and programmatic considerations rather than recommendations on technical or scientific content. The paper presents the debated, agreed, and consolidated views of many authors covering a wide part of the community that we still believe better represent the programmatic presentation of this work.

[revised manuscript text omitted]

---

## Author Response (AR3)

**Reply to Review #3**

We appreciate the extremely knowledgeable, thorough and constructive review. The reviewer has helped identify parts of the paper where we were unclear in our description and wording, and has helped us better express our intended messages. The review has helped us a lot to substantially improve the paper, and the contributions included in the review exceed by far what one usually expects from a review. Thanks a lot to the reviewer for their tremendous work!

**Comment: Review of "Estimating and Reporting Uncertainties in Remotely Sensed Atmospheric Composition and Temperature" by Thomas von Clarmann et al.**
**On reading this first time I got the impression that the authors were biassed towards one kind of remote sounding - spectrally resolved limb sounding. It is not until the end of the Recommendations that I found the reason: "These recommendations have been developed from the perspective of mainly satellite-borne limb sounding and occultation observations but some of these concepts are equally applicable to other types of remote sensing missions." This limitation should be stated right at the start, at least in the introduction, if not in the abstract or even the title. Better still they should attempt to write a paper about generic remote sounding.**
**Reply:** We are perhaps trying to burn the candle at both ends. On the one hand we mention our perspective in the introduction, and make clear already there that a certain focus of the paper is on limb. On the other hand we include some additional examples covering other than spectrally resolved limb measurements. We agree there's a lot of overlap between different approaches, and we try to mention what we can. This review paper essentially tries to find the balance between an algorithm description document for a single approach and a textbook that could cover all approaches.
**Action:** 1. Added to paragraph 3 of the introduction: "Major parts of this work have been carried out from the perspective of passive satellite-borne limb sounding and occultation observations, which accounts for a bias of the examples presented towards these techniques. The underlying theoretical considerations, however, should be applicable to a wider context."
2. Further examples covering other than spectrally resolved limb measurements have been included.

**Comment: Another impression I got is that in places there is a bias against the use of regularisation, without mentioning the disadvantages of not using it. This seems to be irrelevant to the apparent aim of the paper, namely objectively developing a unified scheme for reporting the characteristics of remotely sensed data.**
**Reply:** We are sorry to have given the impression of being biased against regularization. Most of the authors use and endorse regularization to analyze their

data, and we fully agree that regularization is necessary in many cases.

**Action:** We have changed the wording in places where our original text might have given the impression that we are biased against regularization, and we have added arguments as to why regularization is necessary in many cases. In particular, we have included a criticism of non-regularized coarse-grid retrievals in the "Discretization" section. For further changes of the text carried out to remove this bias, see our replies to the specific comments.

**Comment: Done properly, the description of any information- conserving retrieval in terms of a linearisation point, an averaging kernel and an error covariance matrix contains all of the information of the measurement in a unified and standard way, such that the user does not need to know much about the instrument. It is simply a given linear function of the atmospheric state, about a given linearisation point, together with a given error covariance, valid in so far as the forward function is linear within the error bounds. As such it could be used as data for any kind of further retrieval or transformation that the user may wish to do. (It may be sensible to divide the error covariance into different independent sources in cases where there is temporal correlation between different measurements.) It is not important whether the retrieval is ML or MAP based.**

**Reply:** We fully agree that the error reporting should be such that the data user should not have to care about details of the instrument or retrieval scheme. This is stated in our third condition of adequacy. For the data provider, however, the instrument characteristics and retrieval schemes are necessary for performing the error estimation properly, because the usual idealizations and simplifications used give rise to additional errors.

**Action:** None.

**Comment: The paper is much too long, though I am at a loss to recommend in detail how it might be shortened. A lot of attention to removing unnecessary detail would help. A couple of sections - Onion Peeling and Chahine's Method seem a bit obsolete. Are they really needed?**

**Reply:** The situation is that there are still data generated with onion peeling or Chahine's method in use (e.g. HALOE of SOFIE for onion peeling). We still have it on our TUNER agenda to provide the relevant diagnostic data for these. This is why we describe these schemes. These descriptions are not meant as recommendations for these methods. We have detected a silly LaTeX error: One subsubsection was by error a subsection, and this messed up the entire hierarchy and organization of Section 6. With a correction of this error in place, we think that the paper is clearly enough structured that the reader can easily skip sections they are not interested in.

**Action:** We have removed the paragraph on the subjective vs. frequentist conception of $\mathbf{S}_a$, because this issue is not referred to elsewhere in the paper. Further, we have removed some superficial statements on data assimilation.

**Comment: Specific comments**
**Page 3 line 8: In what way are the characterisation schemes described in Rodgers 2000 not of general applicability?**
**Reply:** It was not our intention to say that the characterisation schemes described in Rodgers 2000 are not of general applicability. The point we wanted to make was that the characterization schemes were largely discussed in the context of maximum a posteriori retrievals.
**Action:** Changed to "Importantly, however, our discussion of the data characterization is done in the context of retrieval schemes beyond those endorsed by Rodgers, including many in every day use among remote sounding teams."

**Comment: Page 3 line 27: It seems to me that this requirement is not possible. The requirement should be that means should be provided for transforming error estimated errors from one grid to another.**
**Reply:** For all errors except the smoothing error, we have $\mathbf{S}_{\mathrm{new}} = \mathbf{T}\mathbf{S}_{\mathrm{old}}\mathbf{T}^T$, where $\mathbf{T}$ is the grid transformation matrix, and CoA #2 is fulfilled. The smoothing error is an exception because the generalized Gaussian error propagation law does not apply. We have rephrased CoA #2, using the suggested wording.
**Action:** Rephrased: "The estimated errors should be independent of the vertical grid in the sense that correct application of the established error propagation laws to the transformation of the data from one grid to another yields the same error estimates as the direct evaluation for a retrieval on the new grid would do. For characterization data not fulfilling this criterion, means should be provided for transformation from one grid to another."

**Comment: Page 4 after section 2: Should there be a requirement that spatial and temporal correlations in errors between retrievals should be described? This is mentioned in the recommendations at the end but I feel it should be in the conditions of adequacy.**
**Reply:** The CoAs are presented very early in the paper when none of the statistical and technical concepts have been introduced yet, and they are formulated in a quite non-technical language. Thus, a CoA dealing with correlations in different domains seems to be somewhat out of context. Furthermore, the need of the description of these correlations follows from CoAs #3 and #5.
**Action:** We have added to CoA #3: "The error budget and characterization data shall contain all necessary information needed by the data user to use the data in a proper way. [The error budget shall be useable without detailed technical...]"

**Comment: Page 4 line 24: sentence starting "The interested reader" and ending on line 29. I got lost in the sentence. Use a list papers and then say a few words about each reference.**
**Reply:** Agreed
**Action:** We now list the references first. Then we give a short description of each, as suggested.

**Comment: Section 3.1: This whole section reminds me of angels dancing on pinheads. Various committees are producing different varieties of camels, to mix metaphors. (This is a complaint about committees, not about this paper)**
**Reply:** We agree.
**Action:** None, since neither angels nor camels do fall in our area of expertise.

**Comment: Page 6 line 3: "two different ways to evaluate this quantity." It seems to me that one of these ways is a way of validating the other.**
**Reply:** We fully agree, and this is what we have written at the end of the paragraph: "This study focuses chiefly on ex ante error estimation. To validate these estimates, ex post error estimation is relevant, as expounded, e.g., by Keppens et al. (2019)".
**Action:** None.

**Comment: Page 7 line 7: Insert "an explicit" after "because"**
**Reply:** Agreed
**Action:** "an explicit" inserted as suggested.

**Comment: Page 9 line 12: "holds" is a bit vague. How about something like "has a Bayesian interpretation"**
**Reply:** Agreed
**Action:** Changed as suggested.

**Comment Page 10 section 5.1: You should remind the reader here that discretisation itself is a kind of regularisation, and if the discretisation is too fine, further regularisation will needed to deal with the ill-posedness of the inversion.**
**Reply:** Agreed.
**Action:** The first paragraph of this section has been rewritten: "At least on macroscopic scales, atmospheric state variables are construed as continuously varying in space and time. In the retrieval equations they are, however, represented by vectors with a finite number of elements. A frequent discretization is the representation of the atmospheric state at a limited number of gridpoints. The profile shape between these gridpoints depends on the interpolation scheme chosen. Often profiles are conceived as piecewise linear. The finite grid can be conceived as a surrogate regularization because it places a hard constraint on the shape of the profile between two gridpoints. If the discretization is too fine, a stronger regularization is needed to fight ill-posedness of the inversion, while a too coarse discretization can cause errors in the radiative transfer modelling and limits the spatial resolution of the solution. Also the abrupt gradient changes tend to be more and more unphysical the coarser the grid is. In a maximum likelihood retrieval, the grid-width is identical to the theoretical spatial resolution of the retrieval. However, if the gridwith is chosen too fine, the useful

resolution of the maximum likelihood retrieval will be much worse because the fine structures of the profile will be masked by the noise.

Alternatively, vertical profiles can be conceived as a set of layers, each represented by layer-averages of atmospheric state values and/or partial column amounts of species. In this case no assumptions on the profile shape between the layer boundaries are obvious but they would be implicit because the details of the averaging may depend on the them."

**Comment: To say that in a maximum likelihood retrieval, the grid width is identical to the spatial resolution of the retrieval may be formally correct but can be misleading. A fine structure whose amplitude is less than the retrieval noise is not usefully resolved.**
**Reply:** Agreed
**Action:** The entire paragraph has been rewritten, see above.

**Comment: Page 12 line 18: "problems" problems is not quite the right word perhaps "considerations" would be more appropriate. These are not problems unless they are ignored, dealing with them is part of the process.**
**Reply:** Agreed
**Action:** 'problems' replaced with 'considerations'.

**Comment: Page 15 line 2: Using layer values instead of level values should not make much difference unless the layers are made thicker. It is simply a different kind of representation with the same number of unknowns.**
**Reply:** Not necessarily. In a limb retrieval where the $n$ levels are chosen according to the tangent altitudes, there are usually $n$ layers, the uppermost one ranging from the uppermost tangent altitude to the assumed top of the atmosphere. In a level representation, an $n+1$st level is needed at the upper boundary of the uppermost layer. In an unconstrained ML retrieval this can cause instabilities.
**Action:** Added: "[...is prone to instabilities] because $n$ layers go along with $n + 1$ neighbouring levels."

**Comment: Page 18 line 7: Marquardt was aware that using $\lambda$I lead to various kinds of problems. He suggested using a scaling matrix D with different elements down the diagonal. Many variants of Levenberg-Marquardt are available in the literature.**
**Reply:** Thanks for the info.
**Action:** The text around Eqs 15 and 16 has been rewritten: "[... Levenberg (1948) and Marquardt (1963) suggested a method that limits the stepwidth $\vec{x}_{i+1} - \vec{x}_i$ and turns it towards the direction of the steepest descent of the object function. The simplest formulations of this scheme are, for unconstrained and constrained inverse problems, respectively,
(Eq. 15)

and (Eq. 16)

where $\lambda$ is a scalar that is adjusted during the iteration according to the local non-linearity of $\mathbf{F}$ and $\mathbf{I}$ is unity. There exist many variants of this approach, particularly with respect to the dynamical choice of $\lambda$ and the rescaling of the problem to avoid problems associated with the $\lambda\mathbf{I}$ term, first recognized by Marquardt (1963). Marks and Rodgers (1993) and Rodgers (2000) suggest the following variant:"

**Comment: Page 19 line 8: should "inaccurate instrument model" be included in this list or is it implied by item 1?**
**Reply:** Currently these are covered under 6.1.3. under "Instrument Characterization Errors". They indeed are missing in the numbered list. We shy away from adding another item to the list because the numbers in the list are meant to correspond to the numbers of the subsequent subsections. We have, however, discovered that this envisaged structure has been disrupted somehow.
**Action:** The structure has been fixed to make subsection numbers consistent with the numbers of the list. The text "[errors caused by less than perfect measurements, which include ...] and a less than perfect characterization of the instrument by the instrument model," was added to the list under #1.

**Comment: Page 20 bottom paragraph: Another problem with residual-based noise characterisation is that systematic spectroscopic errors will lead to systematic retrieval errors, whereas random noise will lead to random retrieval errors. The user needs to be able to distinguish these.**
**Reply:** Agreed
**Action:** We have inserted: "[...nor does it allow for decomposition of the error budget into its components.] In particular, it will not be possible to separate random error components from systematic components. Residual-based error analysis [is suitable ...]".

**Comment: Equation (19) applies separately to each independent source of error which can be described by a covariance matrix. Examination of the spectral residual is an important part of validation of the instrument error model.**
**Reply:** We do agree. However, the general error propagation formula is presented already in Eq. 18. Thus we add some text there instead near Eq. 19.
**Action:** Text slightly rephrased, following the suggestion: "[... Eq. 18], which is applicable separately to each independent error that can be described by a covariance matrix $\mathbf{S}_q$ and represents the uncertainty of an input variable of $\vec{q}$. $\mathbf{S}_r$ is the error covariance matrix of the output variable $\vec{r}$, and $\mathbf{J}$ is the Jacobian with elements $\partial r_i/\partial q_j$." Further, we have added near the end of the paragraph on residual-based noise estimates: "In turn, the analysis of the fit residuals is an important means to validate error estimates."

**Comment: Page 21 rest of 6.1.1: Again, problem (line 2) is not quite**

**the right word. Propagated noise from preceding retrieval steps may be formally dealt with internally as a parameter error, but as far as the data user is concerned it is still noise and should be presented as such.**

**Reply:** Agreed.

**Action:** We have replaced the term 'problem' with 'complication' and have added "[... is formally dealt with as parameter error,] although from a user perspective it is still noise."

**Comment: You need a clear definition from the users point of view of what is meant by noise and what is meant by parameter error. It seems to me that anything which is uncorrelated between successive measurements is noise (primarily as a result of detector noise), and anything which is correlated, or even constant, between successive measurements should be classified as a parameter error. From the user's point of view internal details of how you do the retrieval should not matter, but the experimenter needs a formal way of calculating it correctly, the analysis should treat the retrieval as a whole, however it works internally.**

**Reply:** We've been classifying things from the point-of-view of the cause, not the effect. Section 6 is about error sources (i.e. the causes) and how the data provider can deal with them. The user may not care where the noise estimates came from, but the provider needs to understand the process to properly provide the information. We agree that a definition of the term 'noise' has to be provided, however, for us noise does not encompass all random error components but only those arising from indeterministic processes (scene noise due to emission of photons of the atmosphere is an indeterministic process, and detector noise because the emission of photons by the instrument itself is an indeterministic process, and similar). The user need only be concerned with what is random and what is systematic, and thus this issue is covered by the recommendations.

**Action:** The beginning of Section "Measurement noise" has been rephrased to include a definition: "Although often 'measurement noise' is conceived as all errors which are uncorrelated in succesive measurements, we use a narrower definition. In our terminology, noise encompasses only the statistical uncertainty of the measured signal caused by the indeterministic or unpredictable nature of radiative processes in the atmosphere or the instrument. Measurement noise is described by the error variance of each single spectral data point. The uncertainties are considered as ..."

**Comment: My overall impression of section 6 is that it is something of a ragbag, going into too much detail in the case of some kinds of instrument while ignoring other kinds of instrument almost completely. E.g. nadir sounders, radiometers, GPS occultation.**

**Reply:** We concede that, due to a technical LaTex error which disrupted the hierarchy of subsections and subsubsections, it was not clear how Section 6 was

organized. We have fixed this, and we think, with this technical correction in place) Section 6 is clearly structured. We take the reviewer to be criticising the biased selection of examples. We admit this bias, but it is related to illustrative examples only but not to the general scheme behind them. The authors of this paper working on nadir instruments have confirmed that the general formalism presented is perfectly applicable to the class of nadir instruments they are dealing with.

**Action:** The technical LaTex error has been fixed to make the organization clear again. The first five subsections in Section 6 deal with the 5 different classes of errors introduced in the ordered list in main Section 6.

In the introduction the following statement has been added: "Major parts of this work have been carried out from the perspective of passive satellite-borne limb sounding and occultation observations, which explains a bias of the examples presented towards these techniques."

**Comment: Page 23 line 29: I suggest you replace "but are not part of" by "but are not usually thought of as part of"**
**Reply:** Agreed.
**Action:** Changed as suggested.

**Comment: Page 23 bottom line: I do not see a list of "the most prominent auxiliary data uncertainties", but just a discussion of pointing.**
**Reply:** We agree.
**Action:** Further examples have been added. Care has been taken to select further examples that are not only applicable to limb spectrometric measurements.

**Comment: Page 24 lines 20 and 21: this only applies for some kinds of instrument.**
**Reply:** The part "by integration over the finite field of view and by convolution with the spectral instrument response function" should not be understood as a general statement but only as an example. We admit that this was not clear in the text.
**Action:** Rephrased: "The radiative transfer model used in the retrieval solves the radiative transfer equation and usually involves an instrument model which makes the signal comparable to what the instrument would see. Depending on the instrument type, the instrument model will include the integration of the radiance field over the finite field of view, the convolution with the spectral instrument response function, etc."

**Comment: Page 28 lines 24-26: "one": there will normally be an infinite set of atmospheric states which are consistent with the pure measurement information, not just one. This is true regardless of the number of levels used in the retrieval representation. The atmosphere is effectively a continuum. The purpose of the grid is to reduce the dimensionality of the problem, so as to search for a solution in a finite a dimensional sub-space. Thus a grid is just another form of regu-**

larisation. Furthermore, there is no particular reason for choosing a sub-space defined by a grid, any nonsingular representation could be chosen. Even if a grid is chosen, the implied interpolation rule must be specified. A linear interpolation, for example, introduces non-physical gradient discontinuities.

**Reply:** We do agree with the spirit of the reviewer's comment here but we are not sure if this is the best place to state these concerns. This might be better fit in with the "Discretization" Section. We prefer to discuss these issues there and to include here only a short reminder. The specification of the interpolation is important and will be mentioned in the recommendations. For here, slight rephrasing should suffice.

**Action:** We have changed "the one" to "that one of the discrete profiles". The general discussion of this problem is now included in the 'Discretization' section. We have added to recommendation #12 "The discretization must be specified. If the retrieval is reported as state value over altitude, the applicable interpolation rule must be reported."

**Comment: Any proper retrieval method with its correct characterisation will conserve information, as long as the problem is not grossly nonlinear, hence there is no "degradation" of the data, and the grid is fine enough to represent all singular vectors of the weighting functions for which there is non-neglible signal to noise. The retrieval can always be adjusted to allow for a required change in (or removal of) prior data, if used. Think of the characterisation as the forward model for a new retrieval.**

**Reply:** We admit that our wording sounded dismissive.

**Action:** Instead of "As a consequence, this can introduce additional bias and distortion, and the resolution can be degraded with respect to the true state of the atmosphere beyond the degradation caused by the finite retrieval grid." we now write "Depending on the regularization chosen, the profile can be pushed towards the a priori profile and the vertical resolution can be worse than what the gridwidth might suggest."

**Comment: A retrieval method that does not use a priori can be distorted by large errors and degraded by resulting in retrieval noise so large that the error analysis no longer remains in the near-linear region. It also discards information available in the measurement by using a too coarse grid.**

**Reply:** We do agree, but the purpose of this paragraph is not a pro/con discussion of regularization but shall merely justify why averaging kernels are useful and necessary. We think that with the rewording (see previous comment) it should now be clear that we do not intend to dispraise regularization.

**Action:** None.

**Comment: Page 28 line 25: The meaning of resolution "with respect to the true state of the atmosphere" is unclear. We do not know**

what the true state of the atmosphere is. Resolution is a property of an instrument, including its noise characteristics, conceptually it is the spacing of two delta functions that can be distinguished from each other, and that depends critically on both the size of the delta functions and the noise of the instrument. Hence we need to know something about the variability of the atmosphere before we can make sensible judgements about resolution.

**Reply:** In our reworded text the term "resolution with respect to the true state of the atmosphere" no longer occurs.

**Action:** See above.

**Comment: Page 29 line 29: you should define precisely what you mean by biased. There will only be bias if the prior state and covariance matrix, if that is what is used, are inadequate. Any bias would be towards the prior state only in regions where there is not much real information from the measurement.**

**Reply:** This subsection is not only about optimal estimation but about regularized retrieval in general. Depending on the *ad hoc* regularization chosen, a lot of funny behaviour of the retrievals is thinkable. Even if stochastic a priori profiles and covariances are chosen in the sense of a maximum a posterioi retrieval, the a priori represents the best knowledge of a rational agent but not necessarily the (unknown) true frequentistic distribution of true states. Even optimal estimates in the MAP sense would be bias-free only if these two distributions were the same. This, however, cannot be taken for granted. Furthermore, we think that the statement "Any bias would be towards the prior state only in regions where there is not much real information from the measurement." is only correct in a retrieval grid which is coarse enough. If the retrieval grid is fine enough not to limit the resolution of the retrieval, there will always be a sensible influence by the a priori. Conversely, if there is no influence of the formal a priori, the stability of the retrieval is due to the coarse discretization.

**Action:** None

**Comment: It might be worth including somewhere the usual warning about averaging retrievals which include a priori information.**

**Reply:** This is already mentioned in the Section "Related Issues" as the third item in the list.

**Action:** None.

**Comment: Page 30 line 1: These so-called distortions occur generally in areas where there is no real information in the measurement. The user should be aware of this.**

**Reply:** The point we want to make here is that averaging kernels can be asymmetric. Such averaging kernels have a tendency to move a signal up or down in altitude. We will mention that symmetry of averaging kernels is an important criterion.

**Action:** Added "If the averaging kernel is asymmetric, [the resulting profile

shape can be distorted...]".

**Comment: Page 30 line 12: I am unhappy about the use of the word "criticised" in this context. The use of (28) is quite appropriate in its own context. This section is not "criticism" as you usually used in English but rather a discussion of the appropriate context. A discussion of retrieval grids and interpolation in the context of prior data can be found in Rodgers (2000) section 10.3.1. (Note the obvious error in the heading for section 10.3.1.3.)**
**Reply:** Indeed von Clarmann (2014) does not criticise the smoothing error as such but its inclusion in the error budget.
**Action:** Rephrased: "While in principle this formulation is a direct consequence of generalized Gaussian error propagation, the inclusion of the smoothing error in the reported error budget has been critically discussed by von Clarmann (2014). His main argument [refers to the fact that...]

**Comment: Page 30 paragraph starting line 19: the interpretation of the matrix $\mathbf{S}_a$ is not relevant to error analysis. The retrieval is some more or less known function of the true state. All that is required for error analysis are the derivatives of this function with respect to quantities that might have error components.**
**Reply:** We still think that this paragraph is correct but it is not very relevant to any other part of the paper, thus we have decided to delete it.
**Action:** This paragraph has been deleted.

**Comment: Page 30 line 24-26: I agree completely.**
**Reply:** We are glad to hear this.
**Action:** None.

**Comment: Page 31 lines 1-2: For a priori you have to use what information you have – that's what prior information means. Like the measurement, it needs properly assessing and validating. The better your a priori the better your result will be.**
**Reply:** We do agree. We merely present an additional criterion which qualifies a good choice of prior information. The requirement that the actual case must be a member of the distribution defined by mean $\vec{x}_\mathrm{a}$ and covariance $\mathbf{S}_\mathrm{a}$ is not enough. We recognize that the example we have presented is not optimally intuitive to illustrate what we mean.
**Action:** Rephrased: "For example, to calculate the smoothing error of a tropical ozone profile retrieval we cannot use a global ozone climatology, although the particular tropical profile can be conceived as a member of the distribution formed by the global average and the global variance. To calculate the smoothing error of a tropical ozone retrieval, we must not use an ozone climatology built from a whole year of global – including polar – ozone data, because this would over-estimate the ozone variability. The most specific reference class will be a homogeneous reference class whose internal variability is, as far as known,

purely random."

**Comment: Page 31 line 8: "true": perhaps "appropriate" would be a better word.**
**Reply:** Agreed.
**Action:** Reworded as suggested.

**Comment: Page 31 lines 9-11: If they sample different parts of the atmosphere, you are not going to compare them.**
**Reply:** Agreed; we mean that both sample roughly the same part of the atmosphere; or at least that there is an overlap. E.g. a stratospheric profile from a limb sounder can be used as prior information for a nadir retrieval which is chiefly sensitive to the troposphere.
**Action:** Rephrased [The use of actual measurement data] of the same part of the atmosphere [from independent sources].

**Comment: Page 32 line 19 and 20: If profile retrievals are to be assimilated the proper way to use them is the characterisation of the retrieval as a smoothed version of the profile, with the averaging kernel and the appropriate error covariance. This includes all the information in the measurement, and nothing else. The a priori in that case is simply a linearisation point. However modern data assimilation systems use radiances not retrievals. It makes validation and allowance for temporally correlated errors easier.**
**Reply:** Mentioning data assimilation here opens another can of worms, and to tackle this at adequate depth is far beyond the scope of this paper. The point we want to make is that in many quantitative comparisons we use an object function similar to the $\chi^2$ (validation, hypothesis testing, whatever...). In the comparison of a coarsely resolved profile with a better resolved reference profile acting as 'surrogate truth', the object function will be over-estimated. We have too options to solve this problem.

1. We can make the differences in the numerator of the object function smaller degrading the resolution of the reference data by application of the averaging kernel of the coarser resolved profile. This is our interpretation of "the retrieved state as an estimate of the smoothed truth"

2. We can 'subtract apples from oranges' in the numerator but must account for the artificially large difference by in the denominator by means of consideration of the smoothing error. Here the surrogate truth is not touched. This is our interpretation of "the retrieved state as a smoothed estimate of the true state."

Any application of Eq. 31 is associated with option #1. To mention data assimilation in this context seems to raise more questions than it answers. Thus we have decided to delete this remark. The direct assimilation of radiances is mentioned elsewhere in the paper.

**Action:** We have deleted "In data assimilation the averaging kernel has to be included in the observation operator"

**Comment: Page 33 line 10: Please state what you mean by the vertical resolution of the instrument itself. If you mean the vertical spacing of observations for limb sounders then please remember that not all instruments are limb sounders.**
**Reply:** We agree that this statement is misleading. We have reworded the statement without using the term 'vertical resolution of the instrument itself'.
**Action:** Rephrased "Vertical resolution of the retrieval, not to be confused with the vertical sampling implied by the tangent altitude increments or the instantaneous field of view of a limb sounder, describes the ability to distinguish..."

**Comment: Page 33 lines 11-15: The vertical resolution is limited primarily by the physics of the measurement, as expressed by the continuous weighting functions, together with the instrumental noise. A profile representation such as a vertical grid should be chosen not to limit the resolution of the measurement.**
**Reply:** We agree that the wording 'is limited' is inadequate.
**Action:** Rewritten: "It [the vertical resolution] cannot be better than the width of the vertical retrieval grid on which the results are presented. The latter should thus be be chosen not to limit the resolution of the measurement. All information on the vertical resolution is included in the averaging kernel matrix."

**Comment: It is easy to be misled by the nature of a limb sounder to think that the resolution must be something to do with the measurement spacing, and hence grid spacing (I think this is what you mean, but it isn't clear). Different points in the measured spectrum will have different weighting functions, and consequently it is quite possible for there to be information on a finer vertical scale. Noise matters too, because it can hide the structure you are trying to resolve. To understand the useful resolution of a system you need to have some idea of the size of the structures you want to see.**
**Reply:** This is exactly what we mean.
**Action:** We think that our reworded "not to be confused...' remark above clarifies this.

**Comment: Alternatively if you mean that you choose a grid spacing so the the problem is formally over-constrained, then you have to choose so that the problem is not unstable, and hence the choice of grid size becomes your method of regularisation. Noise and resolution are intimately linked by a trade-off.**
**Reply:** We fully agree but we think that this is better stated in the section on "Discretization".
**Action:** We have mentioned in the "Discretization" section that a coarse grid will act as a regularization.

**Comment: Page 34 line 5: The Backus-Gilbert spread was developed in order to design a retrieval method which optimises resolution; this is why it was designed to suppress sidelobes. I do not understand why you think that a retrieval on a finer grid will produce more pronounced sidelobes.**
**Reply:** We observe more pronounced side-lobes of the averaging kernel on a finder grid. They are not an artefact but are the response of the retrieval to perturbations on a finer scale. These side lobes are simply not resolved by the coarse-grid averaging kernel. The coarse grid averaging kernel does not describe the responses to the individual fine grid perturbations but only the common response to the aggregate perturbations in the coarse grid. This leads to a cancellation of the side lobes. The account presented in the addendum of the review fully supports this view: If we conceive the coarse grid averaging kernel as a superposition of the fine grid averaging kernels, the cancellation takes place.
**Action:** Added: "[... on a coarser grid.] If we conceive the coarse-grid averaging kernel as a superposition of finde-grid averaging kernels, the side- lobes cancel out. [The Backus-Gilbert spread is...]"

**Comment: Page 34 line 18: You can say this more simply by stating that the averaging kernel characterises the vertical resolution of the difference between the retrieved profile and the a priori.**
**Reply:** Yes, that's exactly what we mean.
**Action:** The text has been reworded as suggested.

**Comment: Page 34 line 25: You could append "but can result in noise which makes the retrieval useless"**
**Reply:** We fully agree.
**Action:** Appended as suggested.

**Comment: Page 35 line 1: Rodgers (2000) uses the term degrees of freedom for signal. This is associated with the degrees of freedom for noise, and is not particularly to do with retrieval. The sum of the two is the total number of degrees of freedom for the measurements, m. Degrees of freedom for signal is a good guide to a suitable number of levels to use in a grid, or number of elements in some other representation.**
**Reply:** Thanks for the information. We might have misunderstood some content of a conversation handed down. This information makes the footnote obsolete.
**Action:** The footnote has been deleted and the text has been adjusted.

**Comment: Page 35 line 16: Please define what do you mean by bias and distort.**
**Reply:** The definition of the bias is given in line 17 (of the old version of the

paper. It is $\langle \hat{\vec{x}} - \vec{x} \rangle_{\text{regul.}}$. We realize that we do not discuss the distortion issue any further in this Section, thus we should not mention it at the outset of this Section. This issue, which is related to asymmetrical averaging kernels, is discussed elsewhere in the paper.

**Action:** "or distort it" deleted.

**Comment: What normally happens is that in a part of the profile where the ML retrieval has low noise, an MAP retrieval on the same grid gives the same value and the same averaging kernel. Where the noise is large, the MAP retrieval moves smoothly to the a priori, while the ML gives only noise. Which you prefer depends on your application. You can have "bias" or noise.**

**Reply:** This paragraph does not only refer to MAP retrievals only but to regularized retrievals in general, with any, possibly *ad hoc*, regularization. The purpose of this paragraph is not to dispraise any method but to offer tools to characterize the result in quantitative terms. We have, however, identified a logical flaw in the structure of this section and have thus rewritten it. Again, agree that "an MAP retrieval on the same grid gives the same value and the same averaging kernel" but in this case the regularization is again hidden in the discretization.

**Action:** Modified: "If the a priori profile $\vec{x}_a$ deviates from the expectation value of the true profiles $\langle \vec{x} \rangle$, the regularized retrievals may be biased. The component of the bias related to regularization, $\langle \hat{\vec{x}} - \vec{x} \rangle_{\text{regul.}}$ is

$$\langle \hat{\vec{x}} - \vec{x} \rangle_{\text{regul.}} = \langle (\mathbf{I} - \mathbf{A})(\vec{x}_a - \vec{x}) \rangle \tag{1}$$

We have to distinguish two cases: Firstly, smoothing can cause biases, because, e.g., a sharp maximum of the true profile will always be reproduced too low, and the wings of the maximum will be reproduced too high. This typically occurs when a smoothing constraint as discussed after Eq. (5) is used in combination with an unstructured a priori profile. This type of bias,...

**Comment: Page 35 line 23: "Systematically": I presume here you are referring to ensembles of retrievals rather than a single retrieval. Perhaps you should give the usual warning about averaging retrievals containing a priori. I.e. the a priori component should be removed before averaging.**

**Reply:** We agree.

**Action:** Changed to plural ("result**s**") to make clear that we talk about ensembles of retrievals.

**Comment: Page 35 line 25: This concept goes back to Backus and Gilbert who use the constraint that the sum of the rows of the average kernel matrix should be unity in developing their retrieval method.**

**Reply:** Thanks for information!

**Action:** We now state only that the first application to **atmospheric** sciences

was by Eriksson (2000) and Baron et al. (2002).

**Comment: Page 36 line 17: "fake": only if the retrieval method has been badly designed, for example by using an unsuitable prior covariance matrix.**
**Reply:** We still think that our argument is valid but the wording was too harsh.
**Action:** Reworded: " ... where air density is low will correspond to unrealistic mixing ratios. Thus, averaging kernels evaluated in units of number density can make the dependence of the result on values at higher altitudes appear large although the underlying perturbation in terms number density does not represent any realistic condition. These unrealistic large contributions from higher layers inhibit the correct interpretation of the measurement response."

**Comment: Page 36 line 26: What is the case under discussion?**
**Reply:** We agree that this is unclear.
**Action:** Rephrased: "The regularized joint fit lies between..."

**Comment: Page 36 line 29: This line appears to mean "in order to avoid problems due to regularisation, regularisation by means of a coarse discretisation is used".**
**Reply:** Agreed.
**Action:** Reworded as suggested.

**Comment: Page 37 line 7: It is not clear how this would be done.**
**Reply:** First a coarse grid is defined. A preceding regularized retrieval on a fine grid provides the information how fine the coarse grid for a stable maximum likelihood retrieval can be chosen, and where the new gridpoinds shall be situated. In the simplest case the gridpoints of the coarse grid are a subset of the gridpoints of the fine grid. In the next step a regularization matrix is built which emulates a coarse grid retrieval along with its associated interpolation scheme in the fine grid. Using Rodgers Eq. 10.48 the old regularized retrieval is transformed to the new regularization matrix. The new result still is represented on the fine grid, but all values between the coarse gridpoints are fully determined by the interpolation scheme. Thus, it is sufficient to report the values at the coarse gridpoints along with the regularization scheme.
**Action:** We have added this information; we have further included the following statement: "The new representation is neither considered as generally superior over the old one which involves explicit regularization nor is it meant to supersede it. It is meant only as an alternative representation where time-variant averaging kernels cause problems.".

**Comment: Page 37 line 15: The English is a bit peculiar here. How about "Even if the prior information can be conceived as the...." See also comment on Page 31 lines 1-2. Any deviation of your estimate of instrument noise covariance from reality is likewise a source of error.**
**Reply:** Thanks for the rewording suggestion. We agree on the noise covariance

issue, but this belongs in Section 'Instrument characterization'. While in most cases instrument noise is quite reliably known, often much less reliable information is available on $\mathbf{S}_a$.

**Action:** Reworded as suggested. Remark added to instrument characterization section: "Wrongly estimated instrument noise will not only lead to incorrect error estimates but will directly affect the results. The reason is twofold. First, each element of the measurement vector $\vec{y}$ is weighted by its uncertainty, and distorted weights can lead to different results. And second, incorrect noise estimates will change the weights of the a priori information and the information contained in the measurement."

**Comment: Page 39 last sentence: I strongly agree; should this have been in the conditions of adequacy?**
**Reply:** The conditions of adequacy are non-technical. All the technical consequences go into the recommendations.
**Action:** We extend CoA #3 in a sense that the user should be provided with all necessary information. From this the related recommendation can be inferred.

**Comment: Page 42 line 16: Sensitivity studies don't necessarily have to be based on delta functions, this may sometimes lead to numerical problems. Any kind of suitable complete set of orthogonal functions will do.**
**Reply:** Agreed
**Action:** 'delta' deleted.

**Comment: Page 42 line 20: I suggest that "ideally the data provider" be replaced by "the data provider must".**
**Reply:** "Ideally..." might be too weak; "must" might be too strong (who are we to tell others what they must do?)
**Action:** As a compromise between the extremes we now write " The data provider should...".

**Comment: Page 42 line 23: I suggest adding somewhere "For retrievals given on a grid, the implied interpolation scheme must be specified".**
**Reply:** Excellent suggestion!
**Action:** Included as suggested.

**Comment: Page 42 line 31 onwards: If smoothing error is reported, the a priori covariance on which it is based should be given. However I am inclined to think that you should recommend that smoothing error should not be reported, for the reasons you give.**
**Reply:** Agreed. Since the smoothing error is a very established diagnostic, we initially shied away from such an extreme recommendation. However, since the recommendation not to include the smoothing error in the error budget does not attack the concept of the smoothing error as such, we agree to change the

recommendation.

**Action:** We now write: "The smoothing error should not be included in the error budget. Instead the data users should be provided with the averaging kernel matrices calculated on a sufficiently fine grid allowing them to evaluate the smoothing error on their working grid to which they will transform the data."

**Comment: Page 43 line 13: Precisely what is meant by a "profile representation" must be defined somewhere. It not only includes values on a grid, it also includes an implied interpolation between the grid points. This interpolation must be specified.**
**Reply:** We fully agree.
**Action:** We have included this in R12" "The discretization must be specified. If the retrieval is reported as state value over altitude, the applicable interpolation rule must be reported."

**Comment: Page 44 recommendation 18: This does not sound like a recommendation. Either construct a recommendation from this discussion, or separate it as an explicit discussion section.**
**Reply:** Agreed
**Action:** The recommendation now reads: "The error estimates should explain observed differences between measurements of the same airmass" and the old text of the recommendation has been moved to the commenting text beneath the recommendation.

**Comment: Page 45 line 5: This mention of satellite limb soundings and occultation observations is important it should not be left as a throwaway comment at the end of the paper, it should be mentioned right at the start. Preferably the recommendations should have been developed from the perspective of remote sounding generically.**
**Reply:** Agreed.
**Action:** A respective remark has been included in the introduction.

**Comment: Minor points**
**Page 2 line 3 "shall" should be "should".**
**Reply:** Agreed
**Action:** Changed as suggested.

**Comment: Page 3 line 20: The sentence starting "We refer to diagnostic metadata" is unclear. I assume it means something like "By diagnostic metadata we mean"**
**Reply:** Yes, this is what we want to say.
**Action:** Rephrased as suggested.

**Comment: Page 4 line 3 Insert "proper" before combination.**
**Reply:** Agreed
**Action:** 'proper' inserted.

**Comment: Page 4 line 8: QA4EO should be spelled-out at its first use.**
**Reply:** Thanks for spotting.
**Action:** QA4EO spelled out.

**Comment: Page 5 line 32: This sentence is unclear, I assume it means "no particular terminology" rather than "not having a terminology".**
**Reply:** Agreed.
**Action:** Rephrased 'No particular terminology...'

**Comment: Page 7 line 21: "$m > n$" does not necessarily imply over-determined. It is quite possible for the rank of the Jacobian to be less than n if weighting functions are not linearly independent. (I'm nitpicking)**
**Reply:** Well, this seems a matter of definition what 'overdetermined' is. If we understand our tutorial material correctly, a system of equations is over-determined if $m > n$, regardless of the rank. However, there may be other definitions around. To avoid quibbling about words we rephrase this sentence without using the term 'over-determined'.
**Action:** Rephrased without using the term 'over-determined'.

**Comment: Page 12 line 30: Limb sounding was first used to my knowledge for Mariner Mars in the 1965 in the form of radio occultation sounding**
**Reply:** Thanks for this info.
**Action:** We restrict our statement to "optical limb sounding of the Earth's atmosphere"

**Comment: Page 24 line 20: Replace "makes the" by "computes a".**
**Reply:** This statement has already been rewritten, following an earlier comment.
**Action:** No further action.

**Comment: Page 24 line 22: Knowledge of radiative transfer**
**Reply:** Thanks for spotting!
**Action:** Corrected.

**Comment: Page 24 section 6.2.1: Do we need a single subsection within a section? This comment also applies to section 6.4.1.**
**Reply:** Oops, something went wrong with the hierarchy of sections, subsections, and subsubsections. 6.3 should be 6.2.2, 6.3.1 should be 6.2.3, etc, and 'Parameter errors' should be 6.3. With this correction in place, the subsection numbers correspond to the numbers of the ordered list on pages 18/19.
**Action:** (Old) Subsection 6.3 degraded to a subsubsection.

**Comment: Page 25 line 23: Replace "along" by "according to" or "by". Reply:** Agreed.
**Action:** Corrected.

**Comment: Page 34 line 15: Delete comma after retrieval.**
**Reply:** Thanks for spotting.
**Action:** Comma deleted.

**Comment Page 37 line 30: Replace "exclude that they all are" by "assume that they are not"**
**Reply:** Agreed.
**Action:** Changed as suggested.

**Comment Review of "Estimating and Reporting Uncertainties in Remotely Sensed Atmospheric Composition and Temperature" by Thomas von Clarmann et al. Addendum**
**Strictly, the averaging kernel and the weighting functions are continuous functions. The discrete versions, with an interpolation rule, are approximations.**
**In the near-linear case, the continuous averaging kernel is a linear combination of the continuous weighting functions. A fine enough grid should be chosen to approximate it well – the singular vectors of the weighting functions are a good guide. It can be approximated onto a coarse grid such as one used for a ML retrieval, but it cannot be restored by interpolation from that grid, as its fine structure is lost. For example, the fine grid averaging kernel for a coarse grid retrieval with linear interpolation is definitely not triangular, as the interpolation rule might imply.**
**Data users should be able to access the averaging kernel computed on a fine grid so they can evaluate smoothing error on whatever grid they need. This applies to any retrieval method, not just ML.**
**Reply:** Thanks a lot for this valuable account. We will use this in the text after Eq. 25 and in the recommendations.

[revised manuscript text omitted]